# A unified framework to estimate the origins of atmospheric moisture and heat using Lagrangian models

Jessica Keune[1], Dominik L. Schumacher[1], Diego G. Miralles[1]

[1]Hydro-Climate Extremes Lab (H-CEL), Ghent University, Ghent, 9000, Belgium

*Correspondence to*: Jessica Keune (jessica.keune@ugent.be), Dominik Schumacher (dominik.schumacher@ugent.be)

**Abstract.** Despite the existing myriad of tools and models to assess atmospheric source–receptor relationships, their uncertainties remain largely unexplored and arguably stem from the scarcity of observations available for validation. Yet, Lagrangian models are increasingly used to determine the origin of precipitation and atmospheric heat, scrutinizing the changes in moisture and temperature along air parcel trajectories. Here, we present a unified framework for the process-based evaluation of atmospheric trajectories to infer source–receptor relationships of both moisture and heat. The framework comprises three steps: (i) diagnosing precipitation, surface evaporation and sensible heat from the Lagrangian simulations, and identifying the accuracy and reliability of flux detection criteria, (ii) establishing source–receptor relationships through the attribution of sources along multi-day backward trajectories, and (iii) performing a bias correction of source–receptor relationships. Applying this framework to simulations from the Lagrangian model FLEXPART, driven with ERA-Interim reanalysis data, allows us to quantify the errors and uncertainties associated with the resulting source–receptor relationships for three cities in different climates (Beijing, Denver and Windhoek). Our results reveal large uncertainties inherent in the estimation of heat and precipitation origin with Lagrangian models, but they also demonstrate that a source- and sink bias-correction acts to reduce this uncertainty. The proposed framework paves the way for a cohesive assessment of the dependencies in source–receptor relationships.

## 1 Introduction

There exists a variety of moisture tracking models aiming at determining the source regions of precipitation, i.e., the land or ocean area from which moisture available for precipitation originally evaporates (Gimeno et al., 2012). These models have been frequently used to estimate the rainfall that originates from evaporation in the same region (often referred to as the precipitation recycling ratio; Brubaker et al., 1993; Trenberth, 1999). These models include 1D (e.g., Budyko, 1974) and 2D analytical models (e.g., Brubaker et al., 1993; Eltahir and Bras, 1996; Dominguez et al., 2020), Eulerian models (e.g., Goessling and Reick, 2011; van der Ent et al., 2014), model-internal water vapor tracers (e.g., Koster et al., 1986; Bosilovich and Schubert, 2002; Sodemann et al., 2009; Knoche and Kunstmann, 2013; Singh et al., 2016; Insua-Costa and Miguez-Macho,

2016), and Lagrangian models (e.g., Stohl and James, 2004; Dirmeyer and Brubaker, 2007; Sodemann et al., 2008; Stein et al., 2015; Sprenger and Wernli, 2015; Miltenberger et al., 2013; Tuinenburg and Staal, 2020). The latter have gained interest in recent years due to their ability to define atmospheric trajectories in space and time. Whereas all Lagrangian models are subject to uncertainties arising from the accuracy of the modelled trajectory pathway, among other model-specific parameters (such as the employed convection scheme and the number of parcels being tracked), the setup of these models can fundamentally differ. Some trace 'water parcels' of equal mass that are released with each evaporation event and lose moisture during precipitation events (e.g., Tuinenburg and Staal, 2020; Dirmeyer and Brubaker, 2007), while others trace 'air parcels' and their properties, such as water vapor content but also density and temperature; the latter is the approach followed by models such as FLEXPART (Stohl et al., 2005; Pisso et al., 2019) or LAGRANTO (Wernli and Davies, 1997; Sprenger and Wernli, 2015).

Tracking air parcels enables the state of the atmosphere and its changes in space and time to be inferred, and thus facilitates the estimation of the origin of precipitation and heat (Schumacher et al., 2019). Contrary to the tracking of water parcels, which are released when processes (such as evaporation) take place, the tracking of air parcels requires the estimation of surface evaporation ($E$) and precipitation ($P$) based on changes in water vapor content ($q$) from air parcels, which are distributed homogeneously in space and time. Utilizing the atmospheric mass balance of water, Stohl and James (2004) demonstrated that $E$ and $P$ can be approximated by aggregating the relative moisture changes of all parcels residing above an area and at a certain time step, identifying positive and negative sums as either $E$ ($E–P > 0$) or $P$ ($E–P < 0$), respectively. Further tracing air masses backward in time, source regions of moisture could be inferred: prescribing a maximum trajectory length, typically set to the average or maximum residence time of water vapor in the atmosphere (see e.g., Gimeno et al., 2020), regions with positive contributions ($E–P > 0$) illustrate the qualitative source regions of moisture. Ever since, this approach has become the standard for a multitude of studies (e.g., Drumond et al., 2019; Stojanovic et al., 2018; Ramos et al., 2019; Sorí et al., 2017; Miralles et al., 2016; Vázquez et al. 2016; Nieto et al., 2014). Yet, this approach remains merely qualitative: all air parcels, regardless of their location in or above the atmospheric boundary layer (ABL), are evaluated and their net moisture gain or loss over a time step is interpreted as either evaporation ($E–P > 0$) or precipitation ($E–P < 0$), respectively. As such, precipitation and evaporation cannot coexist at the same time step. Source regions of precipitation are, furthermore, subject to the maximum length of the trajectory, which needs to be prescribed by the user. While this trajectory length can be calibrated to minimize precipitation errors (Nieto and Gimeno, 2019), moisture losses between source and sink regions are not accounted for. In particular, precipitation *en route* between the identified source locations and the sink region leads to a distorted picture of the source locations, thus precluding a mass-conserving, quantitative analysis of source regions and recycling ratios.

To overcome the restrictions of this qualitative perspective, Sodemann et al. (2008) introduced a process-based analysis of air parcel trajectories. If enough air parcels are tracked, parcels can be filtered according to the processes they undergo: moisture increases in parcels within the (well-mixed) ABL respond to surface evaporation during that time step. Furthermore, following

the convection parameterization by Emanuel (1991), air parcels with a relative humidity larger than 80% and decreasing moisture content contribute to a precipitation event. Parsing parcels accordingly enables process-based tracking, and not only permits precipitation and evaporation to co-exist at one time step, and over a region or grid cell, but also facilitates the quantification of rain *en route* by discounting prior source region contributions. This discounting is frequently done in a linear manner, assuming well-mixed conditions (Sodemann et al., 2008). This quantitative approach renders the calibration of trajectory lengths obsolete and enables the estimation of the lifetime (or residence time) of water vapor in the atmosphere (Läderach and Sodemann, 2016). Nonetheless, the approach by Sodemann et al. (2008) still requires the definition of thresholds, such as a minimum moisture increase for evaporation to be identified. This threshold may be calibrated for the respective study regions to filter for noise arising from a large number of parcels. Recently, following the approach from Sodemann and Stohl (2009) and Winschall et al. (2014), Fremme and Sodemann (2019) and Sodemann (2020) relaxed these requirements and considered that also parcels above the ABL may be indirectly affected by surface evaporation through moist convection and mixing. In recent years, variations of these process-based approaches have been frequently applied, as it makes a mass-conserving attribution of the source region contribution to precipitation possible (e.g., Sodemann and Zubler, 2010; Martius et al., 2013; Winschall et al., 2014; Sun and Wang, 2014; Chen and Luo, 2018; Zhou et al., 2019; Keune and Miralles, 2019).

Today, despite all the efforts to converge towards a common understanding of source–receptor relationships, the reliability and uncertainty inherent in existing attribution methodologies remains largely unaddressed. This lack of information partly relates to the sparsity of observations that can be used to validate the origin of moisture (such as isotope measurements), the magnitude of the resulting fluxes, or the lifetime of moisture in the atmosphere. The latter, in particular, has been the subject of intense discussion in recent years due to the large discrepancies shown by existing approaches (Läderach and Sodemann, 2016, van der Ent and Tuinenburg, 2017). Sodemann (2020) recently argued that these discrepancies relate in part to the definition of lifetime, and proposed to employ the distribution of the residence time of moisture in the atmosphere. This highly skewed distribution, where few source contributions greatly exceed the average residence time, is better represented by the median — as further reconciled by Gimeno et al. (2021). Nevertheless, there is a ubiquitous lack of uncertainty quantification in literature studies, and the few intercomparisons of moisture tracking methods that exist often remain restricted to individual events (Winschall et al., 2014) and demonstrate large discrepancies (Hoyos et al., 2018). Consequently, to advance our knowledge on source–receptor relationships, a systematic and standardized evaluation of the reliability and uncertainty of the applied approaches should become a priority.

Moreover, while a multitude of models and tools is readily available to assess source–sink relationships of atmospheric moisture, fewer studies assess sensible heating and heat transport. Nonetheless, early studies already tracked latent and sensible heating in space and time and paved the way for Lagrangian model analyses beyond moisture. For example, Whitaker et al. (1988), Reed et al. (1992) and Rossa et al. (2000) illustrated the importance of latent heating for the development of cyclones.

More recently, Pfahl et al. (2015) unravelled the role of latent heat release for the formation of atmospheric blocking. Further, Bieli et al. (2015) demonstrated that Lagrangian models may be used to identify mechanisms associated with temperature extremes: an air parcel's temperature increase may be caused by adiabatic descent, radiative processes or heating from the surface. Analogously, a temperature decrease along a trajectory may be caused by adiabatic ascent and radiative cooling.

Recently, a number of studies employed this knowledge to study large-scale processes that determine heatwave temperatures: Quinting and Reeder (2017) illustrated that near-surface heatwave temperatures over Australia are influenced by adiabatic descent and diabatic heating of air masses outside the heatwave region. Similarly, Zschenderlein et al. (2019) found that the intensity of European heatwaves is largely influenced by subsidence and diabatic heating near the surface few days before the event. Focusing on the spatial heatwave propagation, Schumacher et al. (2019) identified the terrestrial 'origin of heat', i.e.,

the regions in which surface sensible heating leads to a temperature increase in the overlying air parcels; the term *heat source regions* was used in analogy to the moisture source regions. In their study, Schumacher et al. (2019) evidenced the exacerbating impact of upwind droughts on downwind heatwaves via heat advection. Despite these efforts to embrace the study of heat transport using Lagrangian models, the combined evaluation of heat and moisture remains largely unstudied. Only in a follow-up study, Schumacher et al. (2020) illustrated the merits of a combined diagnosis of heat and moisture source regions to study

the impact of dry and hot air advection on ecosystem productivity; nonetheless, the uncertainties inherent in the methodology remained largely unaddressed.

Here, we aim to advance this study field by assessing the uncertainty and reliability of heat and moisture source–receptor relationships emerging from Lagrangian models. To do so, we target two objectives. First, we introduce validation measures

that allow us to infer the accuracy inherent in the source region estimation. Second, we unravel the uncertainty associated with the evaluation of Lagrangian trajectories to establish source–receptor relationships. In particular, we evaluate the uncertainty associated with the process detection criteria, the assumption of well-mixed conditions, and the bias correction. To achieve these objectives, we introduce a unified framework for the process-based evaluation of atmospheric trajectories. The framework comprises three steps (see Fig. 1), (i) a global diagnosis of relevant fluxes (surface evaporation and sensible heat,

as well as precipitation) from Lagrangian trajectories, and the quantification of errors associated with this diagnosis, (ii) the construction of source–receptor relationships from multi-day trajectories via the attribution of sources following mass- and energy-conserving algorithms, and (iii) the bias correction of these source–receptor relationships with the diagnosed flux errors from the first step. Moreover, in the first part of this manuscript, we introduce new criteria to diagnose surface fluxes via sensible and latent heating in a coupled manner (i.e., heat dependent on moisture and moisture dependent on heat). We also

quantify the accuracy and reliability of these coupled criteria, enabling a comparison to already existing criteria, such as those proposed by Sodemann et al. (2008), Fremme and Sodemann (2019), Schumacher et al. (2019), and Schumacher et al. (2020). The identification of errors associated with these detection criteria facilitates a bias-correction of source–receptor relationships. In the second part, we evaluate the impact and associated uncertainties of several attribution algorithms and bias-correction methods. Therefore, we introduce a novel attribution methodology, and compare it to the previously proposed linear

130 discounting of moisture losses between a source region and the sink; and thereby evaluate the impact of the well-mixed assumption. Finally, this paper concludes with a discussion of the uncertainties inherent in the evaluation of Lagrangian simulations and provides a summary of the results.

## 2 Methods and data

135 This section describes the framework and the generic workflow for the evaluation of air parcel trajectories from a Lagrangian model (Sect. 2.1). The diagnosis part describes heat and moisture diagnosis criteria as well as measures to validate their accuracy and reliability using two-step trajectories (Sect. 2.2). Further, the estimation of source regions and contributions using multi-step trajectories is presented (Sect. 2.3) and both source and sink bias correction methods are described (Sect. 2.4), before information on the selected model for this study is provided (Sect. 2.5).

 For simplicity, the framework presented here refers to a typical 'backward' analysis, i.e., one identifies all parcels residing over an area of interest — a receptor or sink region — and follows these parcels backward in time to estimate their moisture and heat origins. However, the methodology is equally applicable to 'forward' analyses to determine the fate of surface evaporation or sensible heat. Note that we refer to *receptor* region when discussing heat and moisture advection, while the use

145 of *sink* region is reserved for the context of precipitation. Further, we note that the diagnosis of fluxes is based on simulations with a spatially homogeneous distribution of parcels that represent the entire atmospheric mass within a regional- to global domain — this domain-covering tracking of air masses facilitates the bias-correction of source fluxes, that is not feasible with heterogeneous distributions of air parcels (e.g., from point-scale simulations). However, the framework is also largely applicable to Lagrangian simulations that only track air or moisture from specific locations.

150 **2.1 Workflow**

 The framework can be subdivided into three steps (Fig. 1): (1) diagnosis, (2) attribution and (3) bias-correction. In the first step, all air parcels from a Lagrangian simulation are evaluated independently over two consecutive time steps. This analysis enables the detection and quantification of processes, such as precipitation, evaporation and sensible heating (see Sect. 2.2). As all parcels and time steps are evaluated, a global data set of process detection accuracy and reliability can be constructed

155 (see Sect. 2.2.2) and be used for bias-correction in the third step (see Sect. 2.4). In the second step, air parcels residing over the region of interest (i.e., the receptor region) are filtered to construct backward trajectories: air parcels are traced backward in time and source locations along those backward trajectories are determined using the process-based detection criteria from the first step. To achieve mass- and energy-conservation along trajectories, all losses of moisture or heat between a source location and the receptor region must be considered, and the corresponding source location contributions adjusted accordingly

160 (see Sect. 2.3). For moisture, source contributions may be constrained by means of a receptor quantity, e.g., their contribution to precipitation in the sink region can be estimated. Similarly, moisture source contributions can be scaled to match the

integrated water vapor over a receptor region (not shown here). The resulting relative contributions of each source location to the total receptor or sink quantity may depict biased estimates of the source–receptor (or source–sink) relationships as determined with the Lagrangian simulation. In a third step, these relationships are bias-corrected, employing the accuracy dataset from the first step (see Sect. 2.4). Here, both source and sink/receptor quantities can be used to bias-correct the source–sink/receptor relationships.

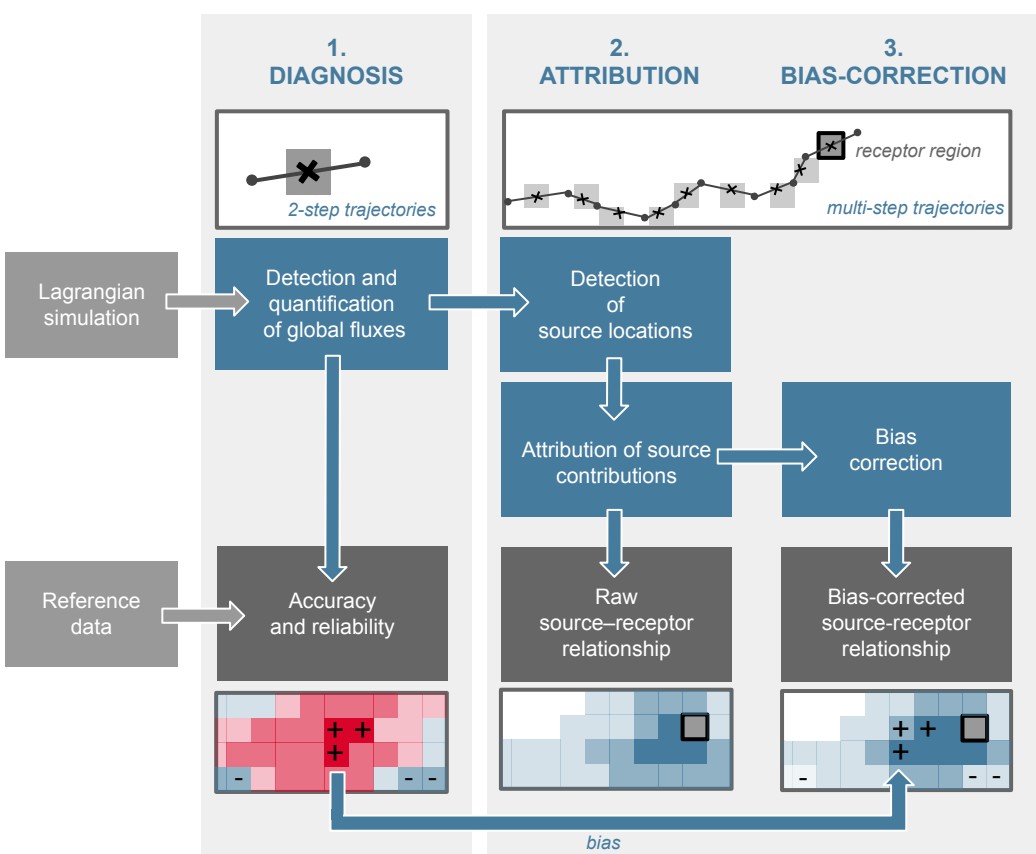

**Figure 1.** Workflow for the process-based evaluation of Lagrangian trajectories to establish source–receptor relationships. In a first step, the output from a Lagrangian model is used to diagnose processes and quantify fluxes, which enables a validation of the accuracy and reliability of the methodology. In a second step, the fluxes along with individual (backward or forward) trajectories are evaluated and source contributions are evaluated/adjusted to enable a mass- and energy-conserving attribution of sources. This step also entails the attribution of sources to a receptor/sink, which is further aggregated to predefined space and time scales (e.g., daily values on a grid) and yields a first estimate of the quantitative source–receptor relationship. In a third step, these estimates are bias corrected using data from the first step, and the resulting dataset is used for analysis. The images at the bottom show the bias of the Lagrangian simulation (red colors indicating an overestimation of the local flux, blue colors indicating an underestimation of the local flux) as estimated in the diagnosis step (left), the biased source regions as estimated in the attribution step (middle; the grey cell indicates the receptor region), and the bias-corrected source

regions after the bias correction step (right). The color intensity indicates the magnitude of the bias and the source region contribution. Note that '+' ('−') signs indicate substantially lower (higher) values in the Lagrangian simulation for easier reference across all steps.

## 2.2 Diagnosis

Using a regional or global simulation, in which air parcels are distributed homogenously in space and time, allows us to diagnose fluxes ($P$, $E$ and $H$ defined in Sect. 2.2.1) for the entire domain. In this diagnosis step, all two-step differences (irrespective of their start and end points) are evaluated and aggregated to a reference flux $R_{LM}$ from the Lagrangian Model (LM), that is not conditioned on a specific sink or receptor region. We highlight that this diagnosis of fluxes is only meaningful if parcels are homogeneously distributed in space and time and not released from individual source- or sink/receptor regions. Moreover, a source bias correction (see Sect. 2.4) relies on an accurate detection of fluxes over the full domain. In this study, all fluxes are aggregated to a global 1°x1°grid — but any other spatial unit (such as watersheds, countries, etc.) could be used.

To characterize the physical processes influencing the air parcels, changes in air parcel properties, such as changes in specific humidity ($q$, kg kg⁻¹), temperature ($T$, K), potential temperature ($\theta$, K), and density ($\rho$, kg m⁻³), are calculated and traced in space and time. A property change (e.g., $\Delta q$) between two timesteps ($t_0$ and $t_{-1}$) is allocated to the midpoint between the corresponding locations ($\vec{x}(t_0)$ and $\vec{x}(t_{-1})$), so that the property change can be calculated as (e.g.) $\Delta q(t_0 ; t_{-1}) = q(\vec{x}(t_0)) - q(\vec{x}(t_{-1}))$. Hereafter, for simplicity, the time step notation when referring to changes in a given parcel property is omitted, thus we refer to, e.g., $\Delta q_0$ instead of $\Delta q(t_0 ; t_{-1})$. Similarly, $q_0$ refers to the specific humidity at time step $t_0$.

### 2.2.1 Detection criteria

#### 2.2.1.1 Precipitation

The atmospheric moisture balance is utilized to detect processes, such as $E$ and $P$ from the change in specific humidity in air parcels (Stohl and James, 2004),

$$e - p = m * \Delta q \tag{1}$$

with ($e - p$) (kg) as the net moisture flux and $e$ (kg) and $p$ (kg) indicating $E$ and $P$ at the parcel level, respectively, $m$ (kg) being the parcel's mass, and $\Delta q$ (kg kg⁻¹) being the change in specific humidity. To select parcels that contribute to a precipitation event, criteria analogous to the convection scheme after Emanuel (1991) can be employed, as demonstrated by Sodemann et al. (2008): if a parcel experiences a net loss of specific humidity between two timesteps ($\Delta q < 0$) and the (mean) relative humidity ($\overline{RH}$) exceeds 80%, the air parcel is assumed to have contributed $\Delta q$ to the precipitation event. Total $P$ (mm) over an area $A$ (m²) can then be quantified as

$$P = \frac{1}{A}\sum_{i=1}^n m * \Delta q_i \ (\Delta q_i < 0 \text{ g kg}^{-1} \ \wedge \overline{RH_i} > 80\%) \tag{2}$$

aggregating over $n$ parcels that fulfill the criteria.

#### 2.2.1.2 Surface fluxes

Analogous process-based criteria for the detection of $E$ and surface sensible heat fluxes ($H$) from trajectories of $q$ and $\theta$, respectively, have been constructed. In Sodemann et al. (2008) — hereafter referred to as SOD08 — all humidity increments larger than 0.2 g kg$^{-1}$ for a 6h timestep of parcels residing in the vicinity of the ABL are aggregated to $E$ as

$$E_{SOD08} = \frac{1}{A} * \sum_{i=1}^{n} m * \Delta q_i (\Delta q_i > 0.2 \text{ g kg}^{-1} \wedge \overline{z_i} < f_z * \overline{h_{ABL}}) \tag{3}$$

with $\overline{z}$ (m) being the mean parcel height, $\overline{h_{ABL}}$ (m) the mean ABL height between the two time steps and parcel locations, and
215 $f_z$ (-), a vicinity factor originally set to $f_z$=1.5. Recently, Fremme and Sodemann (2019) — hereafter FAS19 — and Sodemann (2020) lowered the $\Delta q$ threshold to 0.1 g kg$^{-1}$ in a 6-hour time interval. Moreover, they allowed parcels outside the ABL to contribute to the $E$ calculation, arguing that parcels above the ABL are indirectly affected by surface evaporation through moist convection and mixing:

$$E_{FAS19} = \frac{1}{A} * \sum_{i=1}^{n} m * \Delta q_i (\Delta q_i > 0.1 \text{ g kg}^{-1}). \tag{4}$$

In both approaches (Eq. 3–4), a minimum humidity change is prescribed as a means to filter for noise. Note that the threshold of 0.1 g kg$^{-1}$ was determined in a calibration procedure specific for the Yangtze Valley (Fremme and Sodemann, 2019) — but that it is applied without calibration here.

To detect $H$ instead of $E$, Schumacher et al. (2020) — hereafter SCH20 — followed a similar rationale, exploiting changes in
$\theta$ as

$$H_{SCH20} = \frac{1}{A} * \sum_{i=1}^{n} m * c_p * \Delta \theta_i \left( \Delta \theta_i > 1 \text{ K} \wedge \left[ z_i(t_0) < h_{i,ABL}^{max} \vee z_i(t_{-1}) < h_{i,ABL}^{max} \right] \right) \tag{5}$$

with $c_p$ (J kg$^{-1}$ K$^{-1}$) being the specific heat of dry air. Here, $H$ was detected if an air parcel was warmed by more than 1 K in 6 hours and was within the maximum ABL height at one of the time steps. Schumacher et al. (2019) — hereafter SCH19 — additionally constrained the detection of $H$ on the change in specific humidity, i.e.

$$H_{SCH19} = \frac{1}{A} * \sum_{i=1}^{n} m * c_p * \Delta \theta_i \left( \Delta \theta_i > 1 \text{ K} \wedge \frac{|\Delta q_i|}{q_{i,-1}} < 0.1 \wedge \left[ z_i(t_0) < h_{i,ABL}^{max} \wedge z_i(t_{-1}) < h_{i,ABL}^{max} \right] \right). \tag{6}$$

Thus, an air parcel must fulfill three criteria to be used in the computation of $H$. First, the potential temperature of the air parcel must increase by more than 1 K in a 6-hour time interval. Second, the absolute change in specific humidity must be comparably small to rule out that the warming is caused by latent heat release or by mixing with free tropospheric air, i.e., specific humidity is expected to vary by less than 10%. And last, the air parcel must reside within the maximum ABL height at both time steps.
Note that this detection of heat based on potential temperature is analogous to the detection of heat based on dry static energy by Schumacher et al. (2019).

Here, we aim to compare the accuracy of the above-mentioned detection criteria for $E$ and $H$, and their impact on the source–receptor relationships. In addition, we introduce a novel criterion for the detection of surface fluxes, that takes temperature

changes for the detection of $E$ and moisture changes for the detection of $H$ into account. While the detection of $H$ from Schumacher et al. (2019) already conditions on specific humidity changes, changes in $q$ are independent from the prevalent temperature of the air parcel. Thus, these criteria do not consider the high saturation point of warm air, impeding their use for global applications or seasonal climates. To overcome these restrictions, we condition the detection of both $E$ and $H$ on changes in relative humidity ($RH$) as a function of temperature and specific humidity:

$$E_{RH-20} = \frac{1}{A} * \sum_{i=1}^{n} m * \Delta q_i (\Delta q_i > 0 \text{ g kg}^{-1} \wedge |\Delta \text{RH}_i| < 20\% \wedge [z_i(t_0) < h_{i,ABL}^{max} \vee z_i(t_{-1}) < h_{i,ABL}^{max}]) \tag{7}$$

and

$$H_{RH-10} = \frac{1}{A} * \sum_{i=1}^{n} m * c_P * \Delta \theta_i (\Delta \theta_i > 0 \text{ K} \wedge |\Delta \text{RH}_i| < 10\% \wedge [z_i(t_0) < h_{i,ABL}^{max} \vee z_i(t_{-1}) < h_{i,ABL}^{max}]). \tag{8}$$

Therefore, for the detection of $E$ and $H$, the absolute change in $RH$ is required to be lower than 20% and 10% (respectively) between two consecutive time steps. Moreover, one parcel height must be within the maximum ABL height of both time steps. This reasoning follows observation-based results (e.g., Ek and Mahrt, 1994), that indicate that large $RH$ changes are typically associated with ABL growth and warming, and entrainment of dry air from the free troposphere. Moreover, at 1-hourly intervals, $RH$ changes associated with $E$ remain restricted to less than 15%. Here, we assume that similar thresholds are applicable on 6-hourly intervals, but we allow moisture to vary more depending on temperature ($|\Delta RH| < 20\%$ for the detection of $E$) compared to the maximum temperature change depending on moisture ($|\Delta RH| < 10\%$ for the detection of $H$). Note that the reasoning behind this $RH$ criterion is contrary to the above-mentioned filtering based on minimum thresholds: instead of filtering out noise, we argue that a maximum threshold must be applied as a means to filter out changes associated with the aforementioned mixing processes. Thus, the absolute $\Delta \theta_i$ and $\Delta q_i$ thresholds for the detection of $H$ and $E$ are lowered to 0 K and 0 g kg$^{-1}$, respectively, and the relative humidity change is required to be considerably low ($|\Delta RH| < 10\%$ or $< 20\%$; cf. Fig. S1).

To evaluate the impact of these detection criteria on the source–receptor relationships, we compare different combinations of criteria as listed in Tab. 1. An additional experiment — referred to as ALL–ABL — is introduced to evaluate the impact of the $RH$ criterion and the use of minimum thresholds in a disjunct manner. It is highlighted that the four criteria here are complementary, i.e., ALL-ABL lies in between SOD08 and RH-20: compared to SOD08, ALL-ABL does not consider a minimum $\Delta q$ threshold and hence indicates if filtering for a minimum threshold improves the detection of $E$. Compared to RH-20, the ALL-ABL criterion does not consider a maximum $|\Delta RH|$ threshold and hence allows us to infer the suitability of this temperature-dependent threshold. The FAS19 criterion in turn also accounts for uptakes above the ABL. Note that all thresholds are employed globally, despite the fact that some thresholds were calibrated for specific study regions (SOD08, FAS19). In this context, it is noted that the ABL criterion is unified to facilitate the comparison: except for FAS19, the detection of surface fluxes is restricted to the ABL. Aiming to move towards a process-based detection, we require one of the parcel positions to be within the maximum ABL ($h_{ABL}^{max}$). We point out that considerable discretion remains when it comes to the selection of parcels within the ABL: one or both occurrences could be required to be within the (maximum) ABL height. The

impact of the applied ABL height criterion is distinctly small for 6h time steps from the ERA-Interim reanalysis (see Fig. S4–S5).

**Table 1** Overview of detection criteria for evaporation.

| Experiment name | $\Delta q$ | $\Delta \theta$ | $\|\Delta RH\|$ | ABL height | Reference |
|---|---|---|---|---|---|
| FAS19 | > 0.1 g kg$^{-1}$ | - | - | no | Fremme and Sodemann (2019) |
| SOD08 | > 0.2 g kg$^{-1}$ | - | - | yes | Sodemann et al. (2008) |
| RH–20 | > 0 g kg$^{-1}$ | - | < 20% | yes | - |
| ALL–ABL | > 0 g kg$^{-1}$ | - | - | yes | - |

**Table 2** Overview of detection criteria for sensible heat.

| Experiment name | $\Delta \theta$ | $\|\Delta q\|$ | $\|\Delta RH\|$ | ABL height | Reference |
|---|---|---|---|---|---|
| SCH19 | > 1 K | < 10% | - | yes | Schumacher et al. (2019) |
| SCH20 | > 1 K | - | - | yes | Schumacher et al. (2020) |
| RH–10 | > 0 K | - | < 10% | yes | - |
| ALL–ABL | > 0 K | - | - | yes | - |

### 2.2.2 Multi-objective validation

Using a multi-objective validation approach, we evaluate several cost functions and consider both accuracy and reliability to assess the criteria for the detection of fluxes from air parcel trajectories. Accuracy is determined based on the bias, calculated as

$$s_{bias} = R_{LM} - R_{obs}, \tag{9}$$

where $R_{LM}$ represents the generic flux $R$ as determined with the Lagrangian model — i.e., any of the fluxes as defined from two-step trajectories in Sect. 2.2.1 ($P$, $E$, $H$) — and $R_{obs}$ represents the observation from a reference data set; both averaged over a pre-defined time period. Here, the observation can either come from the same dataset used as atmospheric forcing (e.g., ERA-Interim in this case) or any other dataset. Beyond the bias, we aim to evaluate the reliability of the detection criteria at various time scales. Especially the detection of $E$ and $P$ from changes in $q$ is conditional (either $E$ or $P$ can be detected from $\Delta q$ at a specific time step and for a specific parcel), and hence calls for a validation that can incorporate the probability of detection ($s_{pod}$) and the probability of false detection ($s_{pofd}$). We combine these two probabilities to compute the Peirce's skill score as

$$s_{PSS} = s_{pod} + (s_{pofd} - 1) \tag{10}$$

where

$$s_{pod} = \frac{c_h}{c_h + c_m} \tag{11}$$

and

$$s_{pofd} = \frac{c_{fa}}{c_{fa} + c_{cn}} \tag{12}$$

as calculated with a contingency table, with $c_h$ being the number of hits, $c_m$ the number of misses, $c_{fa}$ the number of false alarms, and $c_{cn}$ the number of correct negatives (see, e.g., Jolliffe and Stephenson, 2012 for details). Here, the contingency table is calculated using a minimum daily threshold of 1 W m$^{-2}$ for $H$ and 0.1 mm d$^{-1}$ for $E$ and $P$. The $s_{PSS}$ indicates how well the diagnosis criteria separate events from "no events", and yields a score ranging from –1 to 1, with 0 indicating no skill and 1 indicating a perfect separation. The multi-objective validation considers both $s_{bias}$ and $s_{PSS}$, and thus quantifies the accuracy and reliability inherent in the methodology. We note, however, that an *a posteriori* bias-correction may correct not only the bias ($s_{bias}$) but also the false detection events ($s_{pofd}$). Hence, we argue that, in this step, it is desired to achieve a high probability of detection over a low probability of false detection $s_{pofd}$ and low biases.

## 2.3 Attribution

Source–receptor relationships from Lagrangian models are typically established using backward trajectories, e.g., from a precipitation event. In this case, all parcels fulfilling the precipitation criteria from Eq. (2) at a specific date and location may be traced backward for (e.g.) 15 days as a proxy for the globally averaged maximum lifetime of water vapor in the atmosphere. Analogously, all air parcels over a receptor region may be traced backward to estimate the origins of their heat and moisture, and to approximate heat and moisture advection (Schumacher et al., 2020). To establish a quantitative source–receptor relationship, property changes along parcel trajectories must be accounted for, e.g., rain *en route* has to be discounted from previously detected moisture uptakes (Sodemann et al., 2008), or nighttime cooling has to be considered for the advection of heat (Schumacher et al., 2019). Due to the consideration of changes in $q$ or $\theta$ between the source and the sink/receptor, the analysis conserves mass- and energy along individual trajectories. Source contributions can hence be aggregated over the source regions of interest — e.g., grid cells, basins, countries — and provide a quantitative perspective of the source–receptor relationship. For the analysis in this study, the moisture and heat source regions for three cities (Denver, Beijing, Windhoek) are estimated using a 3°x3° receptor/sink region around each city, and source contributions from 15-day backward trajectories are aggregated to a 1°x1° grid. In the following, we present and explore two different approaches to perform this attribution.

### 2.3.1 Linear discounting of *en route* losses and linear attribution

Sodemann et al. (2008) proposed a linear algorithm that discounts moisture losses between a source location and the sink region to ensure mass conservation along trajectories, which was then adopted by Schumacher et al. (2019) to discount heat losses (i.e., cooling) — we refer to this procedure as 'linear discounting'. Similarly, the same linearity assumption is made to attribute the precipitation event to the identified source locations. This linear approach follows the assumption that an air parcel is perfectly-mixed at all times. For moisture, this assumption implies that the age composition of specific humidity content in

an air parcel remains the same before and after the precipitation event (or any other moisture loss); and that all sources contribute to a precipitation event with their exact relative contribution to the pre-precipitation moisture. An example is given in Fig. 2a–c: if a specific source time step/location contributes 20% of the specific humidity content before the precipitation event, it also supplies exactly 20% of moisture to the precipitation event (hereafter, we refer to this estimation of source contributions as 'linear attribution') — and also after the precipitation event, this specific source constitutes 20% of the moisture left in the air parcel. Thus, moisture losses *en route* are *discounted* from the source fluxes, while the analyzed precipitation event is *attributed* to the identified source locations (but the procedure remains the same). Note that the approach presented here is identical to the one by Sodemann et al. (2008) but follows a different convention, and it is further generalized to be applicable to additional variables, such as the potential temperature (or any other variable that may be tracked).

In the discounting step, the raw source region contribution to the sink region at time step $t$ ($\Delta\Psi_t$, with $\Psi$ being $q$ or $\theta$ for the detection of $E$ or $H$, respectively) is corrected by considering all losses (in $q$ or $\theta$) along the trajectory, i.e., between the time step of the increase associated with a source time step ($t$), and the timestep of arrival in the receptor region ($t_0$). This discounting follows the assumption that the $q$ (or $\theta$) that is lost originated in previous increases *en route*, and in the exact proportional manner to the magnitude of these uptakes to the specific humidity content of that air parcel. Therefore, a loss at time step $j$ ($\Delta\Psi_j < 0$) between a source location and time step $t$ and the receptor region ($t < j < t_0$, with $t$ and $j$ being smaller than 0 due to backward tracking) is accounted for in all previous source contributions ($t < j$). The corresponding source fluxes ($\Delta\Psi_t > 0$) are then *discounted* via

$$\Delta\Psi_t^* = \Delta\Psi_t \; - \sum_{j>t} \frac{\Delta\Psi_t^*}{\Psi_{j-1}} * |\Delta\Psi_j| * \mathcal{H}_{(\Delta\Psi_j < 0)} \tag{13}$$

and thus reduced assuming that they contributed linearly to the absolute quantity prior to the loss or cooling ($\Delta\Psi_t/\Psi_{j-1}$). Here, $\Delta\Psi_t^*$ refers to the discounted source flux, and $\mathcal{H}_{\Delta\Psi_j < 0}$ is the Heaviside function that returns 1 if a quantity decrease is encountered ($\Delta\Psi_j < 0$), thereby successively discounting all uptakes with losses between the source region and the receptor region (i.e., $t < j < t_{-1}$ for precipitation and $t < j < t_0$ for any advection quantity, such as heat advection). As every time step $t$ also corresponds to an uptake location $\vec{x}(t)$, time and location are interchangeable here (but are always considered to be *forward*, i.e., from the source to the receptor). Subsequently, the relative source region contribution $f_t$ to a sink or receptor quantity can be calculated as

$$f_{t,linear} = \frac{\Delta\Psi_t^*}{\Psi_{-1}} \text{ or } f_{t,linear} = \frac{\Delta\Psi_t^*}{\Psi_0}, \tag{14}$$

respectively. This coefficient $f$ describes the fraction of moisture prior to the precipitation event and the fraction of sensible heated energy in the parcel originating from time step (and location) $t$ (see example in Fig. 2c).

The contribution of surface evaporation (at timestep $t$) to precipitation detected for the same parcel in the sink region at a later timestep $t$, is then calculated as the relative source region contribution $f_t$ multiplied with the moisture decrease over the sink region $\Delta q_0$ (or precipitation amount), i.e.,

$$e_{t,linear} = f_{t,linear} * \Delta q_0 \; (\Delta q_0 < 0 \text{ g kg}^{-1} \wedge \overline{RH_0} > 80\%). \tag{15}$$

Aggregated over all time steps $n_t$ of a trajectory then sums up to the precipitation amount of the parcel (if the full precipitation event could be attributed; see Supplementary Material Sect. 4). Note, however, that heat advection is not constrained by a receptor quantity. Here, only discounted values of surface-induced potential temperature changes are aggregated along a trajectory, i.e.

$$h_{\text{adv}} = \frac{1}{A} \sum_{t=-n_t}^{0} m * c_P * \Delta\theta_t^*. \tag{16}$$

### 2.3.2 Random attribution for moisture

To evaluate the impact of the assumption that air parcels are perfectly mixed and hence source locations always contribute according to their share of moisture in the air parcel prior to the $P$ event, we introduce a random attribution methodology designed for precipitation and moisture advection only (i.e., not applicable to heat advection). This random attribution determines physically reasonable limits of source region contributions along a trajectory to ensure mass conservation in the evaluation of source–sink relationships. Through the evaluation of the minimum specific humidity content along a trajectory (rather than the specific humidity changes as employed in the linear discounting and attribution), this approach indirectly considers moisture losses between source and sink/receptor and hence does not require any additional discounting of losses *en route*. We thus refer to this approach as 'random attribution' only, but emphasize that mass conservation along the trajectory is also fulfilled.

The maximum attributable moisture at an uptake location along a trajectory is bound by the uptake itself (analogous to the linear attribution following Sodemann et al., 2008), as well as the minimum moisture content between the uptake location and the sink region. In contrast to linear discounting, which reduces the source contribution by (biased) estimates of precipitation and other moisture decreases, we make use of the (unbiased) humidity content to constrain the contributions and indirectly account for moisture losses *en route*. In addition, the attribution depends on other uptake locations (analogously determined as for the linear discounting and linear attribution, i.e., source locations fulfilling the criteria for $E$ as defined above) and thus incorporates spatiotemporal dependencies. For a trajectory of length $n_t$, the random attribution of a sink (precipitation) or receptor (moisture content) quantity is performed in three steps:

1. The maximum contribution for each predefined identified source time step/location $t$ is calculated as follows:

$$e_{t,max} = min\left(\left(q_{t,...,-1}\right) - \sum_{i=-n_t}^{t} e_{i,random}, \Delta q_t - e_{t,random}\right) \tag{17}$$

2.  The source location along a trajectory is determined by drawing a random number $t$ between $-n_t$, …$-1$ potential source time steps and locations; where the probability of each source time step/location is determined by its maximum potential contribution relative to the other time steps/locations (i.e., $e_{t,max}/\sum_{-n_t}^{-1} e_{t,max}$).

3.  The contribution $e_{j,random}$ is drawn from a uniform distribution ranging from 0 to $min\,(e_{t,max}, \Delta q_0/n_{min})$ for precipitation; and is attributed to the source time step and location $t$.

These steps are repeated until either the full sink/receptor quantity is attributed, or the physical maximum attributable fraction of the considered trajectory length is reached. To reduce the random factor and increase the reliability of random sampling, the sink or receptor quantity ($\Delta q_0$ or $q_0$, respectively) may be divided by a factor $n_{min}$ to assure that at least $n_{min}$ iterations are performed (and not the entire $P$ event is attributed to one identified source location), thus increasing the likelihood of a widespread attribution along the trajectory. Unless mentioned otherwise, $n_{min}$ is set to the number of identified source locations, but does not go below $n_{min} = 10$ iterations in this study. It is noted here that we refer to this attribution algorithm as 'random' as it involves the selection of random source locations and random contributions in an iterative procedure; however, source locations and maximum magnitudes are still determined physically. In addition, since all precipitation events are evaluated independently and the moisture from a single evaporation event can rain out in multiple subsequent precipitation events, aggregated source contributions may exceed diagnosed evaporation occasionally if compared at individual time steps or on a daily basis.

### 2.3.3    Idealized examples of attributing precipitation events

Figure 2 shows two idealized examples of the estimation of source contributions for a precipitation event using the linear discounting/attribution and the random attribution. The first example in Fig. 2 a–f shows a precipitation event at time step 0 that is traced backward in time for six time steps ($n_t=6$). An air parcel associated with the precipitation gained moisture during all previous five time intervals (+2 g kg$^{-1}$ during each time interval; grey bars in Fig. 2a). The parcel loses 50% (–5 g kg$^{-1}$) of its specific humidity content during the precipitation event. Using the linear attribution, each source time step/location is assumed to contribute exactly 1 g kg$^{-1}$ of moisture to the precipitation event (Fig. 2b); i.e., 20% of the precipitation is one time step old, 20% of the precipitation is two time steps old, and so on (Fig. 2c) — thus representing perfectly-mixed conditions. On average, the random attribution also reflects well-mixed conditions: averaged over 1000 realization of the random attribution for this trajectory, each source time step/location contributes ~20% of moisture to the precipitation event (red bars in Fig. 2f). However, individual realizations may divert from this condition: in extreme conditions, a single source time step contribution may approach 2 g kg$^{-1}$ (grey bars in Fig. 2e), however, 50% lie between ~0.65 g kg$^{-1}$ and 1.33 g kg$^{-1}$ for all source time steps/locations (error bars in Fig. 2e); thus allowing deviations from the perfectly-mixed assumption in the linear attribution algorithm but preserving an average well-mixed condition (see also other examples in Fig. S3).

The second example in Fig. 2 g–l shows a similar trajectory but includes a precipitation event *en route*. Analogous to the first

example, the air parcel in this trajectory gains + 2 g kg$^{-1}$ of moisture at each source time step/location, but it loses 3 g kg$^{-1}$ at time step $t=-3$, before 50% of the specific humidity content is lost during the precipitation event at $t=0$ (i.e., –2.5 g kg$^{-1}$). To ensure mass conservation, the intermediate precipitation event must be accounted for. Assuming perfectly-mixed conditions, the source time steps $t=-5$ and $t=-4$ contribute each 50% of moisture to that intermediate precipitation event — and thus via the linear discounting, their contributions are reduced to 0.5 g kg$^{-1}$, i.e., they can only contribute 10% of moisture to the specific humidity content of the air parcel before the tracked precipitation event at $t=0$. In the attribution step, these two earlier source locations are then assumed to contribute each 10% of moisture to the final precipitation event (i.e., 0.25 g kg$^{-1}$, see Fig. 2i+h). Thus, in perfectly-mixed conditions, the two source locations associated with time steps $t=-2$ and $t=-1$ contribute each 40% (1 g kg$^{-1}$) of moisture to the precipitation event and the two source locations at time steps $t=-5$ and $t=-4$ contribute each 10% (Fig. 2i+h). Again, the average contributions as estimated with the random attribution approach well-mixed conditions (Fig. 2k–l). However, instead of discounting the source contributions with losses *en route*, the minimum specific humidity content of the air parcel is used to ensure mass conservation: together, the source locations associated with time steps $t=-5$ and $t=-4$ can maximally contribute 1 g kg$^{-1}$, i.e., the specific humidity content at $t=-3$. The maximum contribution of these sources is thus 1 g kg$^{-1}$ (assuming the other location contributes no moisture at all) — and thus considerably lower than the maximum contribution of the sources associated with time steps $t=-2$ and $t=-1$. On average, out of 1000 realizations, the oldest two source locations are estimated to contribute ~0.38 g kg$^{-1}$ and the newer source locations contribute each 0.86 g kg$^{-1}$ (red bars in Fig. 2k), but again allowing for deviations (error bars in Fig. 2k–l). In this case, ~30% of the final precipitation event is attributed to source time steps/locations $t=-5$ and $t=-4$ (~15% each), and ~70% is attributed to the most recent source time steps $t=-2$ and $t=-1$ (~35% each, see Fig. 2l). The average contributions are thus slightly shifted towards older sources if compared to perfectly-mixed contributions.

The two examples here illustrate that the random attribution approaches well-mixed conditions: in the first example, the random attribution allows deviations from perfectly-mixed conditions, but average contributions resemble perfectly-mixed conditions. In the second example, average contributions as estimated with the random attribution slightly divert from the perfectly-mixed assumption. It is noted, however, that the random attribution only approaches perfectly-mixed conditions if a sufficiently large number of parcels is tracked and evaluated; larger deviations from well-mixed conditions may be expected for smaller sample sizes (not shown). Thus, for individual days and/or events, the random attribution may also illustrate deviations from well-mixed conditions.

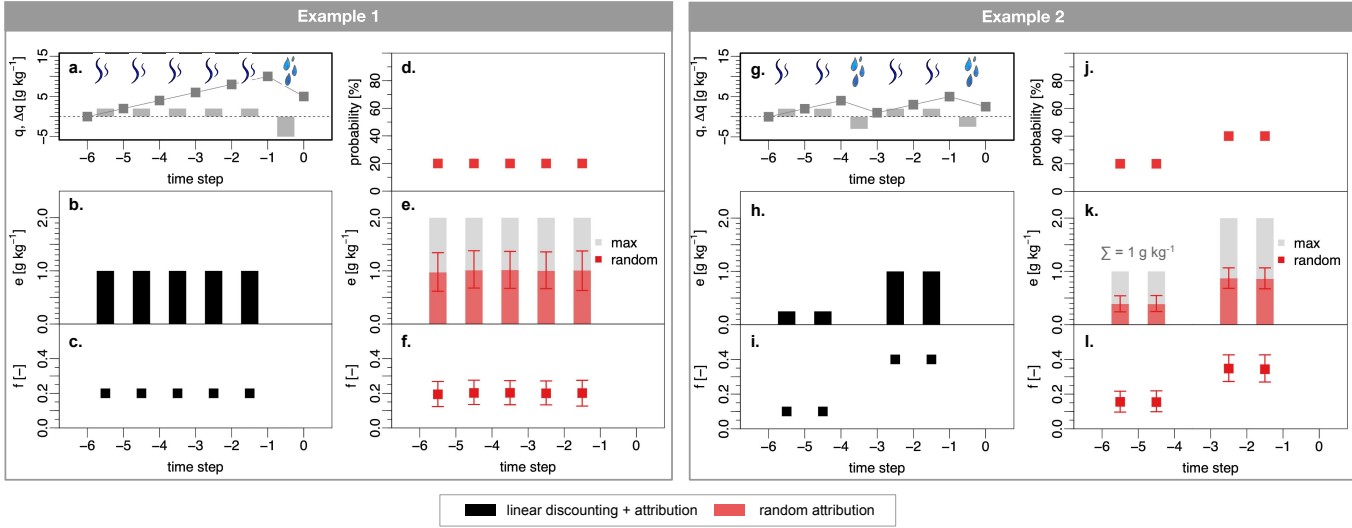

**Figure 2.** Illustration of backward tracking precipitation from two idealized air parcel trajectories: **(a+g)** timeseries of specific humidity (dark grey lines and points) and specific humidity changes (grey bars) of the air parcels that lose moisture through precipitation at time step 0. The air parcels are each tracked backward in time for six time steps, during which they gained moisture through surface evaporation. In both examples **(a+g)**, the air parcel gains +2 g kg$^{-1}$ of moisture at each source and loses 50% of its specific humidity content during the (final) precipitation event. In the second example **(g)**, an additional rain event takes place *en route*, at time step –3, reducing the specific humidity to 1 g kg$^{-1}$. **(b+h)** Absolute contribution of each source to the precipitation event in step 0 as calculated with the linear attribution, assuming perfectly-mixed conditions. In **h**, the absolute contributions of time steps –6 and –5 are lower because of the rain *en route* event. **(c+i)** Relative contributions as calculated with the linear discounting and attribution. **(d+j)** Probability of selection for each source time step/location as used in the random attribution — that is based on the specific humidity content of the air parcel instead of specific humidity changes as employed for the linear discounting and attribution. **(e+k)** Maximum contribution for the random attribution (grey bars; assuming $n_{min}$=1 for illustrational purposes) and absolute contribution (red bars) of each source to the precipitation event as calculated with the random attribution; averaged over 1000 random attribution realizations. Error bars illustrate the interquartile range of these realizations (i.e., 25–50%). **(f+l)** Corresponding average relative contributions and deviations as calculated with the random attribution.

### 2.3.4 Aggregation to establish (biased) source-receptor relationships

Analogously to the diagnosis (Sect. 2.2), source region contributions are aggregated over source region $x$ with area $A$ and $n$ trajectories — but also the backward time dimension $n_t$. Therefore, the contributions $e_{i,t}$ from $i=1,...,n$ parcels that arrive in the receptor region and resided over the source region $x$ at the same time are summed up to estimate the overall source region contribution from the Lagrangian model. The source contribution of $x$ (e.g., an individual grid cell) to $P$ over the sink region is calculated as

$$S_{LM}^x = \frac{1}{A} \sum_{i=1}^n \sum_{t=-n_t}^{-1} m * e_{i,t}, \tag{18}$$

where $e_{i,t}$ can be the contributions from linear or random attribution, respectively. Aggregated over all source regions $x$, $S_{LM} = \sum_x S_{LM}^x$ sums up to the total $P$ in the sink region that is attributed to the identified source locations. It is noted here that — depending on the setting and the attribution methodology — not the entire precipitation volume may be attributed to source locations along individual trajectories (i.e., $R_{LM} > \sum_x S_{LM}^x$, see Supplementary Material for further explanations). However, to facilitate the comparison between different approaches, we always employ upscaled contributions in this study, i.e., we attribute the full precipitation volume to the identified source locations along a trajectory (see Supplementary Material).

The contribution of surface sensible heating over the source region $x$ to heat advection towards the receptor region is summed up analogously, i.e.

$$S_{LM}^x = \frac{1}{A}\sum_{i=1}^{n}\sum_{t=-n_t}^{-1} m * c_P * \Delta\theta_{i,t}^*. \tag{19}$$

Aggregated over all source regions $x$, $S_{LM} = \sum_x S_{LM}^x$ represents *heat advection* as defined in this study.

## 2.4 Bias correction

While $s_{PSS}$ offers the possibility to tune the detection criteria, the magnitude of the diagnosed fluxes from Lagrangian trajectories may be biased for multiple reasons. First, $E$ and $P$ are both detected from $\Delta q$ in an air parcel, which also reflects phase changes such as condensation and vaporization due to, e.g., the formation or dissolution of clouds, but also mixing processes (Sodemann, 2020). Second, even if parcels are filtered for processes so that $E$ and $P$ may co-exist over a specific region in which multiple parcels are present, the $\Delta q$ per parcel always reflects the net of $E$ and $P$, which may lead to an underestimation of both fluxes. Finally, the uncertainty associated with the applied convection parameterization, but also the numerical noise and interpolation errors as a result of the setup of the Lagrangian model, lead to biased estimates of all fluxes. Yet, as long as $s_{PSS}$ indicates a high reliability, these fluxes can be bias-corrected. While the bias correction of precipitation as a sink flux is commonly performed, studies also indicate an overestimation of surface evaporation from air parcel trajectories (Sorí et al., 2017), which is typically not corrected for. Here, we describe three possibilities for bias-correcting source–receptor relationships: the correction of source fluxes, the correction of sink fluxes, and the correction of both fluxes. While the advection of heat is limited to a bias correction of the source via the detection of sensible heat fluxes, source–sink relationships from $E$ to $P$ offer the possibility to compare all methods.

### 2.4.1 Bias correction of receptor variables

As commonly performed for source region contributions of precipitation, the sink or receptor variable (precipitation; or the integrated water vapor as a receptor quantity for moisture) can be bias corrected using observations. The corresponding contributions of a source region $S_{LM}^x$ are thus scaled using a bias-correction factor $\frac{R_{obs}}{S_{LM}}$ as

$$S_{receptor-corrected}^{x} = S_{LM}^{x} * \frac{R_{obs}}{S_{LM}} \tag{20}$$

with $S_{LM}$ being the sum over all (area-weighted) source regions $x$, i.e.

$$S_{LM} = \sum_{x} S_{LM}^{x} \tag{21}$$

and $R_{obs}$ being the reference quantity. If 100% of the sink/receptor quantity is attributed, then $R_{LM} = S_{LM} = \sum_{x}^{n} S_{LM}^{x}$. This bias-correction methodology assumes that the Lagrangian model evaluation yields a valid relative source–receptor relationship,

$$\left(\frac{S_{LM}^{x}}{S_{LM}}\right) = \text{const.} \tag{22}$$

i.e., the relative contribution of a source region to the sink/receptor is not changed.

### 2.4.2 Bias correction of source variables

To account for a potential overestimation of surface fluxes ($E$, $H$) in the source regions, these fluxes can be bias corrected as well, using a reference data set $R_{obs}$. Here, source region contributions are corrected as follows

$$S_{source-corrected}^{x} = \frac{S_{LM}^{x}}{R_{LM}} * R_{obs} \tag{23}$$

where $R_{LM}$ is the unconditional flux as detected with the Lagrangian model (evaluating all parcels over the source region $x$). Since this bias-correction method assumes a valid relative source region contribution ($\frac{S_{LM}^{x}}{R_{LM}} = constant$), the relative source–receptor relationship changes due to the correction.

### 2.4.3 Bias correction of source and receptor variables

To remove the bias from both source and sink/receptor quantities, a combined bias-correction can be completed in two steps: (1) a source bias-correction is applied, (2) a sink bias-correction is performed considering the changed quantities arriving in the receptor region:

$$S_{source-and-receptor-corrected}^{x} = S_{source-corrected}^{x} * \frac{\sum_{x} S_{LM}^{x}}{\sum_{x} S_{source-corrected}^{x}} * \frac{R_{obs}}{R_{LM}} \tag{24}$$

which collapses to

$$S_{source-and-receptor-corrected}^{x} = S_{source-corrected}^{x} * \frac{R_{obs}}{\sum_{x} S_{source-corrected}^{x}} \tag{25}$$

if 100% of the sink/receptor quantity is attributed (i.e., $R_{LM} = S_{LM} = \sum_{x} S_{LM}^{x}$). Analogous to the bias-correction of source quantities, this methodology assumes a valid relative source region contribution, but modifies the source–receptor relationship. In Sect. 3.3, we demonstrate the impact of these assumptions on the estimated source regions of precipitation, bias-correcting surface evaporation in the source region and precipitation in the sink region.

### 2.5 Lagrangian model

The appraisal framework presented in this paper is applicable to a wide range of trajectory models but was developed primarily for the evaluation of air parcel trajectories in the atmosphere. To demonstrate its applicability, we employ a global simulation

from FLEXPART version 9.01 (Stohl et al., 2005) driven with ERA-Interim reanalysis at 1° resolution (Dee et al., 2011). Six-hourly ERA-Interim reanalysis (00, 06, 12, and 18 UTC) and respective three-hourly forecasts (03, 09, 15, and 21 UTC) were used to calculate the trajectories, yet the analysis is performed using only the six-hourly reanalysis data for consistency. These simulations span the time period 1980–2016 and comprise ~2 million air parcels that are distributed homogeneously and traced in space and time (corresponding to an air parcel mass of $2.54 \cdot 10^{12}$ kg). FLEXPART requires three-dimensional fields of horizontal and vertical wind, temperature, and specific humidity, and two-dimensional fields of surface pressure, cloud cover, 2-m temperature and dew-point temperature, precipitation, sensible and latent heat fluxes, and N/S and W/E surface stress. These fields are used to improve the physical realism of offline Lagrangian simulations and thereby improve the physical consistency of FLEXPART over other offline Lagrangian models that only consider moisture fluxes of $E$ and $P$ along with — frequently only 2D — wind fields. While 3-hourly forcing data is employed, the timesteps for the calculation of trajectories are adapted to Lagrangian timescales to increase the interaction between horizontal and vertical wind components, resulting in a better representation of turbulence (Stohl et al., 2005), i.e., a 900 seconds synchronization and sampling timestep was used, but turbulence was simulated at smaller time steps where necessary (i.e., by setting CTL=2 and IFINE=4 — see Stohl et al., 2005 for details). Sub-grid terrain effect parameterizations are used to increase mixing heights arising from topographic variance at the grid-cell level. The Emanuel (1991) convection scheme is employed to enhance the simulation of convection (Stohl et al., 2005). FLEXPART's model output comprises binary output files with parcel positions (longitude, latitude, and height) and properties (temperature, density, and specific humidity), as well as the surrounding boundary layer height for each air parcel. Outputs are available every 3 hours; however, only 6-hourly analysis time steps are used for the evaluation. Using the parcels' ID, their trajectories can be constructed to enable a process-based analysis.

## 3    Results

For the purpose of this study, all variables ($P$, $E$, $H$) are diagnosed globally on a horizontal 1°x1° grid, and the reliability and accuracy of this diagnosis is evaluated. To illustrate the impact of the selection criteria and errors associated with the detection, source regions of precipitation and heat are determined from 15-day backward trajectories from 1980–2016 for three cities, belonging to different climates, and their surroundings: Beijing (China), Denver (USA) and Windhoek (Namibia) – see Fig. 3. The uncertainty of the established source–receptor relationships is assessed by varying selection criteria and exploring the impact of attribution and bias correction methods. While results are mainly illustrated for the city of Beijing, analogous figures for Denver and Windhoek are available in the Supplementary Material. Unless otherwise noted, a 3°x3° box around each city center is used as a receptor area. Note that we refrain from evaluating the approach from Stohl and James (2004) as it focuses on the qualitative detection of general source and sink regions of moisture from a dynamic meteorology-perspective. Instead, we focus on the process-based evaluation of trajectories that facilitates a quantitative estimation of source regions contributions that is further applicable to heat.

### 3.1 Diagnosis

#### 3.1.1 Reliability and accuracy

To evaluate the reliability of the detection criteria, the diagnosed fluxes are benchmarked against the forcing data, i.e., ERA–Interim. Figure 3 shows global maps of the average bias, the probabilities of detection and false detection, and the corresponding Peirce's skill score for the process-based detection of $P$, $E$ and $H$ using RH criteria (RH–10 for $H$ and RH–20 for $E$) for daily data from 1980—2016. Biases for $P$ and $E$ range from –13.8 to +3.4 mm d$^{-1}$ and from –3.5 to 6.9 mm d$^{-1}$, respectively. As both $E$ and $P$ are disentangled from the net moisture flux (Eq. 1), the detection of $E$ shows a clear dependency

on $P$ (Fig. 3b): both fluxes exhibit the strongest deviations from the driving forcing around the equator and the intertropical convergence zone (ITCZ). Over land, $P$ is mostly underestimated, except for large parts of Siberia and Australia (Fig. 3a). Conversely, $E$ is commonly overestimated over land (Fig. 3b). The bias of $H$ reaches values up to 100 W m$^{-2}$ for single pixels and shows patterns that follow orographic features (Fig. 3c). The probability of detection reaches values up to 100% almost everywhere and for all three variables (Fig. 3d–f), i.e., at least one parcel fulfills the criteria for $P$, $E$ and $H$ each day. However,

$P$ is not reliably detected on a daily scale over mountain chains and arid regions (e.g., Sahel, Middle East, Mongolia; Fig. 3d). Yet, the corresponding probability of false detection is considerably low (<10%) for $P$ over land, and higher values are limited to oceans (Fig. 3g). The probability of false detection for $E$ is comparably high, especially in arid regions, such as the Sahel, and mountainous regions, where $E$ is falsely detected (Fig. 3h). Only in the tropics, $E$ is almost never falsely detected (Fig. 3h). Together, the detection criteria for both $E$ and $P$ show positive skills at the daily scale almost everywhere (Fig. 3j+k).

There is a positive skill at detecting $H$ larger than 1 W m$^{-2}$ almost everywhere (Fig. 3l). The Sahara region exhibits a very low probability of false detection for $H$ (Fig. 3i), whereas all other regions exhibit a large probability of false detection. Greenland and Antarctica show the lowest probabilities of detection along with the low probabilities of false detection for all variables. The skill of detecting any variable over these regions is thus low, indicating that analyses of source–receptor relationships over these regions should be performed with caution. It is noted here, however, that a global maximum threshold of $RH$ change was

applied for both $E$ and $H$ ($\Delta RH < 20\%$ and $\Delta RH < 10\%$, respectively), but that these thresholds may be calibrated individually for each region or even per pixel. Moreover, if a bias-correction of sink and/or source quantities is performed, this corrects for false detections (lowering the corresponding property). Yet, if a process remains undetected, this cannot be corrected for — hence, a high probability of detection may be preferred over a high probability of false detection, irrespective of the biases. In the following, we thus focus on the probability of detection as a measure of skill.

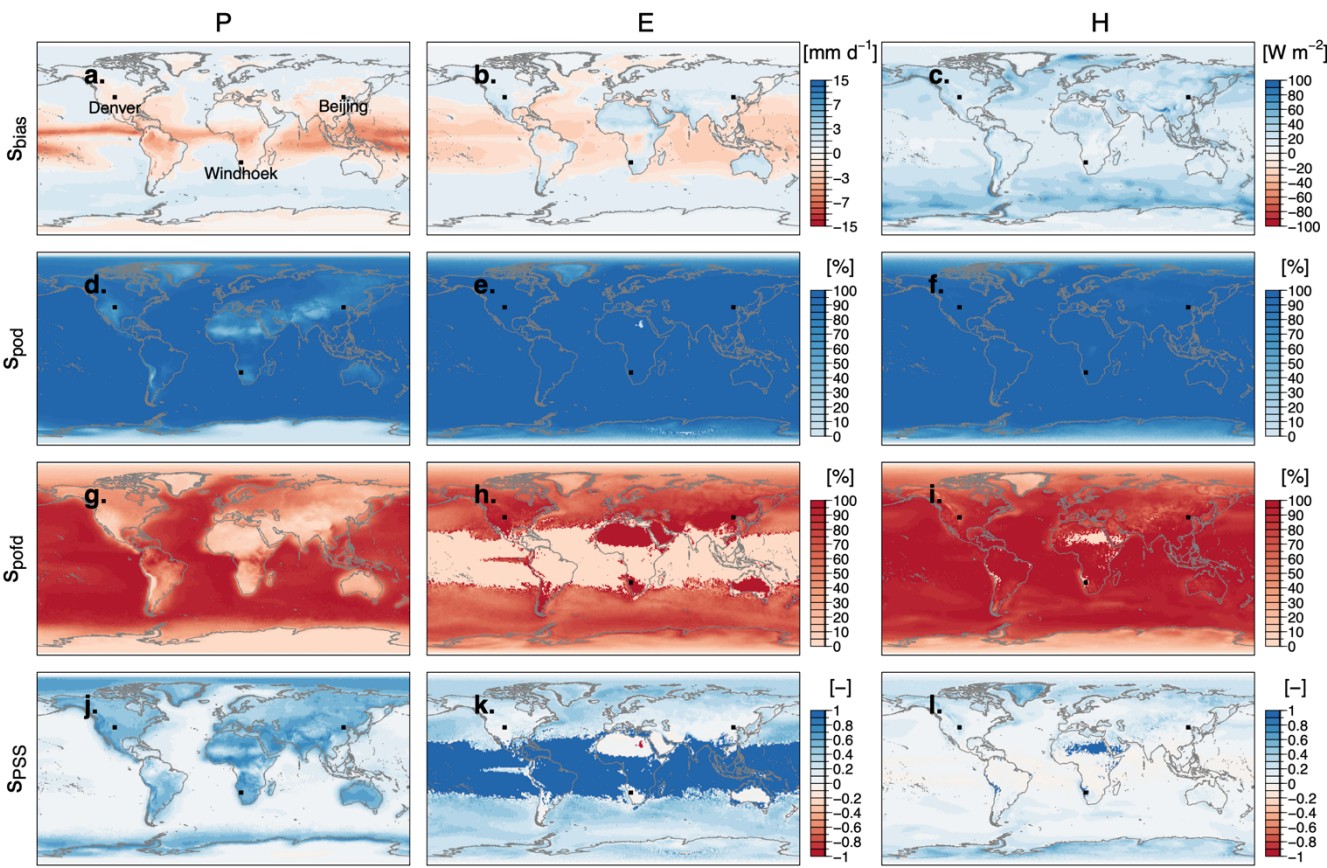

**Figure 3.** Bias ($s_{bias}$), probability of detection ($s_{pod}$), probability of false detection ($s_{pofd}$) and Peirce's skill score ($s_{PSS}$) for daily (**a, d, g, j**) precipitation, (**b, e, h, k**) surface evaporation, and (**c, f, i, l**) sensible heat fluxes for the period 1980–2016. Criteria for precipitation detection follow Sodemann et al. (2008) and the Emanuel (1991) convection parameterization. Criteria for surface evaporation and sensible heat fluxes are based on the RH–20 and RH–10, respectively (see Tab. 1 and 2).

### 3.1.2    Skill improvement

Figure 4 illustrates the skill of various detection criteria and demonstrates the benefit of the proposed heat and moisture diagnosis compared to previously employed methods. Validation per continent confirms previous findings: except for Antarctica, daily $P$ fluxes are reliably detected ($s_{pod} > 65\%$), indicate skill ($s_{PSS} > 0.5$; see Fig. S4) and show biases smaller than –2 mm d$^{-1}$. The highest underestimation of $P$ is found for South America with an average of –1.9 mm d$^{-1}$ (Fig. 4a).

The detection of $E$ shows a similar reliability for most continents (again, with the exception of Antarctica) with $s_{pod}$ values between 74 and 100%, and a consistent overestimation over land (Fig. 4b). Validating $E$ at the daily scale and globally indicates some benefits of filtering parcels for ABL processes: all criteria that filter out parcels not residing in the ABL (ALL–ABL,

RH–20, SOD08) show smaller biases than the FAS19 approach (top boxplots in Fig 4b). At the same time, the ALL–ABL approach yields similarly high probabilities of detection (dark grey and light grey boxplots in Fig 4b). However, only minor improvements are found when RH–20 is compared to ALL–ABL: while mean and median biases decrease and cluster around 0 mm d$^{-1}$ (red boxplot in Fig. 4b), mean and median skills are basically identical (red and light grey boxplots in Fig. 4b–c). Subdividing per continent confirms the north–south gradient from Fig. 3 for all approaches. For $E,$ the RH–20 criterion reduces the biases over land and ocean (red filled square and circle in Fig. 4b) compared to all other approaches (grey and blue filled squares and circles in Fig. 4b) but does not necessarily yield a higher probability of detection.

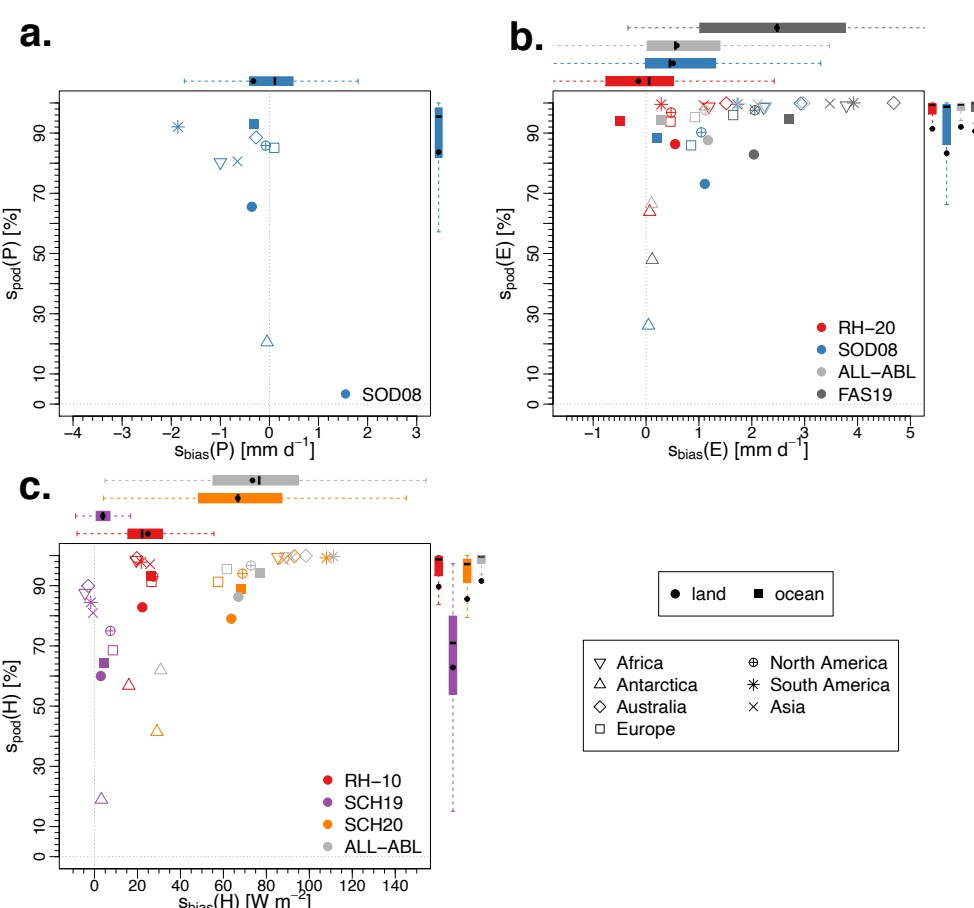

**Figure 4.** Bias ($s_{bias}$) and probability of detection ($s_{pod}$) of daily (**a**) precipitation, (**b**) surface evaporation and (**c**) surface sensible heat flux averaged over continents (indicated by different symbols), all land and ocean pixels and the period 1980–2016. Colors indicate different diagnosis criteria. The boxplots show the average distribution over all pixels globally (the back line shows the median; boxes show the 25% and 75% percentiles; dashed lines show 1.5 the interquartile range); black points in the boxplots indicate the average.

The detection of daily $H$ shows average biases between –4.4 W m$^{-2}$ (Africa) and 8.5 W m$^{-2}$ (Europe) for the SCH19 criteria (see purple points in Fig. 4c). Only considering temperature changes and neglecting humidity changes in the detection of $H$ leads to much higher biases (SCH20; orange points in Fig. 4c). Parsing for temperature changes and small humidity changes with the RH–10 criterion instead leads to intermediate biases ranging from 15.9 W m$^{-2}$ over Antarctica to 27.4 W m$^{-2}$ over North America (red points in Fig. 4c). However, in all cases, the RH–10 criterion leads to slightly higher probabilities of detection compared to the SCH19 criteria: over land, the probability of detection increases from 60.0 to 82.8% (purple and red points in Fig. 4c). On a global average, this causes an increase in reliability from 62.8 to 89.7% from SCH19 to RH–10, and is only topped by the ALL–ABL criterion, which exhibits a global probability of detection of 91.6% — but is also associated with the highest biases of 73 W m$^{-2}$ (see light-grey boxplots in Fig. 4c).

## 3.2 Attribution and bias-correction: heat

In the following, we assess the origins of heat and the impact of detection criteria and the bias correction on the spatio-temporal characteristics of these source regions. Therefore, we evaluate 15-day backward trajectories, arriving in the ABL of the three cities and their surroundings. To assess the impact of the $H$ detection criteria, we compare the three detection criteria (SCH19, SCH20 and RH–10; see Tab. 2) with the approach of counting all potential temperature increases in the ABL (ALL–ABL; see Tab. 2). In addition, we quantify and illustrate the impact of bias-correcting fluxes in the respective source regions.

### 3.2.1    Heat source regions

Figure 5 illustrates the surface source regions of heat advected to Beijing and its surroundings, showing the impact of different detection criteria in the columns and the impact of the bias-correction in the rows. All source region contributions were determined with linear discounting and attribution. Comparing the detection criteria, the source regions of heat appear similar in shape and extent. Air arriving in Beijing is typically warmed by surface sensible heating over land north-west of the city, including the Gobi Desert. To the south, the heat source regions are restricted by the Tibetan Plateau. Yet, the aggregated magnitude of heat advection, i.e., air warmed by sensible heating from the surface arriving in the ABL of the city and its surrounding, is different (Fig. 5a–b): the RH–10 criterion leads to an estimated advection of 498.4 W m$^{-2}$, which corresponds to 203% of the advection estimated with the criteria from SCH19 and 48% as estimated with SCH20 without any bias-correction. If all potential temperature increases in the ABL are considered to reflect sensible heating from the surface, around 1089.4 W m$^{-2}$ are estimated to arrive in the ABL of Beijing and its surroundings (Fig. 5d). If fluxes are bias corrected on a daily basis (Fig. 5e–h), heat advection is reduced to more realistic values: 412.0, 370.0, 403.3 and 406.73 W m$^{-2}$ for RH–10, SCH19, SCH20 and ALL-ABL criteria, respectively. Thus, the bias correction leads to fewer discrepancies in terms of total advection and to only minor differences in the illustrated source regions (Fig. 5e–h). In addition, it is worth mentioning that the source region patterns change as a result of the bias-correction: due to an overestimation of $H$, contributions from the north-east of Beijing are significantly reduced (cf. Fig. 5a and 5e; and Fig. 5d and 5h). For the strictest criteria to detect $H$ (SCH19), the sensible heat flux is often underestimated and leads to an increase of heat advection when biases are corrected for (cf. Fig.

5b and 5f). Contributions from open water and lakes, such as the lake Baikal, are reduced as a result of the bias correction (see Fig. 5e–5h). Results for Denver and Windhoek indicate similar findings (Suppl. Figs. S6–S7). It should be noted, however, that the length of the trajectory plays a significant role for heat advection, which is not constrained by any receptor quantity (unlike in the case of moisture in which precipitation can be used to constrain its advection; not shown).

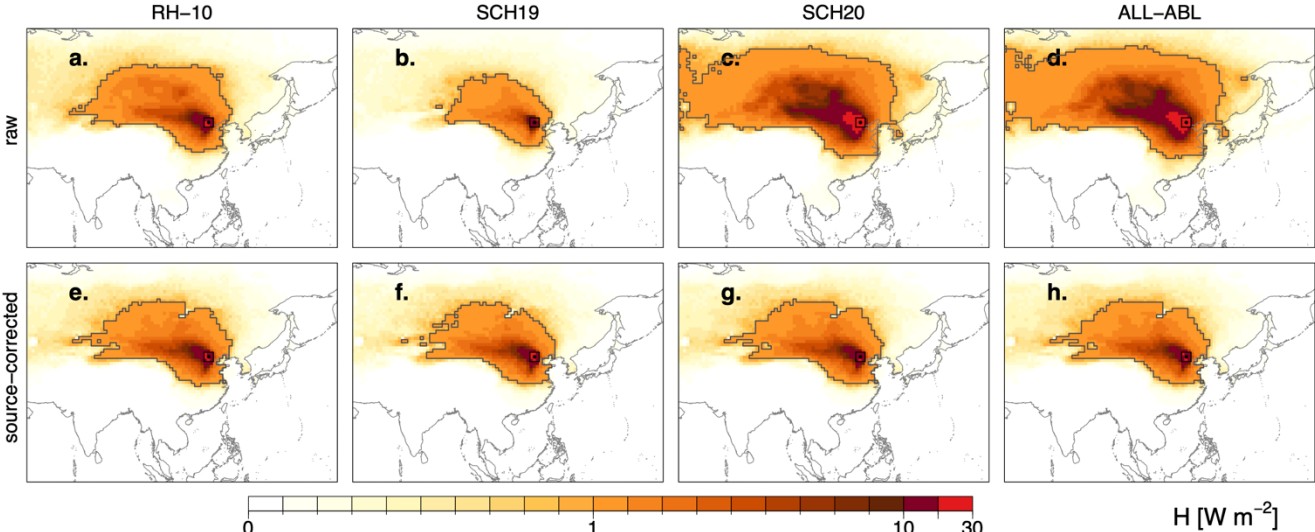

**Figure 5.** Source regions of heat for moisture-dependent thresholds using the RH-10 criteria (**a**+**e**), SCH19 criteria (**b**+**f**), SCH20 criteria (**c**+**g**), and all potential temperature increases in the ABL (ALL–ABL; **d**+**h**); averaged over the period 1980–2016. Rows show the biased source regions (**a**–**d**) and the source-corrected source regions of heat (**e**–**h**), respectively. The dark-grey lines mark the 1 Wm$^{-2}$ source regions, i.e., all contributions larger than 1 W m$^{-2}$. Note that the color scale is non-linear.

### 3.2.2   Relative contributions

To better highlight differences between the estimated source regions illustrated in Fig. 5, we calculate the relative contributions per backward day. Figure 6 shows the corresponding contributions per backward day normalized by each heat advection total (bars) as well as their cumulative sums (lines). The largest contributions originate from source locations 2–4 days away from each city (Fig. 6a). Around 44% of the heat is less than 3 days old, when it arrives in the cities and their surroundings — independent from the detection criteria (lines in Fig. 6a). The largest differences occur for the day of arrival and the first backward day: the relative contributions as detected with the SCH19 criteria amount to 19.3% on the arrival day and are thus 1.6% points higher than the contributions as estimated with RH–10 criteria (purple and red bars in Fig. 6a), 2.5% points higher

than the contributions estimated with SCH20 (purple and orange bars in Fig. 6a), and 2.7% points higher than the contributions estimated with ALL–ABL (purple and grey bars in Fig. 6a). As a consequence, all relative backward day contributions from backward days 2–15 are lower for the SCH19 criterion. In all cases, all contributions older than 5 days are below 5% (backward days 6–15). Source regions 15 days away from the cities contribute only 1.9%. If no bias correction is applied, remote source regions contribute relatively more heat (Fig. 6b). The bias correction thus increases the impact of nearby source regions.

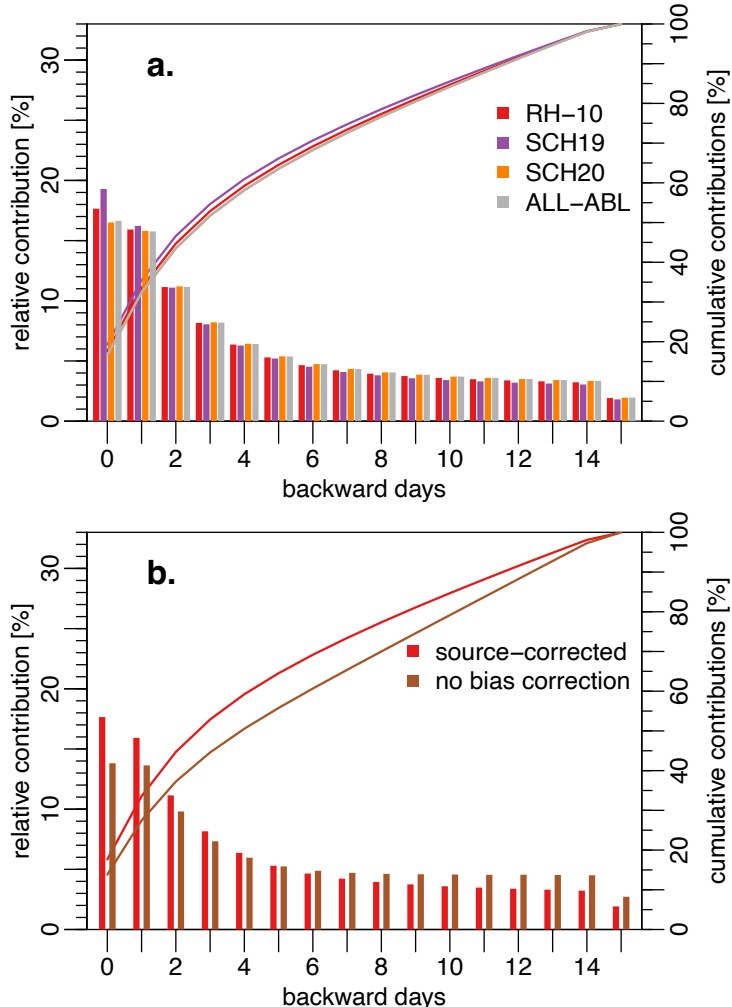

**Figure 6.** Relative (bars) and cumulative (lines) backward day contributions to heat advection (**a**) comparing the detection criteria and (**b**) illustrating the bias correction impacts, averaged over all cities and the period 1980–2016. The red bars/lines show the same setting (RH–10 and source-corrected contributions) in both sub-plots. The brown bars in (**b**) refer to the same criteria (RH–10) but are not bias corrected.

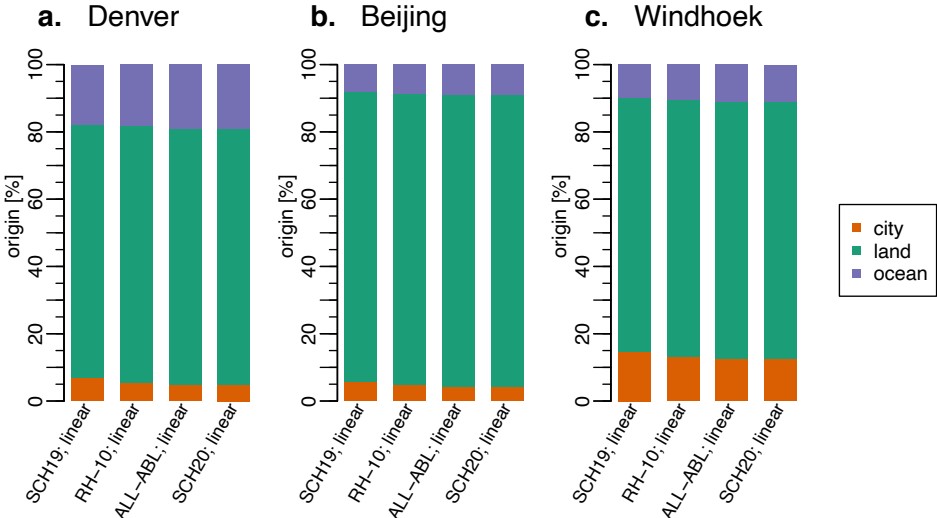

**Figure 7.** Origins of heat subdivided into local (orange), other land (green) and ocean (blue) source regions for (**a**) Denver, (**b**) Beijing and (**c**) Windhoek, averaged over the period 1980–2016, and for different *H* detection criteria (RH–10; ALL–ABL; SCH19; SCH20) along the columns. City origins indicate the percentage of heat originated from a 3°x3° box around the center of each city (i.e., local origins). Columns in each subplot are sorted in descending order of the local contributions (orange bars).

The minor differences in bias corrected backward day contributions, especially for heat that is less than 5 days old (Fig. 6a), lead to only minor differences in the spatial origins of heat. Figure 7 illustrates the origin of heat for all three cities, subdivided into the city and its surroundings, other land origins and oceans. As the same trajectories are being evaluated (all trajectories arriving in the ABL of each city and its surroundings), and only the identified source locations on these trajectories may differ, the choice of the *H* detection criterion has only a minor impact on the relative heat origins. Local origins (i.e., heat originating from the 3°x3° box around each city) vary between 4.8 and 6.7% (Denver; orange bars in Fig. 7a), 4.2 and 5.7% (Beijing; orange bars in Fig. 7b), and 12.6 and 14.6% (Windhoek; orange bars in Fig. 7c). Other land contributions vary between 75.5 and 76.2% (Denver; green bars in Fig. 7a), 86.2 and 86.8% (Beijing; green bars in Fig. 7b) and 75.6 and 76.4% (Windhoek; green bars in Fig. 7c); leaving similarly small variations to the oceanic origins (purple bars in Fig. 7). Note that only bias corrected contributions are compared here — the difference between raw and bias corrected contributions is much larger (not shown). In general, the local contributions decrease with relaxing filter criteria, i.e., the lowest recycling estimates stem from ALL–ABL and SCH20.

### 3.3 Attribution and bias-correction: moisture

Analogous to the origin of heat, the origins of precipitation for the three cities and their surroundings are estimated using 15-day backward trajectories, and the impact of detection criteria and the bias correction on the spatio-temporal characteristics of

these source regions is assessed. Here, four detection criteria for $E$ are compared (RH–20, SOD08, FAS19 and ALL–ABL), using the same $P$ criterion — thus evaluating the same trajectories. In addition, the impact of the attribution methodology is illustrated (linear discounting/attribution and random attribution). As the associated source regions of moisture are further constrained by a sink quantity (i.e., precipitation), multiple bias-correction methods are compared as well.

### 3.3.1    Precipitation source regions

Figure 8 illustrates the source regions of precipitation for Beijing and its surroundings, assessing the impact of different detection criteria in the columns and the impact of the bias-correction in the rows. All source region contributions in Fig. 8 were determined with linear discounting and attribution. The un-corrected source regions of $P$ as determined with different $E$ detection criteria appear visually very similar (Fig. 8a–c). Note that the contributions have been upscaled to match diagnosed $P$ estimates (see Supplementary Material for details). Compared to the source regions of heat, the source regions of $P$ are more

concentrated around the sink region. The largest contributions (>10 mm y$^{-1}$) originate south-west of the city and regions north of the Tibetan Plateau are only minor source regions (<1 mm y$^{-1}$; Fig. 8a–d). As precipitation is slightly overestimated over Beijing and its surroundings (cf. Fig. 4a), bias correcting with $P$ from ERA-Interim leads to a decrease of all contributions but leaves the patterns unchanged (Fig. 8e–h). However, further correcting for $E$ in the source regions changes the relative contributions of each pixel to $P$ over Beijing and thus affects the spatial patterns (Fig. 8i–l): due to the overestimation of $E$

especially over mountainous areas (cf. Fig. 4b), some $E$ contributions are significantly reduced. Averaged over the period 1980–2016, the bias-corrected source regions for all detection criteria are very similar in shape and extent (Fig. 8i–l).

Figure 9 shows the same source regions, again with varying detection criteria along the columns and varying bias correction methods along the rows, but is based on the random attribution of source contributions (see Sect. 2.3.2) — thus allowing for

deviations from perfectly-mixed conditions. Applying the random attribution, the (biased) source regions extend much farther to the west and contributions close to the city appear much smaller (cf. Fig. 8a–d and Fig. 9a–d). Similar to the sink bias correction under well-mixed conditions (i.e., for the linear discounting/attribution), source contributions are slightly increased if $P$ is corrected for (Fig. 9e–h). Yet, due to the larger extent of the source region, an additional source bias correction changes the source shape and extent (Fig. 9i–l): contributions from mountainous areas are again reduced, highlighting the Tibetan

Plateau and the complex terrain around it. As a consequence of the underestimation of $E$ over large parts of the ocean (cf. Fig. 4b), and oceanic contributions south-west of the city are increased. Despite the random factor in the random attribution, the source regions appear similar for all detection criteria (Fig. 9i–l). However, shape and extent as determined with the random attribution appear slightly different compared to the linear discounting/attribution applied to estimate the same source regions in Fig. 8 and highlight the uncertainty associated with the well-mixed assumption. Results are similar for Denver and Windhoek

(see Figs. S8–S11).

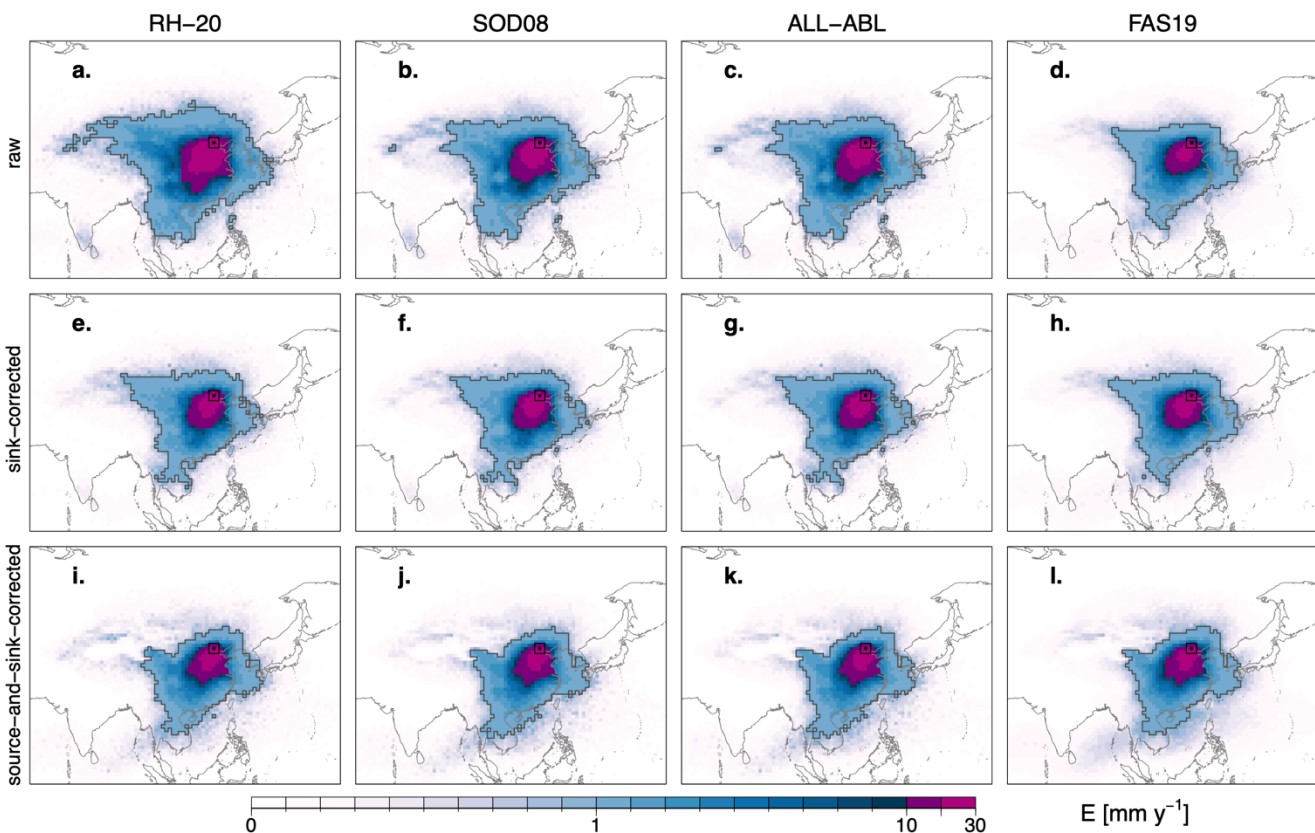

**Figure 8.** Source regions of precipitation for Beijing and its surroundings, varying the detection criteria for E (along columns) and the bias-correction methods (along rows) as estimated with linear discounting of losses *en route* and linear attribution. Along the columns, the following E criteria are employed: RH–20 (**a**+**e**+**i**), the SOD08 criteria (**b**+**f**+**j**), the ALL–ABL criteria (**c**+**g**+**k**), and the FAS19 criteria (**d**+**h**+**l**). Along the rows, the following bias correction methods are employed: no bias correction (**a**–**d**), sink bias correction using $P$ (**e**–**h**) and a source- and sink bias correction using $E$ and $P$ (**i**–**l**). The dark-grey lines mark the 1 mm source regions, i.e., all contributions larger than 1 mm year$^{-1}$. The black dot and the surrounding box mark Beijing and the 3°x3° sink region. All source regions are averaged over the period 1980–2016. Note that the color scale is non-linear.

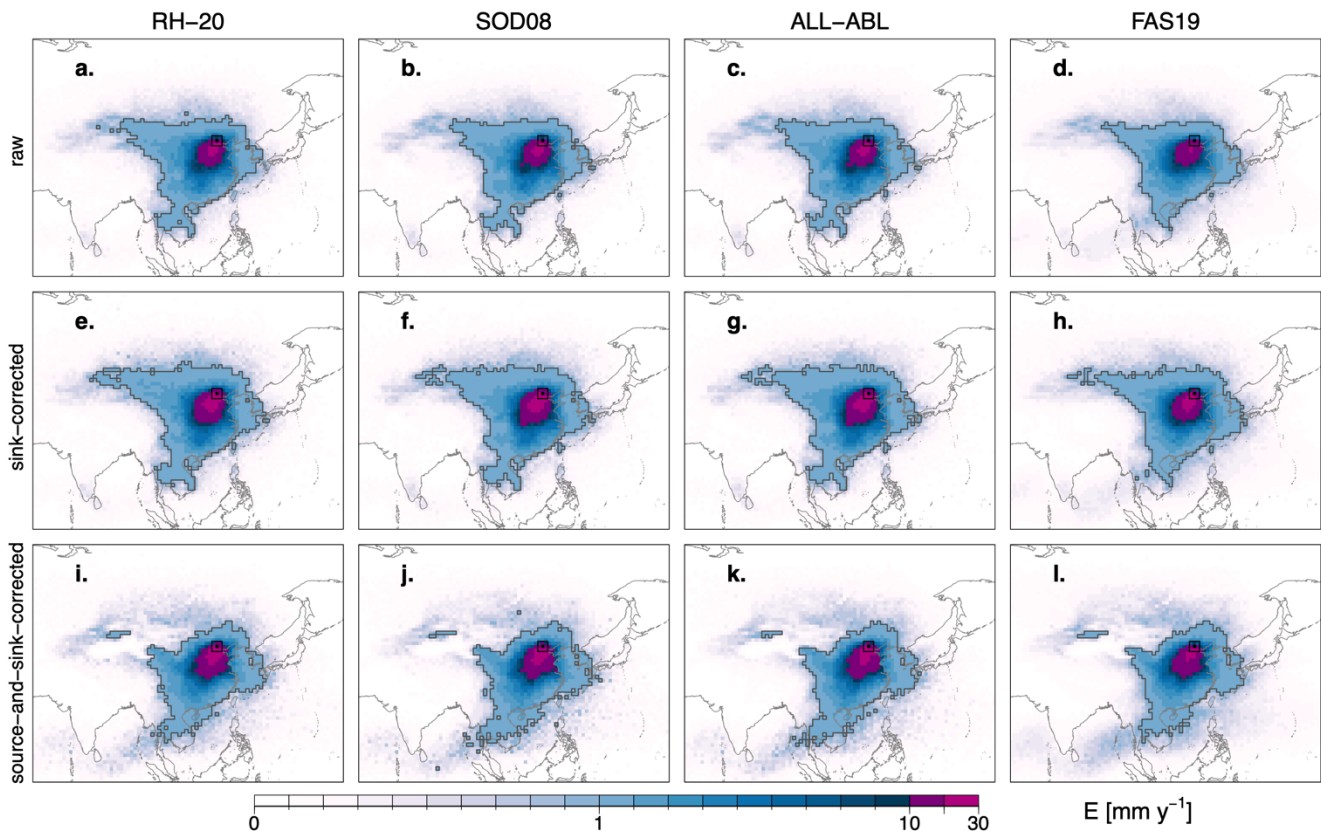

**Figure 9.** Same as Fig. 8 but applying the random attribution method. Note that the color scale is non-linear.

### 3.3.2 Relative contributions

The differences arising from the perfectly-mixed assumption, i.e., between linear discounting/attribution and random attribution, are highlighted in the evaluation of backward day contributions: Fig. 10 shows relative contributions per backward day, analogous to Fig. 6, but for various $E$ detection criteria (Fig. 10a), various bias-correction methods (Fig. 10b) and the two attribution methodologies (Fig. 10c). Again, all contributions are averaged over the three cities evaluated here. For $P$, the largest contributions come from source locations one day away (bars in Fig. 10a) from the cities. The contribution of the same day (i.e., backward day 0) is reduced compared to heat for two reasons: (i) $E$ and $P$ are both estimated from the net moisture flux (Eq. 1) — and since this analysis filters for $P$, corresponding $E$ contributions are lower. Around 54% of moisture originates from source locations less than two days away from each city (Fig. 10a) — independent of the detection criteria. The largest contributions occur on the first backward day, between 25.7 and 27.9% for the FAS19 criterion (accounting for above-ABL sources) and the ALL–ABL approach, respectively (light grey and dark grey bars in Fig. 10a). The FAS19 criterion shows a

slight displacement of contributions towards older contributions (light grey bars in Fig. 10a). This impact is on the same order of magnitude that the bias correction methods exhibit (Fig. 10b): the bias correction of $E$ in the source regions decreases contributions nearby (backward days 1–2) and increases relative contributions farther away, when compared to a sink bias-correction only. The largest differences are, however, a result of the attribution methodology and thus relate to assumptions on the state of individual parcels. Fig. 10c illustrates the difference between the two attribution methods for the RH–20

criterion: permitting deviations from perfectly-mixed conditions by applying the random attribution leads to a shift towards source regions farther away. The 54% of $P$ that originate from source locations two days prior to the $P$ event decreases to 40% if random attribution is applied and ABL processes are filtered for. For the random attribution, trajectories of at least four days are required to attribute most of the moisture (i.e., 57%; black line in Fig. 10c). The remaining contributions are shifted towards source locations farther away, with more than 2.2% evaporating 15 days prior to the $P$ event (purple bars in Fig. 10c). In

perfectly-mixed conditions as estimated with the linear discounting/attribution, this contribution is reduced to 0.9% (red bars in Fig. 10). Similar differences between the attribution methods emerge for the two other detection criteria (not shown). Altogether, these differences suggest that the detection criteria play only a minor role, and that the state of the air parcel represents a crucial assumption for the estimation of source regions. Similarly, bias-correcting not only the sink quantity (i.e., precipitation — cf. Fig. S12 and Tab. S2), but also the source quantity reduces the impact of the detection criteria on the

estimated source regions and increases credibility of the results.

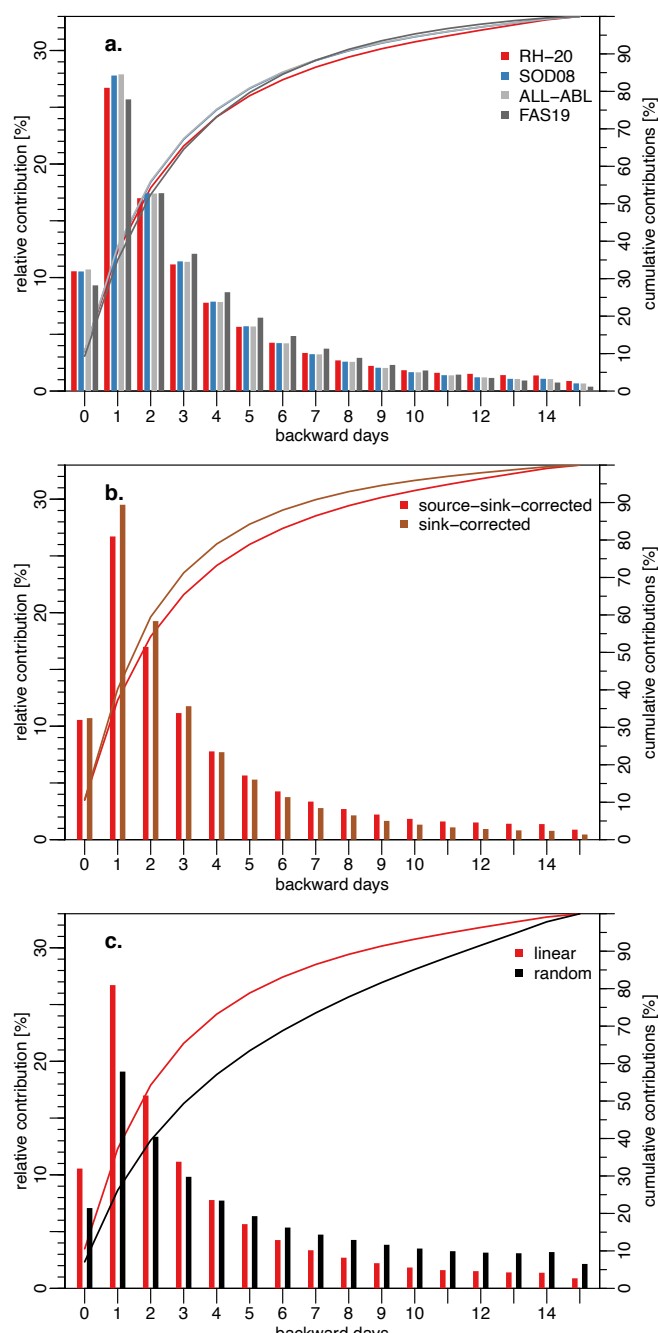

**Figure 10.** Relative (bars) and cumulative (lines) backward day contributions to precipitation (**a**) comparing the detection criteria, (**b**) illustrating the bias correction impacts, and (**c**) varying the attribution method, averaged over all three cities and the period 1980–2016. The red bars/lines show the same setting (RH–20, employing linear discounting and attribution, and source- and sink-corrected contributions) in all sub-plots.

Analogously to Fig. 7, Fig. 11 illustrates the origin of precipitation for all three cities. Here, the orange bars indicate the local precipitation recycling ratio, i.e., the ratio of $P$ originating from the city and its surrounding. Note that only source and sink bias-corrected estimates are shown. The recycling ratios illustrate the uncertainty associated with the detection criteria for $E$ and the attribution method: for Denver, the recycling ratio varies between 6.0 and 10.3% with the largest differences arising from the two attribution methodologies that prescribe the state of the air parcel at the time of precipitation events (orange bars in Fig. 11a). However, the detection criteria also cause differences in the recycling ratio: assuming perfectly-mixed conditions by using linear discounting and attribution of precipitation source regions leads to recycling ratios between 9.3 and 10.3% (for FAS19 and ALL–ABL, respectively; orange bars in Fig. 11a). Accounting for deviations from well-mixed conditions by applying the random attribution results in recycling ratios between 6.0 and 7.3% (for FAS19 and ALL–ABL/SOD08, respectively; orange bars in Fig. 11a). Similar relationships are found for Beijing, but with overall lower estimates for the local origin, ranging from 3.0 to 5.8% (orange bars in Fig. 11b). For Windhoek, the largest recycling ratio of 10.9% are estimated using the SOD08 and ALL–ABL criteria together with linear discounting/attribution (orange bars in Fig. 11c). In contrast, FAS19 criteria together with the random attribution lead to the lowest estimates of local recycling for Windhoek (5.2%; orange bar in Fig. 11c).

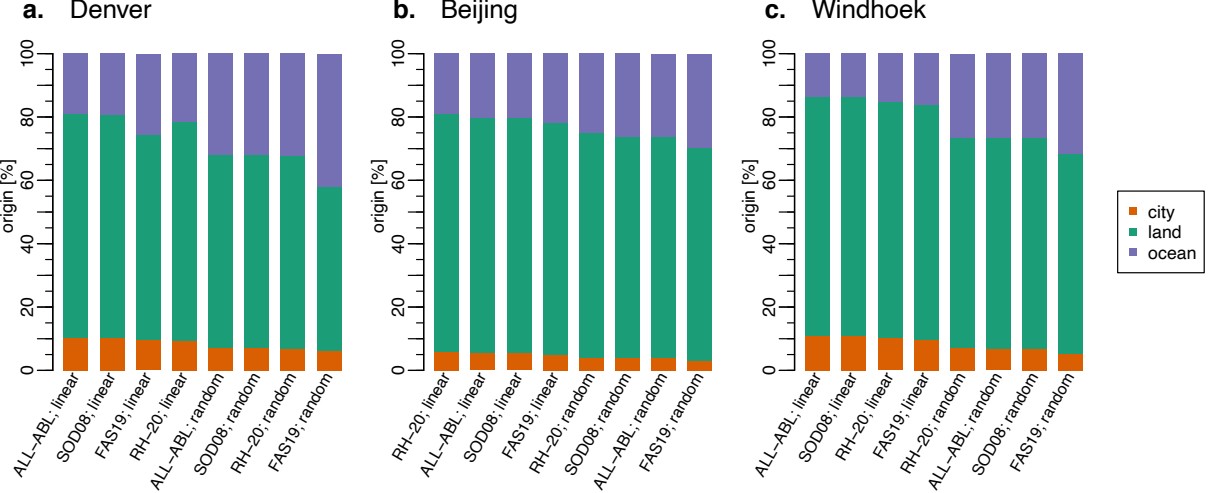

**Figure 11.** Origins of precipitation subdivided into local (orange), other land (green) and ocean (blue) source regions for (**a**) Denver, (**b**) Beijing and (**c**) Windhoek, averaged over the period 1980–2016, and for different $E$ detection criteria (RH–20; SOD08; ALL–ABL; FAS19) and attribution methods (linear discounting and attribution; random attribution) along the columns. City origins indicate the percentage of heat originated from a 3°x3° box around the center of each city (i.e., local origins). Columns are sorted in descending order of the local contributions (orange bars).

The impact of both, attribution methodology and detection criteria, has even larger impacts on the relative contributions from other land areas (green bars in Fig. 11): between 52.0 and 70.5% of $P$ over Denver originates from land (for FAS19 with random attribution and ALL–ABL with linear discounting/attribution, respectively; green bars in Fig. 11a). For Beijing and Windhoek, these values vary between 67.4 and 75.2% and 63.1 and 75.5%, respectively (green bars in Fig. 11b+c). In all cases the attribution methodology and the associated assumption of perfectly-mixed conditions is again showing the biggest influence. Overall, allowing for deviations from the perfectly-mixed conditions along with the least restricting detection criteria (FAS19; accounting for above-ABL source regions) leads to the lowest estimates of local contributions.

## 4 Discussion

This study introduced new criteria for the detection of $E$ and $H$ from specific humidity changes in air parcels from Lagrangian trajectories, further exploiting coupled temperature- and moisture changes. These novel criteria were designed to complement the existing ones and parse out the full uncertainty arising from the detection of processes and source locations. A global validation based on two-step trajectories reveals that the proposed coupled heat and moisture criteria show biases on the same order of magnitude than existing methodologies. However, this coupled diagnosis enables a less subjective use of thresholds for the estimation of source–receptor relationships. Further, region-dependent thresholds may be employed that depend on research question and availability of data/parcels in the simulation and region of interest, i.e., in some cases, it may be better to use relaxed detection criteria in order to evaluate more parcels and increase the probability of detection, rather than relying on fewer parcel trajectories to establish quantitative source–receptor relationships. Yet, if thresholds become less restrictive and parcels are not being filtered for processes (i.e., all parcels are evaluated to contribute to all processes), the analysis may become less quantitative and approaches the methodology proposed by Stohl and James (2004). The presented framework and validation measures are suitable for threshold calibration and may further be used to determine spatio-temporal limits for a trustworthy analysis of source–receptor relationships, given a Lagrangian simulation with a limited number of parcels. These are expected to vary with the driving atmospheric forcing (and its spatial and temporal resolution, see expected impact studies for using ERA5 instead of ERA-Interim, e.g., Hoffmann et al., 2019), but also with the number of parcels that are being tracked. Thus, we suggest the use of these or similar validation measures to increase the credibility and usefulness of Lagrangian analyses beyond this study.

Overall, the detection criteria for $E$ and $H$ explored in this study show little impacts on the resulting source–receptor relations (compared to the attribution algorithm and the well-mixed assumption). While the novel and relative-humidity-based heat and moisture diagnosis criteria did not show substantial improvements over other criteria (e.g., Sodemann et al., 2008), we highlight the more general applicability of these criteria for global applications. Moreover, despite the fact that these criteria sample for different air masses (using a maximum instead of a minimum threshold), the resulting source–receptor relationships

did not deviate much. Among the detection criteria tested here, the ALL–ABL approach and the FAS19 criteria, that also considers above-ABL sources of moisture, showed the largest discrepancies.

Due to the scarcity of observations to validate source–sink and source–receptor relationships, it remains difficult to illustrate the benefit of the detection criteria, and to validate the realism of the random attribution algorithm. However, we believe that
the presented methods may be valuable to address and exploit various scientific questions: the coupled detection criteria could help to assess drivers of climate-induced changes in source–sink relationships, e.g., a recent study showed that changes in sensible heating from the land surface have significantly contributed to global precipitation changes over the last century (Myhre et al., 2018). Tracking the origin of heat and moisture in a coupled manner may help to unravel the regionally dominant drivers of precipitation change. Furthermore, applying linear discounting and attribution, Läderach and Sodemann (2016)
estimated the residence time of water vapor in the atmosphere as the average time between the surface evaporation event and the precipitation event. The resulting residence time was subject to many discussions recently (e.g., van der Ent and Tuinenburg, 2017; Sodemann, 2020), and appeared to be biased towards lower mean residence times of water vapor in the atmosphere compared to other studies. As illustrated here, the discrepancy between these residence times may not only be an issue of definition, but may further be attributed to the assumption that parcels are always perfectly-mixed and thus source
regions contribute linearly to precipitation events, which potentially causes an overestimation of nearby source regions (cf. Figs. 10 and 11). We thus suggest exploiting the presented random attribution methodology to allow for deviations from the well-mixed assumption in the estimation of the average residence time; the results may then be compared to the estimates from, e.g., Läderach and Sodemann (2016) and van der Ent and Tuinenburg (2017). It is further noted that similar linearity assumptions are applied in other Lagrangian studies that track water instead of air parcels (e.g., Tuinenburg and Staal, 2020).

Further, as shown here, a bias-correction can be employed to increase reliability and reduce uncertainty of source–receptor relationships. Further, instead of tracking only moisture for precipitation, the origin of (ABL) moisture in the atmosphere may be tracked to approximate *moisture advection*. In this study, heat and moisture advection have not yet been constrained by a receptor quantity, but integrated water vapor can be employed to constrain moisture advection. Despite this apparent lack of a
850 receptor quantity for heat advection, it is assumed that anomalies are reliably represented if the same thresholds for detection and the same methods for quantification of source contributions are employed. In addition, the choice of a bias-correction methodology may depend on the research question and not all methodologies may appear adequate for all cases.

Finally, it is highlighted that the proposed framework only addresses the uncertainty inherent in the evaluation of trajectories
from Lagrangian simulations. Thus, for the purpose of this study, uncertainty arising from the simulation itself, e.g., through the number of parcels, the employed convection scheme, or errors arising from the accuracy of the analyzed trajectories, are not considered. These are, however, expected to influence the results as shown in, e.g., Sodemann (2020) and Tuinenburg and Staal (2020). Yet, this restriction to uncertainties inherent in the evaluation of trajectories facilitates a general applicability of

this framework to other Lagrangian models than the one employed here (FLEXPART). For example, the framework could be ad-hoc employed to simulations with, e.g., LAGRANTO (Sprenger and Wernli, 2015) or TRACMASS (Döös et al., 2017) that also track air parcels; and parts of the framework (such as the different attribution methodologies) are equally applicable to simulations with, e.g., UTRACK (Tuinenburg and Staal, 2020), that tracks water parcels instead.

## 5 Summary and Conclusions

An increasing body of literature aims at estimating the source regions of precipitation. Simultaneously, other source–sink and source–receptor relationships, such as the source regions of heat, enable the establishment of spatiotemporal dependencies in land–atmospheric processes. However, the established relationships remain difficult to validate due to the scarcity of observations, and little effort has been made to increase the credibility or to assess the uncertainty of these relationships. Here, we introduced a unified framework for the process-based evaluation of atmospheric trajectories from Lagrangian models. The framework entails a coherent diagnosis and validation of land surface fluxes from two-step trajectories using heat and moisture criteria, the attribution of source region contributions to a sink/receptor quantity using multi-day trajectories, and the bias-correction of the established source–sink and source–receptor relationships. As such, the framework offers the possibility to explore and quantify uncertainties inherent in the source–receptor relationships. Illustratively using simulations from the Lagrangian model FLEXPART driven with reanalysis data, we demonstrated the applicability of the framework and reported global error quantities expressed as biases and probabilities of detections for specific processes, such as evaporation and precipitation. Moreover, the uncertainty rooted in the evaluation of Lagrangian simulations to establish source–sink and source–receptor relationships of moisture and heat was assessed for three cities and their surroundings. The comparison showed that the estimation of source regions is subject to several uncertainties: while the choice of diagnosis criteria has an impact, the largest uncertainty of the source regions stems from the attribution methodology and the well-mixed assumption. Bias-correcting source and sink quantities decreases the uncertainty arising from the choice of criteria, but large discrepancies remain between the two attribution methodologies. These results suggested a potential overestimation of nearby source regions using conventional attribution methods (linear discounting and attribution) and may help to explain discrepancies in the estimated residence time of water vapor in the atmosphere.

**Code and data availability.** ERA-Interim data were downloaded from the ECMWF and are publicly available through https://apps.ecmwf.int/datasets/data/. The source code for FLEXPART can be accessed through https://www.flexpart.eu. The output of the global FLEXPART–ERA-Interim simulations (30 TB) is available upon request from the corresponding author. The source code version of the framework, referred to as 'Heat And MoiSture Tracking framEwoRk' (HAMSTER v1.2.0) is available from https://github.com/h-cel/hamster under the GPL-3.0 license. The exact version of the software used to produce the results used in this paper is archived on Zenodo (https://doi.org/10.5281/zenodo.5788506). The post-processed outputs from HAMSTER v.1.2.0 are available on Zenodo (http://doi.org/10.5281/zenodo.5793038). Analysis scripts, to run

HAMSTER and to reproduce the figures, are published on GitHub (https://github.com/jkeune/hamster_analysis_gmd) and Zenodo (http://doi.org/10.5281/zenodo.5793140).

**Author contributions.** J.K., D.L.S. and D.G.M conceived the study. D.L.S. and J.K. created the python framework available on GitHub. J.K. and D.L.S. designed the experiments, processed the data and performed the analyses. J.K. designed the layout of the manuscript and led the writing. All authors have been involved in interpreting the results, discussing the findings, and editing the manuscript.

**Competing interests.** The authors declare that they have no conflict of interest.

**Acknowledgements.** This study is funded by the European Research Council (ERC) under grant agreement 715254 (DRY–2–DRY) and the European Union H2020 project no. 869550 (DOWN2EARTH). J.K. is further funded by the Research Foundation–Flanders (FWO). The computational resources and services used in this work were provided by the VSC (Flemish Supercomputer Center), funded by the FWO and the Flemish Government, Department of Economy, Science and Innovation (EWI). The authors are grateful for the support from Raquel Nieto and Luis Gimeno (Environmental Physics Laboratory, University of Vigo) and Anita Drumond (Institute of Environmental, Chemical and Pharmaceutical Sciences, Federal University of São Paolo) for providing the original global FLEXPART simulation. Finally, the authors thank the editor Simon Unterstrasser for handling the manuscript, and Harald Sodemann and two anonymous reviewers for very constructive comments during the review process.

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
