# Peer review of "A unified framework to estimate the origins of atmospheric moisture and heat using Lagrangian models"

_Geoscientific Model Development, 2021_

## Referee Comment (RC1)

Review of paper

**A holistic framework to estimate the origins of atmospheric moisture and heat using a Lagrangian model**

by J. Keune et al.

submitted to *Geosci. Model Dev.*

This is a much-needed study contributing to quantitatively assess the reliability of Lagrangian source diagnostics. As pointed out by the authors, these diagnostics potentially provide very valuable insight into the atmospheric moisture and heat budgets; however, it is intrinsically difficult to quantify errors and uncertainties associated with these methods. I therefore fully support the intention of this study, and to a large degree also the used methodologies; however, in its current version the paper is difficult to read. I find the notation confusing in several (important) places and I could not understand the idea and implementation of the "random attribution method". Therefore, major revisions are required to streamline the paper, clarify concepts and notation, and make the paper in the end more reader friendly. It will then be a valuable contribution to the field.

**Major comments**

A) Line 89: "the application of these models and tools to assess diabatic heating and heat transport lags behind". I am not sure that I agree with this statement. The study by Pfahl et al. (2015) is an important one but certainly not the first one in this direction. Early applications of trajectory computations with reanalysis data in the 1980ies and 1990ies looked at latent heating in cyclones and warm conveyor belts, and how this latent heating affects the potential vorticity structure of the systems (e.g., Whitaker et al., 1988; Reed et al., 1992; Wernli and Davies, 1997; Rossa et al., 2000). These were not yet full budget studies, but I would claim that Lagrangian methods first looked at latent heating and only about 1-2 decades later also at moisture sources and transport.
References:
Reed, R. J., Stoelinga, M. T., Kuo, Y.-H., 1992: A model-aided study of the origin and evolution of the anomalously high potential vorticity in the inner region of a rapidly deepening marine cyclone. Mon. Weather Rev., 120, 893–913.
Rossa, A. M., H. Wernli, and H. C. Davies, 2000. Growth and decay of an extratropical cyclone's PV-tower. Meteorol. Atmos. Phys., 73, 139-156.
Whitaker, J. S., Uccellini, L. W., Brill, K. F., 1988: A model-based diagnostic study of the rapid development phase of the President's Day cyclone. Mon. Weather Rev., 116, 2337–2365.

B) Line 94: I think this is a slightly misleading summary of the Quinting and Reeder (1997) study. They mainly emphasized the role of adiabatic descent, and their last sentence of the abstract says "Likewise, the role of the local surface sensible heat fluxes is deemphasized.".

C) I am completely lost with understanding the "random attribution" method (section 2.3.2) for several reasons. First, I don't understand the notation "length nt": is this n times t? And then later, what is ix, nx, … an later n_min … ?? Most likely this requires a schematic where you explain also the notation. Then how can you use Delta q,random in step 1 if you calculate it only in step 3? Then I am completely lost with step 2, and I also don't understand the general motivation for doing this. Can you explain this method and the motivation for it in a much better way?

D) Fig. 3b is a key figure of this study. Since, e.g., the methods SOD08 and RH-20 vary in multiple ways (additional RH criterion, different Delta q threshold) it would be interesting to know which change had the largest effect. It would be very useful to have a more in-depth discussion of which criteria affect the results shown in Fig. 3b.

E) While I agree that this study addresses important technical aspects of moisture and heat source identification, the text is rather heavy to read, and the results are mainly presented in a statistical way, which is hiding a bit what is going on technically. To me, it would be useful to have a didactic example, starting with a single trajectory and then a set of trajectories, which helps me better understand the differences between the methods and the effects of the bias corrections etc.

F) The random attribution method has an important effect on estimating the transport time between uptake and rainout (Fig. 9c). With the random attribution method, you have much more "old uptakes" and therefore you have more long-range transport and remote sources (Fig. 10). This is very interesting and most likely an important result of this study (see also your discussion in lines 672-682). My problem is just that I didn't understand the random method (see my point C above) and that I don't find physical reasons in the paper why the random method has these effects compared to the linear attribution method. Again (see my point E above), a case study with a few trajectories might be very helpful for explaining what is going on.

G) How sensitive are your main conclusions with respect to the total number of parcels calculated with FLEXPART? I don't ask you to redo a certain analysis with more parcels (this might be too time consuming), but it might be interesting to look at the effects of reducing the number of parcels.

**Minor comments**

1) Line 18: "synergistic impacts" on what? And what is meant by "a cohesive assessment", maybe "coherent assessment"?

2) Line 28: here reference to Sodemann et al. (2008) would be more appropriate than Sprenger and Wernli (2015).

3) Line 33 and in other places: I think references should be listed in chronological order.

4) Lines 36-74: I appreciate this nice summary of Lagrangian approaches to identify moisture sources. What may be missing is a remark that Lagrangian approaches suffer from accuracy errors of trajectory computations, which can be substantial for trajectory integrations over several days. These errors stem from limitations of the numerical schemes, and most likely more substantially from the limited temporal resolution of wind fields available for offline trajectory computation.

5) Line 88: "A myriad" seems a bit exaggerated.

6) Line 98: maybe this summary of recent Lagrangian heat wave studies should also include the one by Zschenderlein et al. (2019): Zschenderlein, P., A. H. Fink, S. Pfahl, and H. Wernli, 2019. Processes determining heat waves across different European climates. Quart. J. Roy. Meteorol. Soc., 145, 2973–2989.

7) Line 159: I think that the notation $Delta\_q(t_0 - t_{-1})$ is not ideal. $Delta\_q$ does not so much depend on the time difference but rather on the two times themselves. I therefore suggest that $Delta\_q(t_0; t_{-1})$ would be more appropriate, or maybe even $Delta\_q(t_{-1}; t_0)$.

8) Line 174 and elsewhere: units should not be in italics.

9) Line 181: I was first confused and thought that $z$ is a function of $m$, but your m is the unit of z. I don't think that you need to mention units in the text, or you write "$z$ (in m)".

10) Line 190: either "applied" or "used"

11) Line 198: is $Delta\ q_i$ the absolute change? I assume that $Delta\ q_i$ is negative if mixing with free tropospheric air occurs and then the $Delta\ q_i$ condition is trivially fulfilled and does not help to exclude mixing with dry tropospheric air. Please clarify.

12) Lines 213/215: I am not sure that I understand these RH criteria. Evaporation is particularly intense for dry air, and so why shouldn't intense (ocean) evaporation not lead to a strong increase in RH? And for the heat flux H, I assume that H leads to warming and therefore to a

reduction of RH, so Delta RH should be negative, meaning that the criterion Delta RH < 10% is trivially fulfilled(?). Please clarifiy.

13) Line 235: I expect the opposite: with 6-hourly data we estimate the diurnal cycle of the ABL height poorly and therefore the ABL height criterion might be important. If we had hourly data (e.g., with ERA5) then there should be less sensitivity with respect to the design of the height criterion.

14) Line 260: "for E for P" should read "for E and P".

15) Line 274: Strange formulation "Due to the consideration …, mass and energy are conserved …". I think mass and energy conservation is valid independent of what is considered by the algorithm(?).

16) Line 285: "time step (t)" should read "time step $t$" (italics).

17) Line 288: I don't see the need to introduce a new notation 1_Delta Phi … for this function. This is the Heaviside step function, which in this case could be written as H(–Delta Phi_j).

18) Lines 285-299: please clarify whether everything here is identical to SOD08, or whether you introduced some modifications.

19) Lines 300-307: I don't understand why this explained here after the linear discounting, appears a bit out of place.

20) Line 358: I again struggle with the notation: is x here an index? If yes, why then do you write it as a superscript of S_LM? I realize that at this point of the paper I cannot really follow any further, mainly because of confusing notation. What are indices, what are coordinates, what are just subscripts/superscripts …? Does "LM" mean "Lagrangian model" or something else?

21) Line 497: typo in "heat"

22) Fig. 6: I cannot find the information how you define "local"; does this "local region" have the same size for all cities?

23) Figs. 7 and 8: I find it very difficult to see something in these many panels, except that they all look very similar. I think the smooth blue-only color bar does not help. Can you find an alternative way of visualizing the results that is more insightful for the reader and that makes the differences more apparent? Maybe by showing difference fields from a "reference setup".

24) Line 661: why are the new criteria better to assess global warming trends?

---

## Referee Comment (RC3)

**Review of "A holistic framework to estimate the origins of atmospheric moisture and heat using a Lagrangian model" by Keune et al., submitted to GMD**

Keune et al present a framework for the evaluation of Lagrangian methods for quantitative offline-diagnosis of heat and moisture from air-parcel trajectories. There exists quite a number of studies with similar yet different concepts and implementations, and the community is clearly in need of ways to enabe comparison and verification exercises. In this regard, the paper is clearly a needed and welcome contribution to the literature. In addition, the manuscript is well-written and most of the material clearly presented. I have a number of comments with respect to some of the literature and interpretation, and to the presentation of figure material, detailed below. Since my attention has mostly been on the moisture source identification, I mainly focus my comments on those aspects of the paper. I have no reason to conceal my identify, also because it will be quite evident from my comments that I am the main author of one of the methods assessed here.

Harald Sodemann

**Major comments**

1. The description of the accounting procedure is not entirely clear or may miss one important point. I recommend to separate two aspects more distinctly, (a) considering the fractional contributions of the uptakes (source contributions) during the uptake (i.e. how much does a source contribute to what is in the air parcel at the end of the time interval), (b) discounting all previous contributions according to their relative share of all water vapour in an air parcel. Step (a) is a fundamental change from methods without accounting, that only consider the local humidity change, rather than the fractional contribution times the arrival precipitation.
2. It should be mentioned somewhere that there is a physical/theoretical basis for the assumption that all sources contribute to precipitation en route and at the arrival point according to their share of the total water vapour in an air parcel, namely the assumption of well-mixed conditions within an airparcel within a 6-h time interval. The random accounting procedure that is presented in Sec. 2.3.2 does not have such a theoretical basis. Other than being a sensitivity test, it is unclear how reliable/meaningful the results obtained with such a random attribution approach are in terms of physical interpretation.
3. Such Lagrangian offline diagnostics as discussed here will always be imperfect approximations of how water vapour moves in a model simulation. What is your take on the question, what level of accuracy we actually can expect from such

methods?

4. The term "holistic" in the title has in my perception connotations that are not well covered by what the proposed framework actually encompasses (being valid for heat and moisture specifically, rather than "everything" as holistic could imply). How about replacing with a more limited word, such as "unified" or "generic"?

5. The paper currently seems to introduce both a verification framework, and a modified source accounting algorithm with additional parameters. A clearer statement of this dual objective, and a potentially clearer separation of both aspects in the manuscript (method/results) could be beneficial to avoid confusion with the reader about the focus and intent of the paper.

**Detailed comments**

L. 35: "while others trace air parcels and their properties": I first misread this to comprise also the accounting-type methods, such as S08, but then two paragraphs later understood, how you build up the story. Maybe it can be made more clear how you distinguish the different aspects, and still include S08 in the list of references in L28?

L. 35: FLEXPART and Lagranto trace airmass motion and interpolate boundary-field variables to the parcel position.

L. 73: There are a few additional references that use the S08 method, that may be relevant here, including Sodemann and Zuber, 2010; Sodemann and Stohl, 2009 (introducing the FLEXPART basis, and testing trajectory length and deltaq sensitivity for Antarctica); Winschall et al., 2013 (introducing the uptake time perspective).

L. 38: "the tracking of air parcels": add "the tracking of water vapour from air parcels" or something to that effect

L. 58: "if all parcels are homogeneously...": such a global initialisation as used in FLEXPART is just one way to initialize trajectories, one can just as well release particles from just a column or from a regular grid over a specified region

L. 65: "discounting in a linear manner": I do not find the choice of the word "linear" entirely intuitive. There could be a clearer separation between the calculation of fractional contributions and the discounting in case of precipitation in this paragraph (see main comment #1).

L. 68: Sodemann and Stohl (2009) used the dq threshold of 0.1 for polar regions, and evaluated the sensitivity to trajectory length for such regions. 15 days seemed to be a lower limit here, which may be important for the results obtained in Fig. 1 for polar regions. The ABL/no ABL distinction has been topic also in Winschall et al., 2014 and in Sodemann and Stohl (2009).

L. 80: I would express this a bit more nuanced, in that Sodemann (2020) propose to consider the lifetime distribution, and highlight that the long lifetimes that are part of the mean of the distribution are beyond reach or highly uncertain for Lagrangian diagnostics. In addition, the highly skewed lifetime distribution is probably more appropriately described by its median (as is commonly done for highly skewed distributions). See also the recently published review paper by Gimeno et al., 2021.

Figure 1: I like Figure 1 in that it clarifies the flow of the analysis. Could it be possible to add information on the different forms of uncertainty entering the diagnostic, such as from the trajectory calculations, the detection, the attribution etc., that then add to total uncertainty?

In Eq. (2), A needs to be defined, and something be said about A and m are determined in the analysis shown in the results part.

L. 182: I do not understand why a distinction is made between $f_z$ over land and ocean, there may be a misunderstanding of the relevant passage in S08, but if $f_z$ has been applied, an $f_z = 1.5$ has always been applied uniformly over land and ocean.

Eq. (7): The maximum RH change criterion is not yet obvious to me. You state that "large RH changes are typically associated with ABL growth and warming": why is that inconsistent with evaporation?

L. 260: How dependent are your verification results on the chosen thresholds?

L. 269: This reads as if Sodemann (2020) said that 15 days is a proxy for the globally averaged maximum lifetime. I may have overlooked it, but I do not find this statement in the cited paper.

Sec. 2.3.1: I believe this description would be clearer by separating into the fractional accounting of the arrival precipitation, and into the discounting due to precipitation en route (major comment #1).

Sec. 2.3.2: The reasoning in this section is hard to follow. Could you clearer lay out the idea behind the random attribution, and contrast to the idea of the well-mixed air parcel (not well-mixed atmosphere) in the "linear" (or fractional/sequential/well-mixed) accounting? Maybe an example would also help.

L. 346-365: I find the question of bias correction for evaporation quite intriguing. If studies indicate an overestimation, this would cause too local sources (due to overly large contributions at each time step). Potentially it would make sense to mention this already here? Note that such bias correction as applied here is only possible for global-scale studies, at least local studies suffer from the fact that only the share of evaporation contributing to a certain region is diagnosed.

L. 386: Reanalysis data include humidity perturbations from data assimilation (Läderach and Sodemann, 2016), which are another source of uncertainty of these diagnostics, and one of the motivations for using a (large enough) threshold value for humidity changes.

L. 435: The verification is done using your newly introduced additional thresholds. Here a clearer separation from the verification framework introduced just before will be useful.

Fig. 2: How do the results in Fig. 2 compare to the same kind of evaluation for the S08 method?

L. 449: This paragraph starts with the conclusion, before presenting the facts. Consider reverting the order of the paragraph.

L. 462: This statement seems to conflict with the statement in L. 449.

Fig. 3: I believe the regional results here are obtained with fixed dq thresholds. To what extent do the findings argue for the need to adapt the method to a specific study region?

L. 497: Typo in "heat"

Fig. 4, 7, 8 and similar: A more distinct colour bar, with a clearer separation from white will print better. Consider using less colour categories to allow reading off numbers/categories.

Fig. 4, lower row: these graphs are almost identical. Do you have an explanation why the source correction is overriding the diagnostics so strongly?

Fig. 6: This figure may be more informative as a table, maybe with the addition of numbers for the bias corrected results.

L. 532: Change to "There are ..."

L. 540: I am not used to the term "recycling" for heat, is this a well established expression?

L. 572: In what sense do you find the similarity of the source region maps reassuring?

Fig. 10: The colors are very similar and do not print well on all printers. Consider using patterns or a white/light region in the middle segment.

Sec. 4 (Discussion): This section needs a clearer distinction between the part of the study dealing with a verification framework, and with a modified accounting method. Consider combining the Discussion with the Conclusion section, which is now rather a summary of the study, similar to the abstract. You could also list the main findings again as bullet points to facilitate grasping the take-away messages for the reader.

L. 655: This seems to fit better to the conclusions than the discussions (or could re-appear in the conclusions)

L. 677: I think Sodemann (2020) does not claim that the discrepancies is entirely an issue of definition, see the comment to L. 80 above.

L. 707: Given the lack of a real theoretical basis for the random accounting, I would formulate this conclusion more carefully. There is certainly uncertainty in the accounting, but how large the uncertainty stemming from the accouting is in relation to the overestimation of evaporation is not finally answered from your study - and deserves further investigation.

**Supplemental material**

Sec. 4: I could imagine this section to better be placed in the main manuscript (see major comment #1).

L. 123: "Contrary to Sodemann...": It is not entirely clear what you consider to be the sources of the ABL uptakes, if not convective detrainment of BL air into the free troposphere, and on what basis you make your argument here. A more direct reference for the cited statement is Winschall et al., (2014).

**References**

Gimeno, L., Eiras-Barca, J., Durán-Quesada, A.M., Dominguez. F., van der Ent, R., Sodemann, H., Sánchez-Murillo, R., Nieto, R. and Kirchner, J. W.: The residence time of water vapour in the atmosphere. Nat. Rev. Earth Environ., https://doi.org/10.1038/s43017-021-00181-9, 2021.

Winschall, A., Sodemann, H.,Pfahl, S. and Wernli, H., 2014: How important is intensified evaporation for Mediterranean precipitation extremes?, J. Geophys. Res., 119: 5240–5256, doi:10.1002/2013JD021175

Sodemann, H.and Zubler, E., 2010: Seasonality and inter-annual variability of the moisture sources for Alpine precipitation during 1995-2002, Int. J. Climatol., 30: 947-961, doi:10.1002/joc.1932.

Sodemann, H., and Stohl, A., 2009: Asymmetries in the moisture origin of Antarctic precipitation, Geophys. Res. Lett., 36, L22803, doi:10.1029/2009GL040242.

---

## Author Response (AR1)

Ghent, 22nd of December 2021

Dear Dr. Unterstrasser,

Thank you for handling our manuscript entitled "A holistic framework to estimate the origins of atmospheric moisture and heat using a Lagrangian model" (gmd-2021-180).

Enclosed in this letter, you may find a revised version of the manuscript and a copy of the response to all reviewer comments — **updates in our response and changes in the manuscript are indicated by bold blue fonts** — along with a markup version of the manuscript.

We have addressed all comments raised by all three reviewers.

In particular, we have:

- revised the entire methods section (as requested by all reviewers),
- added textual and visual examples to highlight the differences between attribution methodologies and visualize notations (Fig. 2 and Fig. S2 — as requested by all reviewers),
- revised the random attribution methodology and discuss it in the context of the well-mixed assumption (as requested by reviewer #3),
- modified Fig. 1 to better illustrate the workflow (as requested by reviewer #3),
- and revised the writing of the entire manuscript (as requested by all reviewers).

Further, minor changes to the text and the figures, as requested by all reviewers, were performed.

We hope that these revisions make the manuscript eligible for publication in GMD.

Kind regards,

Jessica Keune (on behalf of both co-authors)

Review of paper

**A holistic framework to estimate the origins of atmospheric moisture and heat using a Lagrangian model**

by J. Keune et al.

submitted to *Geosci. Model Dev.*

This is a much-needed study contributing to quantitatively assess the reliability of Lagrangian source diagnostics. As pointed out by the authors, these diagnostics potentially provide very valuable insight into the atmospheric moisture and heat budgets; however, it is intrinsically difficult to quantify errors and uncertainties associated with these methods. I therefore fully support the intention of this study, and to a large degree also the used methodologies; however, in its current version the paper is difficult to read. I find the notation confusing in several (important) places and I could not understand the idea and implementation of the "random attribution method". Therefore, major revisions are required to streamline the paper, clarify concepts and notation, and make the paper in the end more reader friendly. It will then be a valuable contribution to the field.

We thank the reviewer for their thoughtful evaluation and support of our study. We will reply to all comments in detail in the following, **highlighting changes in the manuscript**. In particular, we understand the confusion with the random attribution method and some notations. **We revised some of our notations, and now provide figures that illustrate the concepts and notations (Fig. 2), and provide idealized examples in the main manuscript and the supplementary material** that underlie our statements with numbers.

**Major comments**

A) Line 89: "the application of these models and tools to assess diabatic heating and heat transport lags behind". I am not sure that I agree with this statement. The study by Pfahl et al. (2015) is an important one but certainly not the first one in this direction. Early applications of trajectory computations with reanalysis data in the 1980ies and 1990ies looked at latent heating in cyclones and warm conveyor belts, and how this latent heating affects the potential vorticity structure of the systems (e.g., Whitaker et al., 1988; Reed et al., 1992; Wernli and Davies, 1997; Rossa et al., 2000). These were not yet full budget studies, but I would claim that Lagrangian methods first looked at latent heating and only about 1-2 decades later also at moisture sources and transport.

References:

Reed, R. J., Stoelinga, M. T., Kuo, Y.-H., 1992: A model-aided study of the origin and evolution of the anomalously high potential vorticity in the inner region of a rapidly deepening marine cyclone. Mon. Weather Rev., 120, 893–913.

Rossa, A. M., H. Wernli, and H. C. Davies, 2000. Growth and decay of an extratropical cyclone's PV-tower. Meteorol. Atmos. Phys., 73, 139-156.

Whitaker, J. S., Uccellini, L. W., Brill, K. F., 1988: A model-based diagnostic study of the rapid development phase of the President's Day cyclone. Mon. Weather Rev., 116, 2337–2365.

We thank the reviewer for noticing insight and the references. **We changed 'adiabatic heating' to 'sensible heating'** to better reflect what we meant: while many Lagrangian studies investigate the history of moisture in the air, very few studies have focused on the history of heat — or dry static energy; and even fewer studies have outlined regions where the air was warmed by sensible heating. To better highlight that the concept of tracking latent heating was already applied in earlier studies, **we added an additional sentence on the latent heating of air during the development of cyclones**. However, we also wish to note that the cited statement referred to the number of models and studies published, rather than the temporal occurrence of studies and models to estimate the origins of moisture and heat. Therefore, **we revised the sentence** to make that clear.

B) Line94:I think this is a slightly misleading summary of the Quinting and Reeder (1997) study. They mainly emphasized the role of adiabatic descent, and their last sentence of the abstract says "Likewise, the role of the local surface sensible heat fluxes is deemphasized.".

We thank the reviewer for his comment. **We rephrased this sentence to avoid misinterpretations but wish to highlight that the cited sentence refers to the 'local' impacts. The paper does indeed highlight the role of 'remote' diabatic heating of air masses that are then transported to the heatwave region.**

C) I am completely lost with understanding the "random attribution" method (section 2.3.2) for several reasons. First, I don't understand the notation "length nt": is this n times t? And then later, what is ix, nx, ... an later n_min ... ?? Most likely this requires a schematic where you explain also the notation. Then how can you use Delta q,random in step 1 if you calculate it only in step 3? Then I am completely lost with step 2, and I also don't understand the general motivation for doing this. Can you explain this method and the motivation for it in a much better way?

We agree with the reviewer that a bit more work is needed here. **We polished this section, added a figure that visualizes the concepts (and notations) and now provide examples for both attribution methods, the random attribution and the linear discounting/attribution, in the main manuscript and the supplementary material.**

D) Fig.3b is a key figure of this study. Since,e.g.,the methods SOD08 and RH-20 vary in multiple ways (additional RH criterion, different Delta q threshold) it would be interesting to know which change had the largest effect. It would be very useful to have a more in-depth discussion of which criteria affect the results shown in Fig. 3b.

We wish to highlight that the RH-20 criterion does not explicitly consider a minimum Delta(q) threshold and that the SOD08 criterion does not consider a maximum Delta(RH) threshold (see Table 1). Hence, a direct comparison on the effects of temperature and specific humidity changes on these criteria is difficult. However, the criterion ALL-ABL was introduced to provide a criterion that lies in between SOD08 and RH-20: compared to SOD08, it does not consider a minimum Delta(q) threshold and hence indicates if filtering for a minimum threshold improves the detection of E. Compared to RH-20, the ALL-ABL does not consider a maximum Delta(RH) threshold and hence allows to infer the suitability of this temperature-dependent threshold. **We are now highlighting these in the text.**

E) While I agree that this study addresses important technical aspects of moisture and heat source identification, the text is rather heavy to read, and the results are mainly presented in a statistical way, which is hiding a bit what is going on technically. To me, it would be useful to have a didactic example, starting with a single trajectory and then a set of trajectories,

which helps me better understand the differences between the methods and the effects of the bias corrections etc.

**We have added a visualization and a description of an idealized example to the main text (Fig. 2); further (idealized) examples were added to the supplementary material (Fig. S2). We understand that further realistic examples may be needed to fully grasp the difference; however, it remains very difficult to discuss these in the context of this paper. We hope that the reviewer understands these limitations and hope that our visualizations and textual additions help to clarify remaining issues.**

F) The random attribution method has an important effect on estimating the transport time between uptake and rainout (Fig. 9c). With the random attribution method, you have much more "old uptakes" and therefore you have more long-range transport and remote sources (Fig. 10). This is very interesting and most likely an important result of this study (see also your discussion in lines 672-682). My problem is just that I didn't understand the random method (see my point C above) and that I don't find physical reasons in the paper why the random method has these effects compared to the linear attribution method. Again (see my point E above), a case study with a few trajectories might be very helpful for explaining what is going on.

**We have added a figure that illustrates the notation and helps to visualize the concept of the random attribution — further comparing it to the linear discounting/attribution. In addition, we revised the discussion of these attribution methodologies and highlight their meaning: while the linear discounting/attribution resembles perfectly-mixed conditions, the random attribution allows for deviations from well-mixed conditions (but on average resembles these).**

G) How sensitive are your main conclusions with respect to the total number of parcels calculated with FLEXPART? I don't ask you to redo a certain analysis with more parcels (this might be too time consuming), but it might be interesting to look at the effects of reducing the number of parcels.

We thank the reviewer for this question, which is indeed interesting. It is, unfortunately, true that an additional analysis with more parcels would be very time consuming. An (artificial) reduction of parcels would be feasible, but we do not really see the benefit of doing that for two reasons. First, we believe that the current setup with 2 million parcels globally represents a reasonable minimum number, as it approximates on average 30 parcels per 1°x1° grid cell and resembles at least ~half of the vertical layers from the driving reanalysis (61 layers in ERA-Interim). Considering the vertical distribution of these parcels, less than half of them remain in the ABL and for our evaluation. Further, this setup with 2 million parcels globally remains a common setup used in other studies recently published (e.g., Algarra et al., 2020; Braz et al., 2021; Drumond et al., 2019; Nieto et al., 2019; Vicente-Serrano et al., 2018) and hence provides a state-of-the-art reference. Second, technically, we are already reducing the number of parcels that we are evaluating, e.g., through a minimum threshold of Delta(q) in the SOD08 criterion or through a maximum threshold of Delta(RH) in the RH-20 criterion for the detection of E. Hence, these criteria mimic an (artificial) reduction of the number of parcels. There are, nonetheless, other studies that investigated the impact of the number of parcels on the uncertainty of moisture source regions (e.g. Tuinenburg and Staal, 2020) and could be used as a rough indication. We note, however, that the approach in those studies is different.

Finally, we wish to highlight that our study evaluates the uncertainty inherent in the *evaluation* of Lagrangian simulations. By changing the number of parcels that are being tracked, we

would assess additional uncertainties arising from the simulations directly — which could and should entail other uncertainties, e.g. arising from the convection scheme (see Sodemann, 2020) as well. We hope that the reviewer agrees that this is beyond the scope of this study.

**Minor comments**

1) Line 18: "synergistic impacts" on what? And what is meant by "a cohesive assessment", maybe "coherent assessment"?

What we mean is: without the bias correction, the approaches presented in this study yield large uncertainties. However, this uncertainty is significantly reduced if (source- and sink-) bias-correction is employed. **We revised this sentence.**

2) Line 28: here reference to Sodemann et al. (2008) would be more appropriate than Sprenger and Wernli (2015).

Thanks. The referencing here was meant to encompass the broad range of models that exist. **We now also include a reference to Sodemann et al. (2008).**

3) Line 33 and in other places: I think references should be listed in chronological order.

According to the GMD guidelines, "the order can be based on relevance, as well as chronological or alphabetical listing, depending on the author's preference" (https://www.geoscientific-model-development.net/submission.html). **We updated some of the reference lists to chronological order (where we believe that a chronological order was more appropriate).**

4) Lines 36-74: I appreciate this nice summary of Lagrangian approaches to identify moisture sources. What may be missing is a remark that Lagrangian approaches suffer from accuracy errors of trajectory computations, which can be substantial for trajectory integrations over several days. These errors stem from limitations of the numerical schemes, and most likely more substantially from the limited temporal resolution of wind fields available for offline trajectory computation.

We thank the reviewer for this remark. It is indeed true that errors in the trajectory computation are another source of uncertainty for such analyses. **We revised this sentence to better highlight additional uncertainties.**

5) Line 88: "A myriad" seems a bit exaggerated.

**We substituted 'a myriad' with 'a multitude'.** However, we have noticed the recent development of additional models and tools to track moisture and estimate the origins of precipitation. Just to name a few examples (which are also cited in our introduction): Tuinenburg and Staal (2020) just developed a new Lagrangian model that tracks moisture (UTrack) — but that is not set up to track heat (yet). The Water Accounting Model (WAM-2layers, van der Ent et al., 2014) is the base for many moisture tracking studies — but remains restricted to water. And 2L-DRM (Dominguez et al., 2020) and WRF-WVT (Insua-Costa and Miguez-Macho, 2018) are just two more examples of models that have recently been developed for the purpose of tracking moisture — and that are not readily available for the tracking of heat. And this development of models is in addition to the analytical tools already available (see introduction). Hence, we believe that the statement is not exaggerated.

6) Line 98: maybe this summary of recent Lagrangian heat wave studies should also include the one by Zschenderlein et al. (2019): Zschenderlein, P., A. H. Fink, S. Pfahl, and H. Wernli, 2019. Processes determining heat waves across different European climates. Quart. J. Roy. Meteorol. Soc., 145, 2973–2989.

We thank the reviewer for the reference. **We added a sentence and the reference to the text.**

7) Line 159: I think that the notation Delta_q(t0 – t-1) is not ideal. Delta_q does not so much depend on the time difference but rather on the two times themselves. I therefore suggest that Delta_q(t0; t-1) would be more appropriate, or maybe even Delta_q(t-1; t0).

We wanted to emphasize the direction of this 'backward time axis' — as similar analyses could be performed in a forward manner. Thus, the sign of this difference is important here. However, **we revised our notation following the reviewer's suggestion to Delta_q(t0; t-1).**

8) Line 174 and elsewhere: units should not be in italics.

We thank the reviewer for noticing. **We adjusted the format of units in equations in the revised version.**

9) Line 181: I was first confused and thought that $z$ is a function of $m$, but your m is the unit of z. I don't think that you need to mention units in the text, or you write "$z$ (in m)".

We wish to mention units in the text for the sake of completeness, but **we rephrased some sentences where appropriate.**

10) Line 190: either "applied" or "used"

Thank you for noticing. **We deleted 'used'.**

11) Line 198: is Delta q_i the absolute change? I assume that Delta q_i is negative if mixing with free tropospheric air occurs and then the Delta q_i condition is trivially fulfilled and does not help to exclude mixing with dry tropospheric air. Please clarify.

We thank the reviewer for noticing this mistake: there is an 'absolute' missing in this equation. The correct criteria reads as follows: abs(Delta(q) / q) < 10%. **We fixed the equation in the revised version of the manuscript.** So, while some negative Delta(q) changes are accounted for, also these are restricted. In addition, the height criterion was introduced to filter for mixing processes with tropospheric air. Hence, mixing is (at least partially) excluded.

12) Lines 213/215: I am not sure that I understand these RH criteria. Evaporation is particularly intense for dry air, and so why shouldn't intense (ocean) evaporation not lead to a strong increase in RH? And for the heat flux H, I assume that H leads to warming and therefore to a reduction of RH, so Delta RH should be negative, meaning that the criterion Delta RH < 10% is trivially fulfilled(?). Please clarifiy.

In essence, we argue as follows:

1) The absolute change in RH is not only dependent on evaporation, but is further determined by the temperature (change).

2) Large relative humidity changes are often indicative of mixing processes with free tropospheric air — which we wish to filter out. This is especially true for the detection of H using RH changes.

3) The RH criteria were designed to complement the existing criteria as a means to gauge the uncertainty arising from these detection criteria.

We will elaborate on each aspect below.

First, in general, we agree with the reviewer that intense evaporation can lead to a strong increase in humidity in an air parcel. However, it needs to be emphasized that relative humidity is a function of specific humidity and temperature; the (absolute) relative humidity change in an air parcel is thus a result of both specific humidity changes (e.g., through evaporation) and temperature changes (e.g., through heating from the land surface). Further, both changes are subject to the time step employed — 6 hours in our case. Due to the strong diurnal cycle of all variables affecting RH (e.g., E, H, T; see e.g. Betts and Tawfik, 2016), we often encounter that a moistening of an air parcel is accompanied by warming — which counteracts the relative humidity increase through the specific humidity increase. Further feedback processes, such as the growth of the ABL via heating from the land surface (e.g., van Heerwaarden et al., 2009; Huang et al., 2011), need to be taken into account and affect the relative humidity.

Second, the detection criteria were chosen as a means to ensure that the sampled air parcel's evolution of humidity and (potential) temperature is mainly indicative of ABL processes. Strong changes in RH are often the consequence of additional processes, notably entraining/detraining air parcels mixing with ambient air (and thereby typically experiencing strong specific humidity in/decreases, and potential temperature de-/increases). For the detection of H, it is indeed the case that in the absence of other processes (e.g., evaporation, mixing, fog dissolution), we would expect RH to decrease. Whenever H is strong, however, this implies ABL growth and a subsequent shrinking towards the evening — then, many air parcels previously part of the ABL mix with free tropospheric air, whose potential temperature tends to be higher (and relative humidity is often markedly different). To limit the detection of such events, and thus the overestimation of H, it is useful to sample only potential temperature increases associated with a moderate (absolute) change in relative humidity.

Finally, the RH criteria were designed to complement the existing criteria. In this context, we wish to mention: there are downsides to all of the proposed criteria. From a moisture perspective, the SOD08 criteria use a minimum Delta(q), and one could also argue that small increases in Delta(q) may still be associated with evaporation and should not be neglected. Or in case of the ALL-ABL criteria, one could argue that not all humidity changes are associated with surface evaporation. With the comparison of the three criteria (SOD08, RH-20, ALL-ABL), we wanted to compare the impact of three criteria that filter for opposites: one that counts small increases (RH-20), one that only evaluates sufficiently large increases (SOD08) and one that counts just everything (ALL-ABL). Further, as pointed out in the manuscript, we employed all thresholds globally and did not calibrate any thresholds. And as our results show: the impact of those criteria on the estimation of the source regions is considerably small — if the detected source region fluxes are bias-corrected.

**We added a comment on the differences between all criteria to the manuscript.**

13) Line 235: I expect the opposite: with 6-hourly data we estimate the diurnal cycle of the ABL height poorly and therefore the ABL height criterion might be important. If we had hourly data (e.g., with ERA5) then there should be less sensitivity with respect to the design of the height criterion.

We agree with the reviewer that the accuracy of the simulated ABL heights — and the parcel heights — should get better as the temporal resolution increases. As time steps get smaller, the differences between the ABL heights become smaller as well. We assumed that it would make sense to filter 'more strictly' (i.e., require both occurrences to be within the ABL) for the detection of surface fluxes in that case, because mixing with tropospheric air could be better detected. However, we agree that this is speculative and, in the context in which it is presented, misleading. **We removed this sentence.**

14) Line 260: "for E for P" should read "for E and P".

Thank you for noticing. **We replaced 'for' with 'and'.**

15) Line 274: Strange formulation "Due to the consideration ..., mass and energy are conserved ...". I think mass and energy conservation is valid independent of what is considered by the algorithm(?).

We thank the reviewer for highlighting this sentence. **We revised this sentence.** For the sake of completeness, we wish to clarify this issue in our response here too: The Lagrangian simulations are mass and energy-conserving. However, the way the analysis of the output is conducted is not necessarily mass- and energy conserving. Consider, for example, the following trajectory that extends 7 timesteps into the past:

| Time | t-7 | t-6 | t-5 | t-4 | t-3 | t-2 | t-1 | t0 |
|---|---|---|---|---|---|---|---|---|
| Specific humidity (g kg$^{-1}$) | 1 | 2 | 5 | 2 | 3 | 4 | 5 | 1 |
| Change in specific humidity (g kg$^{-1}$) | | +1 | +3 | -3 | +1 | +1 | +1 | -4 |

Some approaches consider +4 g kg$^{-1}$ between t-7 and t-5 for the estimation of source regions of the precipitation event at t0. However, this is not mass-conserving as the moisture loss *en route* (at t-4) is not considered. And in this particular case, the parcel contained less specific humidity at t-4 than it gained before — the corresponding sources thus depict qualitative source regions but do not facilitate a quantitative assessment. Consequently, if the sinks of moisture along individual trajectories are not considered (e.g. through linear discounting and attribution), the approach is not mass-conserving.

**As mentioned before, we added a similar example to the main manuscript (see Fig. 2 and description thereof).**

16) Line 285: "time step (t)" should read "time step *t*" (italics).

Indeed, **we adjusted the sentence.**

17) Line 288: I don't see the need to introduce a new notation 1_Delta Phi ... for this function. This is the Heaviside step function, which in this case could be written as H(–Delta Phi_j).

Yes, it could. **We are now using the Heaviside function in the revised version of the manuscript.**

18) Lines 285-299: please clarify whether everything here is identical to SOD08, or whether you introduced some modifications.

This is identical to the approach introduced in Sodemann et al. (2008) — just written down differently. We decided to use our own formulation as we found the description in Sodemann et al. (2008) more difficult to follow. **We added a sentence clarifying that this is identical to Sodemann et al. (2008) but follows a different notation.**

19) Lines 300-307: I don't understand why this explained here after the linear discounting, appears a bit out of place.

This part is needed to aggregate the contribution of sources along individual trajectories to coherent source regions, as displayed in Figs. 4, 7, and 8. While it appeared logical to us to add it to this section, analogous to the upscaling of the fluxes in Equations 2–8, it is true that a similar step is needed for the random attribution. **We thus added a subsection "2.3.3 Aggregation to establish (biased) source–receptor relationships" that is equally applicable to both attribution methodologies.**

20) Line 358: I again struggle with the notation: is x here an index? If yes, why then do you write it as a superscript of S_LM? I realize that at this point of the paper I cannot really follow any further, mainly because of confusing notation. What are indices, what are coordinates, what are just subscripts/superscripts ...? Does "LM" mean "Lagrangian model" or something else?

The notation S_LM(x) was introduced earlier already (Eq. 16–17) as the source region contribution as estimated from a Lagrangian Model — the "x" here refers to the conditioning of the flux at a specific point x (in space and time). I.e. for every source grid cell only a subset of all parcels over that grid cell are evaluated using the multi-day backward trajectories. In contrast, the unconditional S_LM version evaluates all parcels over that grid cell. As we do have four coordinates (longitude, latitude, time, backward time) we were hoping to avoid an explicit referencing of those throughout the paper. The "x" thus simply refers to the conditional evaluation for a specific receptor region. **We explicitly introduce this notation now; and we added notations for the diagnosis (S_LM) correspondingly. We hope that this helps to clarify the concept.**

21) Line 497: typo in "heat"

**We removed this typo.**

22) Fig. 6: I cannot find the information how you define "local"; does this "local region" have the same size for all cities?

Yes, 'local' refers to the 3°x3° grid cells around the center of each city. This is defined in l. 408: "Unless otherwise noted, a 3°x3° box around each city center is used as a receptor area.". **We specified this in the text and repeat the information in the captions of Figs. 6 and 10.**

23) Figs. 7 and 8: I find it very difficult to see something in these many panels, except that they all look very similar. I think the smooth blue-only color bar does not help. Can you find an alternative way of visualizing the results that is more insightful for the reader and that makes the differences more apparent? Maybe by showing difference fields from a "reference setup".

It is true that these are quite similar and that differences are difficult to spot. However, we decided to show the absolute source regions for two reasons. First, it is consistent with the source region illustration of heat (Fig. 4), but displays much more similar source regions. This underlines the minor impacts of the detection criteria for the estimation of precipitation source regions compared to heat source regions. Second, we do not want to single out **one** criterion as 'the' reference. Displaying differences would make much more sense if the ground truth was known and/or a 'best' criterion could be highlighted following a validation exercise. This is, however, not the case here as observations for the latter are missing. We thus wish to refrain from showing differences here. We have published the data along with the manuscript, which allows any reader to analyze differences on their own. However, **we adjusted the color scale.**

24) Line 661: why are the new criteria better to assess global warming trends?

If the air is becoming drier under global warming, this will be reflected in the specific humidity of air parcels as simulated with, e.g., FLEXPART. A static specific humidity threshold might thus impact the assessment of trends from such models. Around l. 661, we simply wanted to note that a relative humidity threshold could be used instead and could eliminate these issues. **We removed this statement from the discussion.**

**References**

Algarra, I., Nieto, R., Ramos, A. M., Eiras-Barca, J., Trigo, R. M., and Gimeno, L. (2020). Significant increase of global anomalous moisture uptake feeding landfalling Atmospheric Rivers. Nature Communications, 11(1), 1-7.

Braz, D. F., Ambrizzi, T., Da Rocha, R. P., Algarra, I., Nieto, R., and Gimeno, L. (2021). Assessing the moisture transports associated with nocturnal low-level jets in continental South America. Frontiers in Environmental Science, 9:657764.

Betts, A. K., and Tawfik, A. B. (2016). Annual climatology of the diurnal cycle on the Canadian Prairies. Frontiers in Earth Science, 4, 1.

Dominguez, F., Hu, H., and Martinez, J. A. (2020). Two-layer dynamic recycling model (2L-DRM): learning from moisture tracking models of different complexity. Journal of Hydrometeorology, 21(1), 3-16.

Drumond, A., Stojanovic, M., Nieto, R., Vicente-Serrano, S. M., and Gimeno, L. (2019). Linking anomalous moisture transport and drought episodes in the IPCC reference regions. Bulletin of the American Meteorological Society, 100(8), 1481-1498.

Huang, J., Lee, X., and Patton, E. G. (2011). Entrainment and budgets of heat, water vapor, and carbon dioxide in a convective boundary layer driven by time-varying forcing. Journal of Geophysical Research: Atmospheres, 116(D6).

Insua-Costa, D., and Miguez-Macho, G. (2018). A new moisture tagging capability in the Weather Research and Forecasting model: formulation, validation and application to the 2014 Great Lake-effect snowstorm. Earth System Dynamics, 9(1), 167-185.

Nieto, R., Ciric, D., Vázquez, M., Liberato, M. L., and Gimeno, L. (2019). Contribution of the main moisture sources to precipitation during extreme peak precipitation months. Advances in Water Resources, 131, 103385.

Sodemann, H. (2020). Beyond turnover time: constraining the lifetime distribution of water vapor from simple and complex approaches. Journal of the Atmospheric Sciences, 77(2), 413-433.

Sodemann, H., Schwierz, C., and Wernli, H. (2008). Interannual variability of Greenland winter precipitation sources: Lagrangian moisture diagnostic and North Atlantic Oscillation influence. Journal of Geophysical Research: Atmospheres, 113(D3).

Tuinenburg, O. A. and Staal, A. (2020). Tracking the global flows of atmospheric moisture and associated uncertainties. Hydrological Earth System Sciences, 24, 2419–2435.

van der Ent, R. J., Wang-Erlandsson, L., Keys, P. W., and Savenije, H. H. G. (2014). Contrasting roles of interception and transpiration in the hydrological cycle—Part 2: Moisture recycling. Earth System Dynamics, 5(2), 471–489.

Van Heerwaarden, C. C., Vilà-Guerau de Arellano, J., Moene, A. F., and Holtslag, A. A. (2009). Interactions between dry-air entrainment, surface evaporation and convective boundary-layer development. Quarterly Journal of the Royal Meteorological Society: A journal of the Atmospheric Sciences, Applied Meteorology and Physical Oceanography, 135(642), 1277-1291.

Vicente-Serrano, S. M., Nieto, R., Gimeno, L., Azorin-Molina, C., Drumond, A., El Kenawy, A., Dominguez-Castro, F., Tomas-Burguera, M., and Peña-Gallardo, M. (2018). Recent changes of relative humidity: Regional connections with land and ocean processes. Earth System Dynamics, 9(2), 915-937.

RC2 — Anonymous Referee #2
*received and published: 23 Aug 2021*

Review of Keune et al (2021) A holistic framework to estimate the origins of atmospheric moisture and heat using a Lagrangian model.

This study addresses uncertainty in the estimation of precipitation and heat sources derived from Lagrangian parcel trajectory methods. A framework is proposed to assess uncertainty in the input quantities and the associated trajectories, based on FLEXPART and ERA-Interim reanalysis. The study presents an important contribution to the field and the authors have done well to develop a framework that assesses a complex collection of uncertainties. I support the publication of this study, after the issues outlined here are addressed.

We thank the reviewer for their appreciation and support of our study, as well as their useful comments and suggestions. Our reply to all comments is detailed below. **Bold replies highlight updates to the manuscript**.

General comments

1. I suggest it will be easier for the reader to understand the methodology if the need for such a methodology was more clearly explained in the introduction. The motivation of the study, besides estimating the uncertainty in identified sources, could be clarified by outlining the specific uncertainties you wish to examine. My understanding from reading the manuscript is that you wish to (i) evaluate the sensitivity of identified source regions to the air parcels released and (ii) the loss/gain of moisture/heat from/to those parcels along their trajectories. Is this correct? As for (i), it's unclear why this is necessary – what exactly is the issue with FLEXPART that you are trying to rectify? Is the issue that FLEXPART normally tracks all air, and you want to constrain it to only track moisture for precipitation (or heat) specifically? Why aren't the reanalysis fields of precipitation and evaporation used in the selection and tracking of parcels? And is the impact of the number and height of the parcels released, and trajectory timestep, considered? As for (ii), I'm unclear why it is necessary – could you expand on this? For instance, L53 states: "… moisture losses between source and sink regions are not accounted for." I'm a bit confused here – I thought FLEXPART intrinsically accounted for losses and gains between source and sink through the use of the positive and negative change in specific humidity along the trajectory (as stated in lines 49-50)? Is the problem that precipitation and evaporation must be inferred from the specific humidity change, and that you would like to quantify the precipitation/evaporation explicitly? This is an important point for the reader to understand the rest of the paper, including the need for linear or 'random' attribution of moisture – could you clarify please?

We agree with the reviewer that the need for the framework could be better motivated. **We revised our introduction and explicitly declare two objectives that are going to be tackled in the study. Further, we improved the comprehensibility of the methods through revisions of the text, as well as through the addition of figures and examples to the main text and the supplementary material.**

Here, we also clarify a few outstanding issues:

In general, the reviewer is absolutely right when they question "*Is the problem that precipitation and evaporation must be inferred from the specific humidity change, and that*

*you would like to quantify the precipitation/evaporation explicitly?*". This is exactly the problem for the estimation of source regions of precipitation/moisture (addressing issue (ii)). While the simulations are driven with reanalysis data, including the surface fluxes, data is only available at the grid cell level and needs to be interpolated to the projected parcel locations. At this spatio-temporal resolution, only the specific humidity is available. While the sign of the specific humidity change is indeed a hint at the dominant process taking place (i.e., a gain or a loss of moisture through evaporation or precipitation; see Eq. 1), this criterion only reflects large scale evaporation and precipitation if integrated over sufficient large areas and longer times scales — which is also why a selection of parcels based on reanalysis fields of E and P is not straightforward. Integrating these specific humidity changes over large spatio-temporal scales and along multi-day backward trajectories, however, yields only a qualitative description of the source regions. The resulting source region maps show large areas of both positive (E–P > 0) and negative (E–P < 0) regions, with the former indicating general source regions and the latter general sink regions (e.g., between a source and the receptor region). In this setup, it remains difficult to *quantify* how much a specific source region contributes to, e.g., a precipitation event in the receptor region. Or in other words: the general sink regions between a source region and the precipitation event are not accounted for (l. 53). Hence, if one wants to detect those processes at smaller spatio-temporal scales and in a quantitative manner, additional criteria are needed.

As for (i): no, we do not evaluate the sensitivity of the source regions to the air parcels released. Our simulation is a global simulation that is initialized with a globally homogeneous distribution of 2 million air parcels. These move with the winds in space and time and can be used to establish multi-day backward trajectories, that in turn can be used to estimate source regions. However, the latter sounds rather simple, but it is not: first, criteria for a (reliable and accurate) detection of surface fluxes have to be determined; and subsequently, the quantitative source region contributions have to be estimated. Both steps are subject to uncertainty: in Fig. 2 (and 3), we show that it remains difficult to estimate the fluxes with the presented criteria — in most cases, considerable biases remain. However, these biases can be corrected; e.g., (re-)using the reanalysis data set that was used to force the simulations in the first place. And this is exactly what we do.

This framework for assessing uncertainty really only relates to FLEXPART-type studies. This is fine, but it needs to be discussed somewhere. For example, could you comment on how the framework might be applied to other types of models? This would make the proposed framework more widely applicable.

We thank the reviewer for this broad perspective. This framework could be applied to all models that trace air parcels — and FLEXPART is just one of them. LAGRANTO is another example, to which the framework could be directly applied. Further, parts of the framework (e.g., the different attribution methodologies) could also be applied to other models that trace 'water parcels'. **We added a few sentences to the discussion that highlight the general applicability of the framework.**

2.  The need for and steps involved in the random attribution of moisture needs further clarification. I find the explanation hard to follow, and I'm a bit lost in matching the notation in the 3 steps to the rest of the text. Could you clarify the general idea of the approach, and each of the steps involved? This relates to the first comment above, that the need for such attribution needs further explanation.

We understand and agree with the reviewer, and all other reviewers, that the random attribution needs to be better explained. The general idea behind the random attribution is as

follows: we determine a physical limit for the maximum contribution of a source location to the precipitation event under consideration. Consider, for example, a parcel that gains 5 g kg$^{-1}$ of moisture through surface evaporation, but the parcel's total specific humidity content reduces to 2 g kg$^{-1}$ through phase changes and/or precipitation *en route* afterwards —then the maximum contribution of this particular source location to the precipitation event is 2 g kg$^{-1}$. These limits are set for all identified source locations; and an iterative procedure then distributes the precipitation loss to all identified source locations. The distribution happens in two steps, which are randomly determined: First, a 'random' location among the identified ones is drawn. Second, a 'random' contribution between 0 g kg$^{-1}$ and the maximum one for that source location is drawn. The procedure is repeated until the entire precipitation amount is attributed to all source locations.

**We have revised everything relating to the random attribution: we motivate it better (allowing for deviations from the perfectly-mixed assumption embedded in the linear discounting/attribution), we revised the description and notation, we provide an example and compare it to the linear discounting/attribution, and we discuss it in the context of well-mixed assumptions.**

3. The results figures are clear and well thought-out, but the meaning and implications of the numerical results could be further drawn out. For Figures 3 and onwards, what are the implications of these statistical results, both physically and for future studies? Furthermore, there is a clear difference in results that are based on the two attribution methodologies - but how can the reader assess if either are realistic or even necessary?

We thank the reviewer for this excellent question. We wish we had a straightforward answer. Unless there are observations available that facilitate a validation of the source–sink relationship, we do not know the ground truth and we cannot assess if the presented source regions are realistic.

This lack of observations, however, motivates the need for this uncertainty assessment. Thus, while we cannot assess the realism of the presented approaches, we can show that their assumptions do have an impact on the results. By presenting this uncertainty, we want to make the community aware of it, and we wish to encourage future studies to communicate these uncertainties.

**We revised large parts of the introduction and the manuscript in general — and in particular added a physical meaning to the attribution methods. We hope that these revisions better explain the meaning and the implications of our results.**

Minor comments

5. L18: I'm not sure what is meant by 'synergistic impacts'. Do you simply mean that the bias corrections reduce the identified uncertainties?

Yes, this is exactly what we mean. **We revised this sentence/the wording.**

6. L107 and 111 and other places in the document: I'm not sure what is meant by a 'diagnosis' of surface fluxes. Are you referring to an evaluation between simulated and observed fluxes, or something else?

Yes, by diagnosis we mean the (unconditional) detection and quantification of surface fluxes and precipitation from the Lagrangian model. To detect and diagnose these fluxes, the difference between two consecutive timesteps are considered for all air parcels — and evaluated using one of the criteria detailed in the methods section. The integration of all parcel differences then represents the 'diagnosed' flux on a global grid — and can be validated with observations and/or reference data sets, such as ERA-Interim. **We added an illustrative sketch to Fig. 1 and revised the text on many places, explicitly mentioning the two-step trajectories.**

L131: 'all air parcels … are evaluated independently…' – what is the aim here?

The aim is to detect how well the criteria detect the fluxes; i.e. to estimate the (biased) fluxes from the model and to determine bias correction factors. The 'independently' refers to the fact that we do not condition this on any receptor region but apply it globally using two consecutive timesteps. **Analogously to our previous reply: we revised the description and Fig. 1 to better describe this diagnosis step.**

7. L113: Which 'other existing methods' are you referring to? Could you cite some examples please?

**We added references to this sentence.**

8. L140: '…source contributions can be further constrained by means of a receptor quantity…'. Do you mean source contributions can be scaled to match the precipitation in the sink?

Yes, and no… This is exactly what can be done and is done in many cases (and a bias correction with precipitation yields the same result). With this sentence, we wanted to generalize this meaning a bit as also other receptor quantities could be used; e.g., the integrated water vapor over a region could be used as a target variable instead of precipitation. On the contrary, for heat advection as defined here, no receptor quantity can be applied and the source region contributions are not constrained by a receptor quantity. This is, however, not explicitly stated. **We revised this sentence added a few more sentences on this matter in the method description.**

9. L232: Could you expand a little on the importance of only using parcels that are within the ABL? L235 states the impact is considerably small for 6h time steps. Does this mean that back-trajectory methods don't need to consider the height of the ABL, or that it has a minor impact?

In l. 235, we were referring to the difference of the results if one considers only one or both occurrences to be within the (maximum) ABL. For our case studies and the 6-hourly timesteps, the resulting source locations did not differ much. Our speculation – i.e., that this is impacted by the 6-hourly time steps – was, however, misplaced (as also mentioned by another reviewer). **We revised this sentence and removed this speculation**.

As for the importance of considering ABL changes only: the answer to this question appears to be almost philosophical. While we wish to determine only the *direct* surface source locations, Fremme and Sodemann (2019) argue that changes above the ABL should be considered as they are indirectly influenced by the surface. In turn, other studies (e.g., Stohl and James, 2004; Stohl and James, 2005; Nieto et al., 2006; Drumond et al., 2008; and others), are more interested in the general sources of moisture and examine large-scale

convergence and divergence zones of atmospheric moisture transport; and hence do not limit themselves to surface sources only. With our interest to identify only the direct surface source locations, the ABL criterion helps to increase the likelihood that changes in state variables reflect surface fluxes.

10. L393: '…the timesteps for the calculation of trajectories are adapted to Lagrangian timescales…'. What is the trajectory timestep? How was is determined?

In FLEXPART, parcel trajectories are determined using the grid scale wind, as well as turbulent and mesoscale wind fluctuations. FLEXPART was run in with 900 seconds synchronization and sampling timesteps; which correspond to the default setting of FLEXPART (see Stohl et al., 2005). Turbulence is not well described at this timescale (Stohl et al., 2005) — and hence the timestep can be adjusted through a modification of the Langevin equation, which parameterizes turbulence in FLEXPART.

We do not wish to dive further into this aspect, thus reference to the technical description of FLEXPART in Stohl et al. (2005) for details. However, **we adjusted the corresponding sentence in the main manuscript to better reflect the simulation settings.**

11. L36: Suggest rephrase grammar to "Tracking air parcels enables the state of the atmosphere and its changes in space and time to be inferred, …"

We thank the reviewer for the suggestion, and **we revised the sentence accordingly.**

12. L156: The first sentence of section 2.2. makes it sound as though it follows from what is said above. Perhaps rephrase to something like: 'To characterize the physical processes influencing the air parcels, the changes in air parcel properties…'.

We thank the reviewer for the suggestion. **We revised this sentence along the lines of the reviewer's suggestion.**

13. L497: Change 'he3at' to 'heat'.

**We fixed this typo.**

**References**

Drumond, A., Nieto, R., Gimeno, L., and Ambrizzi, T. (2008). A Lagrangian identification of major sources of moisture over Central Brazil and La Plata Basin. *Journal of Geophysical Research: Atmospheres*, *113*(D14).

Nieto, R., Gimeno, L., and Trigo, R. M. (2006). A Lagrangian identification of major sources of Sahel moisture. *Geophysical Research Letters*, *33*(18).

Stohl, A., Forster, C., Frank, A., Seibert, P., and Wotawa, G. (2005). The Lagrangian particle dispersion model FLEXPART version 6.2. Atmospheric Chemistry and Physics, 5(9), 2461-2474.

Stohl, A., and James, P. (2004). A Lagrangian analysis of the atmospheric branch of the global water cycle. Part I: Method description, validation, and demonstration for the August 2002 flooding in central Europe. *Journal of Hydrometeorology*, *5*(4), 656-678.

Stohl, A., and James, P. (2005). A Lagrangian analysis of the atmospheric branch of the global water cycle. Part II: Moisture transports between Earth's ocean basins and river catchments. *Journal of Hydrometeorology*, *6*(6), 961-984.

RC3 — Harald Sodemann
*received and published: 27 Aug 2021*

**Review of "A holistic framework to estimate the origins of atmospheric moisture and heat using a Lagrangian model" by Keune et al., submitted to GMD**

Keune et al present a framework for the evaluation of Lagrangian methods for quantitative offline-diagnosis of heat and moisture from air-parcel trajectories. There exists quite a number of studies with similar yet different concepts and implementations, and the community is clearly in need of ways to enabe comparison and verification exercises. In this regard, the paper is clearly a needed and welcome contribution to the literature. In addition, the manuscript is well- written and most of the material clearly presented. I have a number of comments with respect to some of the literature and interpretation, and to the presentation of figure material, detailed below. Since my attention has mostly been on the moisture source identification, I mainly focus my comments on those aspects of the paper. I have no reason to conceal my identify, also because it will be quite evident from my comments that I am the main author of one of the methods assessed here.

Harald Sodemann

We are grateful for Harald Sodemann's endorsement and thoughtful comments on our work, which will help us to further improve the quality of the manuscript. We also highly appreciate his decision to forego anonymity. We reply to all comments below. **Bold replies highlight updates to our previous reply and changes in the revised version of the manuscript.**

**Major comments**

1. The description of the accounting procedure is not entirely clear or may miss one important point. I recommend to separate two aspects more distinctly, (a) considering the fractional contributions of the uptakes (source contributions) during the uptake (i.e. how much does a source contribute to what is in the air parcel at the end of the time interval), (b) discounting all previous contributions according to their relative share of all water vapour in an air parcel. Step (a) is a fundamental change from methods without accounting, that only consider the local humidity change, rather than the fractional contribution times the arrival precipitation.

We fully agree with this distinction. We tried to refer to 'linear discounting' for (b) and to 'linear attribution' for (a) — if we interpret the reviewer's definitions correctly, because we are missing the word 'losses' in the discounting procedure in the comment. In our case, we referred to 'linear discounting' as the procedure, in which one 'discounts' uptakes with losses *en route*. Subsequently, we referred to 'linear attribution' as the procedure, in which one 'attributes' how much a source contributes to a sink. Please note also, that our notation differs from the one in Sodemann et al. (2008); in particular, we prefer to think of absolute contributions (Eq. 15) instead of 'fractional contributions'. Yet, the underlying concept remains identical. However, we understand from the reply above that the term 'discounting' may be used differently. Therefore, and because we also agree that this difference could be better highlighted in the manuscript, **we revised parts of this section in the new version of the manuscript.** If the reviewer believes that those changes are insufficient, and that the terminology may still be confusing, we remain open to consider other alternatives.

2. It should be mentioned somewhere that there is a physical/theoretical basis for the assumption that all sources contribute to precipitation en route and at the arrival point according to their share of the total water vapour in an air parcel, namely the assumption of well-mixed conditions within an airparcel within a 6-h time interval. The

random accounting procedure that is presented in Sec. 2.3.2 does not have such a theoretical basis. Other than being a sensitivity test, it is unclear how reliable/meaningful the results obtained with such a random attribution approach are in terms of physical interpretation.

We thank the reviewer for highlighting this; **we added a few sentences to better highlight the well-mixed assumption in the revised version of the manuscript.**

Linear discounting and linear attribution follow the assumption that parcels are perfectly mixed; in which all sources *always* contribute with their *exact* share of specific humidity in the air parcel prior to the precipitation event. However, we believe that the random attribution also follows this same well-mixed assumption — at least on average. By construction, there may be deviations from the 'perfect' well-mixed situation, but these average out over many parcels (and long time scales).

**We are grateful for this comment, which inspired us to dig a bit deeper and understand the implications of the random attribution for well-mixed assumptions. The findings inspired by this comment hence helped us to improve the random attribution methodology. The results in the manuscript are updated accordingly — but it is noted that they show only small differences to the initial version of the random attribution. Nevertheless, we believe that the improvements enhance the credibility the approach; and hence included them in the revised version of the manuscript. For the sake of completeness, we briefly describe our findings below.**

**In particular, we analyzed to which degree our statement — that the random attribution follows a well-mixed assumption on average — holds. Therefore, we constructed a set of idealized trajectories and evaluated them 10.000 times with the random attribution methodology; and compared the results to the linear discounting and linear attribution that represents perfectly-mixed conditions. Three selected trajectories and their (average) attribution for all attribution methods are shown in Figs. R1–R3. The black line shows the reference (linear discounting+attribution) and represent perfectly-mixed conditions. The green line and shading show the average and the interquartile-range of 10.000 evaluations of the initial random attribution, respectively. The blue line and shading represent the attribution following an updated version of the random attribution.**

**Figs. R1–R3 show that the average attribution of the initial version of the random approach reproduces the well-mixed conditions for most of the idealized trajectories. However, in case of exponential increases of moisture (idealized example 2), the uniform sampling of uptake locations leads to a divergence from the well-mixed assumption (green line). This effect is strongest for small precipitation amounts (idealized example 2) but becomes negligible for relatively large events (idealized example 3). Trajectories with rain *en route* are typically more constrained and thus show lesser deviations from the well-mixed assumption (not shown). To correct for this deviation from the well-mixed assumption, we introduced a weighting of source/uptake locations in the sampling, that is based on the relative moisture gain (and thus approaches the linear attribution method). The results of this adjusted sampling better follow the well-mixed assumption (blue lines in Figs. R2–R3).**

[Figure]

**Fig. R1.** *Idealized trajectory #1 (equal moisture uptakes; small precipitation event).*

[Figure]

**Fig. R2.** *Idealized trajectory #2 (exponential increase of moisture uptakes; small precipitation event).*

[Figure]

**Fig. R3.** *Idealized trajectory #3 (exponential increase of moisture uptakes; large precipitation event).*

As indicated above, we updated the manuscript with results from the new random attribution methodology. Nevertheless, we wish to emphasize that the impact of this update is comparably small: Fig. R4 shows the relative contributions of both random approaches compared to the perfectly-mixed approach embedded in the linear discounting and attribution methodology — averaged over all trajectories for Denver, Beijing and Windhoek and the full climatology (1980–2016). The weighted sampling in the random selection of source locations leads to a small shift towards nearby source locations (compare blue and red bars in Fig. R4); but differences between linear and random attribution remain superior. A comparison of local recycling ratios (Fig. R5) for all three cities confirms this finding.

[Figure]

**Fig. R4.** *Relative (bars) and cumulative (lines) backward day contributions to precipitation for the three attribution methods, using the ALL-ABL approach. Contributions are averaged over all three cities and the period 1980–2016 (analogous to Fig. 9 of the main manuscript).*

[Figure]

**Fig. R5.** *Origins of precipitation subdivided into local (orange), other land (green) and ocean (blue) source regions for (a) Denver, (b) Beijing and (c) Windhoek, averaged over the period 1980–2016 – using the 'ALL-ABL' criterion and the different attribution methodologies. 'Random' refers to the initial version of the random attribution methodology; and 'Random2' refers to the new version.*

3. Such Lagrangian offline diagnostics as discussed here will always be imperfect approximations of how water vapour moves in a model simulation. What is your take on the question, what level of accuracy we actually can expect from such methods?

We thank the reviewer for this question. First of all, we wish to clarify: our framework builds up on (offline) Lagrangian simulations, and we 'only' evaluate the uncertainty inherent in the evaluation of these trajectories. As such, our work is limited to the accuracy of these simulations; i.e. our results depend on the accuracy of the trajectories, but also on the number of parcels that are being tracked and the time step and spatial resolution of the reanalysis data that is used to force the simulations. **We are now mentioning this dependency explicitly in the revised version of the manuscript.**

Further, we want to elaborate on the reviewer's question: we agree that these simulations will always remain imperfect — inaccuracies stem from a lot of sources, such as the spatio-temporal resolution of the driving reanalysis, the reanalysis itself, the number of parcels that

are being tracked, and numerical errors in the interpolation scheme, just to name a few. However, at this point, and especially due to the sparsity of measurements to validate these simulations, it remains difficult to assess how (in-)accurate these simulations are. It is true, however, that the inaccuracy can be expected to increase with increasing trajectory lengths as small errors add up. As a result, we refrain from analysing trajectories longer than 15 days; and we would not like to rely on single trajectories. For the estimation of source regions from these trajectories, however, we expect that average source regions over many trajectories and long time scales) are reliably detected as we expect some of the inaccuracies to average out.

From our perspective, there are a few ways forward to improve the accuracy of these trajectories and the resulting source region estimations. First, we believe that the time step of the driving reanalysis is critical too: we have to assume that processes such as E and P take place at the midpoint between two locations and time steps. For 6-hourly (and even 3-hourly) time steps, as employed here, this presents a large uncertainty — that could add to the numerical errors from the simulations. Other studies, such as Tuinenburg and Staal (2020) further show that the vertical structure of Lagrangian simulations can have a large influence on the recycling ratios; and the reviewer's paper (Sodemann, 2020) also shows that assumptions about vertical mixing and the employed convection scheme come along with uncertainties. Consequently, we expect that a higher spatio-temporal resolution of the driving forcing (e.g., using ERA5) and the tracking of more parcels can improve the accuracy of these (offline) trajectories, and reduce parts of the uncertainty of the source region estimation. Nevertheless, we highly encourage and support intercomparisons with other models to gauge the uncertainty inherent in such simulations.

4. The term "holistic" in the title has in my perception connotations that are not well covered by what the proposed framework actually encompasses (being valid for heat and moisture specifically, rather than "everything" as holistic could imply). How about replacing with a more limited word, such as "unified" or "generic"?

The term 'holistic' referred to the merge of tracking heat and moisture; also, to the fact that we introduce a workflow that encompasses all steps, from the unconditional detection of fluxes to the bias-correction, along with possible variations in all of them. However, this comment is in line with another reviewer's comment, and **we replaced 'holistic' with 'unified' in the title and the text.**

5. The paper currently seems to introduce both a verification framework, and a modified source accounting algorithm with additional parameters. A clearer statement of this dual objective, and a potentially clearer separation of both aspects in the manuscript (method/results) could be beneficial to avoid confusion with the reader about the focus and intent of the paper.

We wish to highlight uncertainties along all steps of the evaluation workflow, which leaves us with a few more aspects that require separation. To keep the manuscript short and readable, we highlighted only the steps that introduce the largest uncertainty. However, we understand that a bit more explanation is needed to make the paper accessible to a broader audience. Therefore, and in line with comments from another reviewer, **we updated the introduction and now specify our objectives better; we modified Fig. 1 to highlight the difference between the verification part ('diagnosis') and the source region estimation, and we revised the structure of the methods section.**

**Detailed comments**

L. 35: "while others trace air parcels and their properties": I first misread this to comprise also the accounting-type methods, such as S08, but then two paragraphs later understood, how you build up the story. Maybe it can be made more clear how you distinguish the different aspects, and still include S08 in the list of references in L28?

Our introduction follows the following structure: we first introduce models (e.g., Eulerian, Lagrangian), and then explain differences between Lagrangian approaches (water vs. air parcels) — and in l. 35, we simply mention the main references to Lagrangian models that trace air parcels. Just from there onwards, we specify the methods used to evaluate the latter — which is where we see Sodemann et al. (2008). Our references in l. 28 were not intended to be complete (and **we added 'e.g.' to this reference list**), but we also **added Sodemann et al. (2008) as a main references for Lagrangian models** too.

L. 35: FLEXPART and Lagranto trace airmass motion and interpolate boundary-field variables to the parcel position.

Indeed — FLEXPART (and Lagranto) interpolate from the large-scale air mass motion to the parcel position. This is, to our knowledge, not restricted to the atmospheric boundary layer and boundary-layer variables. Further, while this describes the technical procedure, we believe that the 'tracing air parcels' picture is an accurate visual description of the underlying idea.

L. 73: There are a few additional references that use the S08 method, that may be relevant here, including Sodemann and Zuber, 2010; Sodemann and Stohl, 2009 (introducing the FLEXPART basis, and testing trajectory length and deltaq sensitivity for Antarctica); Winschall et al., 2013 (introducing the uptake time perspective).

We thank the reviewer for providing further references, and **we incorporated some of them**.

L. 38: "the tracking of air parcels": add "the tracking of water vapour from air parcels" or something to that effect

In our understanding, the former includes the latter — which is why we refer to the 'tracking of air parcels and their properties' in most parts of the text.

L. 58: "if all parcels are homogeneously...": such a global initialisation as used in FLEXPART is just one way to initialize trajectories, one can just as well release particles from just a column or from a regular grid over a specified region

That is true. However, from our experience and understanding, most (global) FLEXPART analyses of moisture sources use such a global initialization. While the framework is equally applicable to regional domains with a homogeneous initial condition (and a corresponding boundary condition), it is not applicable to point/column releases. In the latter case, not only the mass of each parcel changes with time; but the (unconditional) detection of fluxes is impossible — and further prohibits a source bias correction. **We removed this statement from this sentence (as the rest of the sentence appears equally applicable to other initialisations) and added a few sentences to the methods that highlight that our framework is designed for homogeneous parcel distributions.**

L. 65: "discounting in a linear manner": I do not find the choice of the word "linear" entirely intuitive. There could be a clearer separation between the calculation of fractional contributions and the discounting in case of precipitation in this paragraph (see main comment #1).

We agree on the separation issue (see our reply above). We note, however, that both steps of the linear discounting/attribution, apply some linearity assumption: (i) moisture (or heat) uptakes between a source and a receptor are 'discounted' by any losses en route — using the ratio of uptake to specific humidity content of the air parcel; and similarly, (ii) the contribution of a source region to the final precipitation event is also estimated using the ratio of the (discounted) uptake to specific humidity content of the air parcel, multiplied with the 'final' moisture loss through precipitation.

L. 68: Sodemann and Stohl (2009) used the dq threshold of 0.1 for polar regions, and evaluated the sensitivity to trajectory length for such regions. 15 days seemed to be a lower limit here, which may be important for the results obtained in Fig. 1 for polar regions. The ABL/no ABL distinction has been topic also in Winschall et al., 2014 and in Sodemann and Stohl (2009).

We wish to emphasize that the global diagnosis as presented in Fig. 2 (we believe that the reviewer is referring to this Figure when referring to polar regions) is based on two consecutive time steps only — the results in Figs. 2–3 are thus independent of the trajectory length. Only the estimation of source region contributions is based on longer trajectories, i.e., up to 15 days (60 consecutive time steps backward from the receptor region). While we agree that trajectory lengths may play a role, our source region analysis is restricted to the cities of Denver, Beijing and Windhoek and hence does not consider polar regions. Further, we find that the linear discounting/attribution often shortens remote impacts, especially for moisture — which is in line with the rather short (average) residence times from Läderach and Sodemann (2016). Consequently, we do not necessarily see the need to examine other trajectory lengths. **We added to the cited sentence that the dq thresholds were calibrated; and we mention that also Sodemann and Stohl (2009) and Winschall et al. (2014) evaluated all sources including above-ABL uptakes.**

L. 80: I would express this a bit more nuanced, in that Sodemann (2020) propose to consider the lifetime distribution, and highlight that the long lifetimes that are part of the mean of the distribution are beyond reach or highly uncertain for Lagrangian diagnostics. In addition, the highly skewed lifetime distribution is probably more appropriately described by its median (as is commonly done for highly skewed distributions). See also the recently published review paper by Gimeno et al., 2021.

**We revised this sentence and updated the reference.**

Figure 1: I like Figure 1 in that it clarifies the flow of the analysis. Could it be possible to add information on the different forms of uncertainty entering the diagnostic, such as from the trajectory calculations, the detection, the attribution etc., that then add to total uncertainty?

**We updated this figure** but wish to remain restricted to the uncertainty inherent in the evaluation of trajectories. While **we now elaborate on additional uncertainties in the trajectories/simulations in the text** (see previous replies; and replies to other reviewers), we refrain from adding this uncertainty to the figures, as it would divert from our objectives and workflow. Further, and also in response to other comments, **we added figures on the attribution differences between linear attribution (assuming perfectly-mixed conditions) and random attribution (allowing for deviations from perfectly-mixed conditions) to the main text and the supplementary material.**

In Eq. (2), A needs to be defined, and something be said about A and m are determined in the analysis shown in the results part.

The area $A$ is defined in l. 173. For the analysis, we introduce the area / receptor region around l. 408 ("a 3°x3° box around each city center is used as a receptor area"). However, **we added a description of the parcels mass $m$ for our global FLEXPART simulations with 2 million air parcels, and explicitly mention the area $A$ for the three regions in the methods section and the captions of Figs. 6+10.**

L. 182: I do not understand why a distinction is made between f_z over land and ocean, there may be a misunderstanding of the relevant passage in S08, but if f_z has been applied, an f_z = 1.5 has always been applied uniformly over land and ocean.

We thank the reviewer for clarifying! This was indeed not clear to us. **We fixed the corresponding sentences in the revised version of the manuscript.**

Eq. (7): The maximum RH change criterion is not yet obvious to me. You state that "large RH changes are typically associated with ABL growth and warming": why is that inconsistent with evaporation?

It is not 'inconsistent' with evaporation. However, one could argue that the relative humidity can change only because the ABL is growing and warming. To clarify: we wanted to question the criteria already published and provide complementary alternatives — to gauge the full uncertainty inherent in these criteria. If only Delta(q) values larger than a minimum are considered for the detection of E (as in Sodemann et al., 2008), one could also question the appropriateness: (i) because Delta(q) displays the difference between e and p, and hence Delta(q) will often be smaller than e (or E); and (ii) because also evaporation can be small. Consequently, (many) small uptakes should be considered as well. To assess the uncertainty associated with that, we wanted to introduce a criterion that does the opposite and filters for a maximum increase.

L. 260: How dependent are your verification results on the chosen thresholds?

The results do, of course, depend on the thresholds for verification. However, we chose small thresholds to evaluate if any parcel detects the flux — which is better than no detection at all, as no detection cannot be bias-corrected. We tested several small thresholds (e.g., 0.01 mm day$^{-1}$, 0.1 mm day$^{-1}$, 1 mm day$^{-1}$) and found minor differences for the global evaluation (Figs. 2-3). Larger fluxes, using larger thresholds, are instead more difficult to detect (not at least because of the e-p issue mentioned above). Such an analysis is, however, out of scope for this study. The chosen thresholds represent adequate values that highlight the differences between the results.

L. 269: This reads as if Sodemann (2020) said that 15 days is a proxy for the globally averaged maximum lifetime. I may have overlooked it, but I do not find this statement in the cited paper.

**We deleted the reference from this sentence.**

Sec. 2.3.1: I believe this description would be clearer by separating into the fractional accounting of the arrival precipitation, and into the discounting due to precipitation en route (major comment #1).

We partly agree. This differentiation is already mentioned in Section 2.31: we define the discounting first (Eq. 13), before the 'attribution' of source region contributions to a sink quantity is introduced (Eq. 14). However, we understand that the term 'contributions' also for the former (e.g. in l. 286) is misleading and **we added better definitions to and revised our wording in this section.**

Sec. 2.3.2: The reasoning in this section is hard to follow. Could you clearer lay out the idea behind the random attribution, and contrast to the idea of the well-mixed air parcel (not well-mixed atmosphere) in the "linear" (or fractional/sequential/well-mixed) accounting? Maybe an example would also help.

We thank the reviewer for this suggestion. **As detailed above, we revised our description of the random attribution, added figures to the main manuscript and the supplementary material that highlight the differences between both attribution methodologies, and now discuss the results in the context of the well-mixed assumption.**

For the reviewer's interest, we also wish to highlight that the random attribution could also be performed on the 'discounted' uptakes along a trajectory (that take losses *en route* into account) — and thus could be used to evaluate the fractional or 'linearity' assumption in the attribution step. This is, however, out of scope for our current study.

L. 346-365: I find the question of bias correction for evaporation quite intriguing. If studies indicate an overestimation, this would cause too local sources (due to overly large contributions at each time step). Potentially it would make sense to mention this already here? Note that such bias correction as applied here is only possible for global-scale studies, at least local studies suffer from the fact that only the share of evaporation contributing to a certain region is diagnosed.

We fully agree with the statement that the overestimation of fluxes could indicate an overestimation of these source regions — if one wishes to identify ocean/land sources only. We understand, however, that the conceptual idea behind some studies is not restricted to oceanic and land origins and instead evaluate large-scale convergence and divergence zones of the vertically-integrated moisture transport; thus also comprising, e.g., phase changes and above-ABL mixing as sources of moisture. Therefore, we do not wish to dive into this discussion and restrict our analysis to the detection of surface source regions — and indeed, using global-scale or regional-scale studies, which track air parcels that are homogeneously distributed over a specific domain. **As noted above, we clarified this restriction.**

L. 386: Reanalysis data include humidity perturbations from data assimilation (Läderach and Sodemann, 2016), which are another source of uncertainty of these diagnostics, and one of the motivations for using a (large enough) threshold value for humidity changes.

We agree with the reviewer that the data assimilation impacts humidity perturbations and that this depicts another source of uncertainty. However, as outlined in one of our previous replies, one could also argue against a minimum threshold for the detection of evaporation from humidity changes. As we wish to evaluate the overall uncertainty arising from these thresholds, we only wish to evaluate the 'post-processing' uncertainty and refrain from diverting towards the uncertainty inherent in the setup of the simulations (such as the number of parcels and/or the driving forcing).

L. 435: The verification is done using your newly introduced additional thresholds. Here a clearer separation from the verification framework introduced just before will be useful.

We are very sorry, but we do not understand this comment. If the reviewer believes that this is important, we kindly ask him to clarify what he means.

Fig. 2: How do the results in Fig. 2 compare to the same kind of evaluation for the S08 method?

The average verification statistics are quite similar (cf. for example Fig. S2, which shows the same maps but for ALL-ABL). We designed the presentation part of the manuscript in such a way that Fig. 2 shows the spatial patterns, and Fig. 3 highlights the differences for all applied detection criteria. We do not intend to include all figures for all criteria in the manuscript, as it would be overwhelming. We are happy to share all our results with the reviewer though.

L. 449: This paragraph starts with the conclusion, before presenting the facts. Consider reverting the order of the paragraph.

We believe that this sentence is a good summary of the findings and makes the text easier to comprehend, and it aligns with the findings from the previous section. It further only refers to the validation of precipitation, whereas the following paragraphs highlight the verification of E and H.

L. 462: This statement seems to conflict with the statement in L. 449.

Each paragraph in section 3.1.2 describes the validation of one flux only. I.e. the first paragraph discusses the validation of precipitation (l. 449–452, Fig. 3a), the second paragraph discusses the validation of evaporation (l. 453–462, Fig. 3b), and the last paragraph discusses the validation of the sensible heat flux (l. 470–478, Fig. 3c). Thus we do not see how these statements conflict.

Fig. 3: I believe the regional results here are obtained with fixed dq thresholds. To what extent do the findings argue for the need to adapt the method to a specific study region?

The reviewer is right that the results in Fig. 3 are obtained with a fixed dq threshold for SOD08 and FAS19 — but also fixed RH thresholds for the RH criteria. It is also true that this figure could be used to argue that a (regional) calibration of these thresholds is needed. We wish to emphasize that none of the presented thresholds is calibrated though. Instead, our intent is to assess the overall uncertainty from, at base, different types of criteria. **As noted above, we do now emphasize that all criteria use a fixed threshold and that especially the dq-thresholds from Sodemann et al. (2008) and Fremme and Sodemann (2019) were calibrated for the respective study regions.**

L. 497: Typo in "heat"

We thank the reviewer for noticing. **The typo was fixed.**

Fig. 4, 7, 8 and similar: A more distinct colour bar, with a clearer separation from white will print better. Consider using less colour categories to allow reading off numbers/categories.

We thank the reviewer for this suggestion; however, a broader classification does not emphasize the differences between the approaches — which was our intent here (which could also be done plotting differences, as mentioned by another reviewer, but we do not want to use just one criterion as 'the' reference). We have published the data along with the manuscript, which allows any reader to analyze differences on their own. However, **to better highlight the different structure of the source regions, we have introduced purple colors**

**for large contributions, and added the outline of the 1-mm-precipitationshed to each plot. Similarly, we adjusted the color scheme for the heat source regions and outline 1-Wm$^{-2}$-heatsheds.**

Fig. 4, lower row: these graphs are almost identical. Do you have an explanation why the source correction is overriding the diagnostics so strongly?

Yes. The source correction has a strong impact on the estimated source region contributions for two reasons. First, the trajectories that are evaluated are the same. The criteria differ only in the identification of source locations along that trajectory and the corresponding attribution. Second, the applied criteria are emblematic of a systematic overestimation (e.g., SCH20 and ALL-ABL) or a systematic underestimation (e.g., SCH19) of the sensible heat flux. Thus, on average, the same source locations are identified — and the source bias-correction does exactly what it is supposed to do: it removes the systematic over- and underestimations of the applied approaches.

Fig. 6: This figure may be more informative as a table, maybe with the addition of numbers for the bias corrected results.

We believe that Figures are easier to comprehend than tables (in this case, this would be a 12x3 table) — and wish to highlight that all numbers are explicitly mentioned in the text. We thus wish to refrain from replacing this figure with a table; hopefully the reviewer is fine with it. **However, we added a Table to the Supplementary Material (Tab. S2) that highlights the numbers associated with precipitation origins (now Fig. 11 in the main manuscript) in comparison for the bias correction methodologies.**

L. 532: Change to "There are ..."

We believe that the sentence is correct and does not require changes.

L. 540: I am not used to the term "recycling" for heat, is this a well established expression?

We thank the reviewer for mentioning this; as this is indeed not well described in the current version of the manuscript — and as we do not believe that this is a well established expression (yet). While we understand that 'recycling of heat' is not as intuitive as the recycling of moisture, energy can be recycled. **Thus, we kept this formulation.**

L. 572: In what sense do you find the similarity of the source region maps reassuring?

While the 'random attribution' does follow physical limits (e.g., through a maximum uptake set by the minimum specific humidity content of the air parcel), it contains two random factors: the source location and the magnitude of its contribution is sampled in an iterative procedure until the sink quantity is fully attributed to a set of source locations. Thus, we believe it is reassuring if the identified source regions are similar in shape and magnitude, adds credibility to the approach. **We removed 'which is reassuring' from this sentence as it does not add much.**

Fig. 10: The colors are very similar and do not print well on all printers. Consider using patterns or a white/light region in the middle segment.

We appreciate the comment — and it is true that this color scale may not print well on all printers; but it is at least colorblind safe. **Thus, we decided to keep the original color scheme.**

Sec. 4 (Discussion): This section needs a clearer distinction between the part of the study dealing with a verification framework, and with a modified accounting method. Consider combining the Discussion with the Conclusion section, which is now rather a summary of the study, similar to the abstract. You could also list the main findings again as bullet points to facilitate grasping the take-away messages for the reader.

The discussion is already subdivided into two parts related to the comments from the reviewer: the first paragraph discusses the verification results; i.e., the different criteria and their accuracy and reliability for detecting surface fluxes and precipitation and how we expect the detection of fluxes to change with, e.g., other forcing data sets, or a changing number of parcels. This is, however, discussed in the context of source region estimation — which is the overarching motivation for this study and the verification exercise. The remaining paragraphs then discuss the source region estimation and additional uncertainties related to that. These parts do not only deal with the impact of the attribution methodology (linear discounting and attribution in comparison to random attribution), and put these into the context of the residence time, but also with the impacts of bias correction and the potential to further develop the framework.

From our perspective, a clear separation of this section to the 'Summary and Conclusions' section is needed. The discussion really focuses on issues that have not been dealt with in this manuscript, it provides an outlook on further applications and developments, and it remains — to a large extent — speculative. Thus, we wish to keep these sections separate if possible.
**We thank the reviewer for the suggestion on the bullet-point list — while we initially intended to follow the suggestion, we decided to keep the textual structure in the end. We hope the reviewer is okay with that.**

L. 655: This seems to fit better to the conclusions than the discussions (or could re- appear in the conclusions)

**We added some issues raised during this review to the discussion and rephrased parts of the summary — but kept the summary constrained to tangible results from this study.**

L. 677: I think Sodemann (2020) does not claim that the discrepancies is entirely an issue of definition, see the comment to L. 80 above.

**We revised this sentence so that it does not appear as if Sodemann (2020) claimed it.**

L. 707: Given the lack of a real theoretical basis for the random accounting, I would formulate this conclusion more carefully. There is certainly uncertainty in the accounting, but how large the uncertainty stemming from the accouting is in relation to the overestimation of evaporation is not finally answered from your study - and deserves further investigation.

As indicated above, there is a physical basis for the random attribution. We do not claim that the comparison of both attribution methodologies spans the uncertainty inherent in the source region/contribution estimation. But as long as alternative and validating measures are lacking, we believe that the random attribution is a valid alternative to assess some

uncertainty. Nevertheless, it is true that further investigation, and observations, are needed to unravel the 'true' sources of uncertainty.

**Supplemental material**
Sec. 4: I could imagine this section to better be placed in the main manuscript (see major comment #1).

Maybe we misunderstood comment #1, but we do believe that these are two different issues. From our perspective, the differentiation between discounting and attribution is already mentioned in the main manuscript (**but we better highlight it now**; see our reply above). The issue described in section 4 of the supplementary material, is a different one. Here, we mention that not all precipitation can be attributed to the identified source regions. While this is, of course, caused by and related to the linear discounting and attribution – it remains a different issue. We decided to move this part to the supplementary material, as (i) the bias correction fixes this issue; and because of that, (ii) we do not show any differences related to this discrepancy in our results. We have done some analysis in this regard and would be happy to share it with the reviewer; but we believe that a thorough investigation is beyond the scope of the manuscript.

L. 123: "Contrary to Sodemann...": It is not entirely clear what you consider to be the sources of the ABL uptakes, if not convective detrainment of BL air into the free troposphere, and on what basis you make your argument here. A more direct reference for the cited statement is Winschall et al., (2014).

For our reply here, we assume that the reviewer means free troposphere uptakes (i.e., \*above\* ABL uptakes).

Our general intention is to identify the surface source regions of moisture only. Therefore, we wish to identify air parcels that are *directly* influenced by surface evaporation, which we assume to be the most dominant source of moisture in the ABL. For a sufficiently high temporal resolution, changes in specific humidity of ABL air should reflect this evaporation (disregarding phase changes for simplicity), even if some of the evaporation ends up detrained through convection. With the 6-hourly time steps from ERA-Interim, however, it proves difficult to disregard all parcels that strictly reside within the ABL. Hence, with the approach presented in the manuscript, we identify air parcels that reside within the maximum ABL of two time steps during at least one time step. As such, we might actually also sample some of the detraining air parcels — if the RH criteria are applied, however, we still disregard parcels that have already (strongly) mixed with tropospheric air and thus exhibit a very different relative humidity. Similarly, we also account for a few entraining air parcels that, e.g., gain moisture through mixing with ABL air masses. However, we wish to exclude air parcels that simply gain moisture through mixing; e.g. an air parcel that just passes over the ABL but gains moisture through mixing and/or convective detrainment. From our perspective, these above ABL moisture sources represent *indirect* moisture sources — whose surface source might be farther away than the place where the mixing occurs. Thus, we assume that these are not representative of ABL processes anymore and that further (secondary) tracking of these air masses would be required to identify the corresponding surface source.

**We thank the reviewer for the alternative reference, which we added to the manuscript**.

**References**
Gimeno, L., Eiras-Barca, J., Durán-Quesada, A.M., Dominguez. F., van der Ent, R., Sodemann, H., Sánchez-Murillo, R., Nieto, R. and Kirchner, J. W.: The residence time of water

vapour in the atmosphere. Nat. Rev. Earth Environ., https://doi.org/10.1038/s43017-021-00181-9, 2021.

Winschall, A., Sodemann, H.,Pfahl, S. and Wernli, H., 2014: How important is intensified evaporation for Mediterranean precipitation extremes?, J. Geophys. Res., 119: 5240–5256, doi:10.1002/2013JD021175

Sodemann, H.and Zubler, E., 2010: Seasonality and inter-annual variability of the moisture sources for Alpine precipitation during 1995-2002, Int. J. Climatol., 30: 947-961, doi:10.1002/joc.1932.

Sodemann, H., and Stohl, A., 2009: Asymmetries in the moisture origin of Antarctic precipitation, Geophys. Res. Lett., 36, L22803, doi:10.1029/2009GL040242.

**References**

Läderach, A., and Sodemann, H. (2016). A revised picture of the atmospheric moisture residence time. Geophysical Research Letters, 43(2), 924-933.

Sodemann, H. (2020). Beyond turnover time: constraining the lifetime distribution of water vapor from simple and complex approaches. Journal of the Atmospheric Sciences, 77(2), 413-433.

Sodemann, H., Schwierz, C., and Wernli, H. (2008). Interannual variability of Greenland winter precipitation sources: Lagrangian moisture diagnostic and North Atlantic Oscillation influence. Journal of Geophysical Research: Atmospheres, 113(D3).

Tuinenburg, O. A. and Staal, A. (2020). Tracking the global flows of atmospheric moisture and associated uncertainties. Hydrological Earth System Sciences, 24, 2419–2435.

---

## Referee Report (RR1)

2nd review of paper

**A unified framework to estimate the origins of atmospheric moisture and heat using Lagrangian models**

by J. Keune et al.

submitted to *Geosci. Model Dev.*

Thanks to the revisions this became in my view an excellent paper that is innovative, well written and of great value for the growing community that cares about diagnostics of moisture and heat sources. I congratulate the authors to this comprehensive and important study and recommend accepting the paper with only very few corrections. In particular, I very much like the introduction and the clear and detailed description of the methodologies. The idea of the "random attribution method" is now much clearer to me (and an interesting idea!). Also, I very much appreciate the examples shown in Fig. 2 – very helpful!

**Minor comments**

1) Line 13: I don't understand "multi-backward-day trajectories" (the term is also not used/explained in the main part of the paper). Do you mean "multi-day backward trajectories"?

2) Line 126: I think all years should be in (…).

3) Fig. 1: lowest panels: not fully clear to me what red and blue indicates.

4) Line 180: I assume that $H$ here is heat flux, why is it underlined?

5) Eq. 13: potentially confusing to use H (heat flux) also for the Heaviside function, maybe use calligraphic H?

6) Line 343: different notation for Heaviside function compared to eq. 13.

7) Eq. 17: I assume that h should be italic.

8) Line 416: here and in other places, I think that t should be italic.

9) Line 803: space missing in "diagnosis enables".

10) Line 806: "to use relax detection criteria" sounds strange, either "use relaxed … criteria" or "relax … criteria".

11) Line 1020: initials should be only "H." not "B. H.".

---

## Author Response (AR2)

Topical Editor — Simon Unterstrasser
*received: 07 Feb 2022*

Dear authors,

Originally I planned to have the revised manuscript be reviewed by two reviewers of the first round. As the second late is still missing and ,more importantly, the existing review is enthusiastic about your efforts during the revision, I am happy to proceed.
The reviewer again made valuable comments, this time mostly about terminology.
Please include those last improvements and then I am happy to accept the manuscript.

Best wishes,
Simon

Dear Dr. Unterstrasser,

Thank you again for handling our manuscript entitled "A unified framework to estimate the origins of atmospheric moisture and heat using Lagrangian models" (gmd-2021-180).

We have addressed the remaining technical comments raised by the reviewer and corrected some additional minor style inconsistencies (replacing ',' with ';' and 'Fig.' with 'Figure' at the beginning of a sentence). Please find attached a short response to the comments along with a markup version of the manuscript.

We hope that these revisions make the manuscript eligible for publication in GMD.

Kind regards,

Jessica Keune (on behalf of both co-authors)

Ghent, 08 February 2022

RC1 — Anonymous Referee #1
*received: 07 Feb 2022*

2nd review of paper

**A unified framework to estimate the origins of atmospheric moisture and heat using Lagrangian models**

by J. Keune et al.

submitted to *Geosci. Model Dev.*

Thanks to the revisions this became in my view an excellent paper that is innovative, well written and of great value for the growing community that cares about diagnostics of moisture and heat sources. I congratulate the authors to this comprehensive and important study and recommend accepting the paper with only very few corrections. In particular, I very much like the introduction and the clear and detailed description of the methodologies. The idea of the "random attribution method" is now much clearer to me (and an interesting idea!). Also, I very much appreciate the examples shown in Fig. 2 – very helpful!

We are very grateful to the reviewer for their time that they spent reviewing our study and for the kind words that express their appreciation. We are happy to report that we incorporated the remaining technical corrections as suggested. Our reply to all comments is detailed below.

**Minor comments**

1)  Line 13: I don't understand "multi-backward-day trajectories" (the term is also not used/explained in the main part of the paper). Do you mean "multi-day backward trajectories"?

We thank the reviewer for noticing this terminology conundrum. It should indeed be "multi-day backward trajectories". We have corrected this notation.

2)  Line 126: I think all years should be in (...).

Indeed. We corrected the style of the references.

3)  Fig. 1: lowest panels: not fully clear to me what red and blue indicates.

We added a description to the caption of Fig. 1 that describes the color scheme qualitatively:

> *"The images at the bottom show the bias of the Lagrangian simulation (red colors indicating an overestimation of the local flux, blue colors indicating an underestimation of the local flux) as estimated in the diagnosis step (left), the biased source regions as estimated in the attribution step (middle; the grey cell indicates the receptor region), and the bias-corrected source regions after the bias correction step (right). The color intensity indicates the magnitude of the bias and the source region contribution. Note that '+' ('–') signs indicate substantially lower (higher) values in the Lagrangian simulation for easier reference across all steps."*

4)  Line 180: I assume that *H* here is heat flux, why is it underlined?

*H* is indeed the sensible heat flux, thus we have removed the underline.

5) Eq. 13: potentially confusing to use H (heat flux) also for the Heaviside function, maybe use calligraphic H?

Thank you for noticing! We replaced the Heaviside H with the calligraphic H ($\mathcal{H}$), as suggested.

6) Line 343: different notation for Heaviside function compared to eq. 13.

Also here: thank you for noticing! We adjusted Eq. 13 to use subscripts as well.

7) Eq. 17: I assume that h should be italic.

We assume that the reviewer meant Eq. 16 – which included a non-italic h. We corrected the font in Eq. 16 and thank the reviewer for noticing!

8) Line 416: here and in other places, I think that t should be italic.

Yes, it should. We checked the entire manuscript for 't' and made sure that these are in italic font.

9) Line 803: space missing in "diagnosis enables".

Thank you. We added a space.

10) Line 806: "to use relax detection criteria" sounds strange, either "use relaxed ... criteria" or "relax ... criteria".

Thank you. We corrected the sentence to use "relaxed criteria".

11) Line 1020: initials should be only "H." not "B. H.".

Thank you for noticing. We are very sorry for the mistake. We corrected this reference.

[revised manuscript text omitted]